# Multi-Level Strategic Classification: Incentivizing Improvement through Promotion and Relegation Dynamics

**Ziyuan Huang** [*] [1]   **Lina Alkarmi** [*] [1]   **Mingyan Liu** [1]

## Abstract

Strategic classification studies the problem where self-interested individuals or agents manipulate their response to obtain favorable decision outcomes made by classifiers, typically turning to dishonest actions when they are less costly than genuine efforts. While existing studies on sequential strategic classification primarily focus on optimizing dynamic classifier weights, we depart from these weight-centric approaches by analyzing the design of classifier thresholds and difficulty progression within a multi-level promotion-relegation framework. Our model captures the critical inter-temporal incentives driven by an agent's farsightedness, skill retention, and a "leg-up" effect where qualification and attainment can be self-reinforcing. We characterize the agent's optimal long-term strategy and demonstrate that a principal can design a sequence of thresholds to effectively incentivize honest effort. Crucially, we prove that under mild conditions, this mechanism enables agents to reach arbitrarily high levels solely through genuine improvement efforts.

## 1. Introduction

Knowledge of a decision process (e.g., what type of input it uses and how the input determines the decision outcome) is empowering in two opposing directions for individuals subject to the decision and desiring a certain outcome: they can use that knowledge to strategically equip themselves with the "right" type of input/qualifications either through honest means (making an effort to attain those qualifications) or dishonest means (faking the appearance of those qualifications), or a mixture of both. For example, a student may focus all their effort on a particular chapter if they know

that chapter carries a disproportionate weight on an exam (honest), or a job-seeker may artificially alter their resume to meet a hiring criterion (dishonest). The rapid rise of algorithmic decision making fueled by advances in machine learning and AI, and its increasingly common adoption in a multitude of domains such as loan approval, admissions, and hiring (Hardt et al., 2015; Haghtalab et al., 2020), has only amplified this problem: on one hand, consumers and regulatory agencies rightfully demand algorithm transparency, especially for those used to make high-stake decisions; on the other hand, more transparency invites more strategic response/manipulation, for better and worse.

This has given rise to the growing field of strategic classification that studies utility-maximizing agents who respond strategically to a (decision-making) classifier (Hardt et al., 2015; Miller et al., 2020). Such an agent is often characterized by an *observable feature*, an *unobservable attribute* which is its private information, and the ability to take actions to *improve* both (Kleinberg & Raghavan, 2019; Milli et al., 2019; Ahmadi et al., 2022; Jin et al., 2022), or to *game*, which only improves the former (Milli et al., 2019; Miller et al., 2020; Jin et al., 2023). This is often formulated as a one-shot interaction between the decision maker and the agent, and the most significant result from this literature is that a strategic agent takes the least costly action so as to cross a decision boundary (Hardt et al., 2015; Braverman & Garg, 2020; Jin et al., 2022; 2023). Combining this with the natural assumption that cheating/gaming is in general less expensive than making an honest effort leads to the fundamental challenge that incentivizing improvement is infeasible (Milli et al., 2019; Jin et al., 2021) without external instruments such as subsidies (Jin et al., 2022).

The key to overcoming this challenge without relying on external mechanisms is to consider repeated interactions with a sequential formulation, which introduces what's known as *inter-temporal incentives*, whereby past actions can impact future outcomes, which does not exist in one-shot settings. Within this context, this paper introduces a novel sequential model of *multi-level* strategic classification that captures how agents behave as they move through a set of classifiers with increasing difficulty and increasing reward. Each level is modeled as a *selective classifier* (Cortes et al., 2016;

---
[*]Equal contribution  [1]Electrical and Computer Engineering Department, University of Michigan, Ann Arbor, MI 48109, USA. Correspondence to: Ziyuan Huang <ziyuanh@umich.edu>.

*Proceedings of the 43rd International Conference on Machine Learning*, Seoul, South Korea. PMLR 306, 2026. Copyright 2026 by the author(s).

Geifman & El-Yaniv, 2017; Lee et al., 2021; Alkarmi et al., 2025), that can abstain from making a prediction when confidence is low. A positive (resp. negative) decision leads to promotion (resp. relegation) of the agent to the next higher (resp. lower) level, while abstention keeps the agent at the same level. Such a sequential, multi-level setting is very common in practice: e.g., a first midterm is followed by a second midterm and a final exam, whereby how much one prepares for the midterms may well impact one's posture heading into the final; skill certifications require increasing breadth/depth of knowledge; other examples abound.

This model is thus a dynamic one, where the agent's state, consisting of its (unobservable) attributes, (observable) features, and its level (of attainment), evolves over time, driven by the agent's action as well as three effects coupling successive stages of the interaction: (i) the agent's *farsightedness*, given by a discount factor that defines how much the agent cares about future reward relative to instantaneous reward, a common feature in sequential problems, (ii) the *retention* of attribute, given by a retention factor that reflects depreciation or degradation of skills or knowledge over time without sustained effort, another common feature in the dynamic decision-making literature (Alston, 1998; Dohmen & Trivedi, 2023; Hastings & Sethumadhavan, 2025), and (iii) a *leg-up* effect, where an agent's attribute is boosted by higher level of attainment. We characterize the agent's optimal long-term strategy under various conditions, and examine the type of classifier design that effectively incentivizes a strategic agent to attain an arbitrarily high level solely through honest means, even though gaming remains the cheaper, instantaneous option.

There have been other studies exploring the use of intertemporal incentives in a similar context. The most relevant to our work is Harris et al. (2021), where a designer designs a set of regression models and a strategic agent aims to maximize the sum of its regression scores while taking advantage of the cumulation of effort. In contrast, our work focuses on classification and, more distinctively, the sequentiality inherent in a progression of classifiers. Furthermore, we model the passage of time, reflected not only in the cumulation of effort but also in the discounting of reward as well as the degradation of the agent's attribute over time, features that are absent in Harris et al. (2021). Additional literature review is provided in Section A.

Our main contributions are summarized as follows.

1. We formulate a sequential strategic classification problem with a progression of classifiers with increasing difficulty and reward. We show how the incentivizable region is strictly larger than that under a one-shot formulation, and interpret the role of each of the three effects (discount, retention, and leg-up).

2. We fully characterize the agent's optimal long-term strategy in the two-level case, analyze the agent's trajectory in different situations, and pinpoint the importance of the leg-up effect in incentivizing higher attributes.

3. We examine the feasibility of the designer/principal's problem and derive conditions under which pure improvement actions are incentivizable up to some target attribute. We design a sequence of classifiers and show that, under mild conditions, it can induce an agent to attain an arbitrarily high level through honest efforts.

The remainder of this paper is organized as follows. Section 2 formulates the problem. Section 3 characterizes the agent's optimal long-term strategy in the two-level (single-classifier) case. Section 4 analyzes the optimal classifier sequence design in the multi-level setting. Section 5 presents numerical results, including ablation studies and real-data simulations. Section 6 discusses the findings.

## 2. Model and Preliminaries

**The model** We consider an agent repeatedly interacting with a progression of $L \in \mathbb{N}$ classifiers, each representing a *level*, starting at the lowest. Each level is effectively a test uniquely associated with a classifier: passing the test allows the agent to move up to the next (higher) level (a promotion); failing it sends the agent back to the previous (lower) level (a relegation); an inconclusive (less confident) test result leaves the agent at the same level (no change in status). A higher level comes with a higher reward. The agent is reward-seeking, and can exert a costly *improvement* effort or a less costly *gaming* (cheating) effort, either of which (or a mixture of both) in sufficient quantity will lead to its passing a test but only the former results in an actual improvement in the agent's true qualification. The system designer (also referred to as the principal)'s objective is to design the sequence of classifiers to incentivize the agent to remain honest (i.e., choose improvement over gaming) over time. Below we detail the components of this sequential decision problem as well as the underlying dynamics illustrated in Figure 1.

**The agent** Let $t \in \{0\} \cup \mathbb{N}$ denote the discrete *time* (or *stage*) index. Consider an agent who possesses an (unobservable) *attribute* $x_t$ and an (observable) *feature* $z_t$, both in $\in \mathbb{R}_{\geq 0}$[1] Only the latter is taken as input to the classifiers, while the former is the agent's private knowledge. The agent can exert an *effort* $\vec{a}_t := [a_t^+, a_t^-]^\top \in \mathbb{R}_{\geq 0}^2$ in order to secure a favorable classifier outcome, where $a_t^+$ (resp. $a_t^-$) denotes an *improvement* (resp. *gaming*) action, with a unit cost $c^+ \geq 0$ (resp. $c^- \geq 0$). We adopt the common assumption that cheating is cheaper than improvement, i.e., $0 < c^- < c^+ < \infty$.

---

[1]This is extendable to multi-dimensional settings (Section 6).

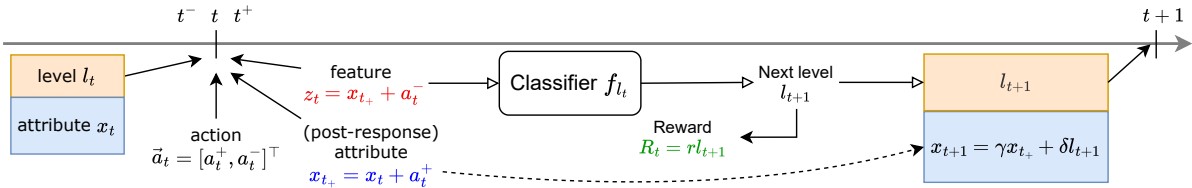

*Figure 1.* Dynamics of the Agent's Decision Process.

Only the improvement action leads to an inherent change to the agent's attribute. That is, following the action $\vec{a}_t$ chosen at time $t$, the agent attribute becomes $x_{t_+} = x_t + a_t^+$.[2] and $z_t := z_{t_+}$ throughout; in particular, the post-action feature is $z_t$ (not $z_{t+1}$).

In contrast, the feature $z_t := x_t + a_t^+ + a_t^-$ is driven equally by improvement and gaming actions, reflecting their indistinguishable effect on the immediate outcome.

After its action at time $t$, the agent's attribute $x_{t_+}$ then undergoes two further modifications before the next time step, following two critical assumptions. (1) We assume the agent's attribute depreciates over time, so that only a $\gamma \in (0, 1)$ fraction of $x_{t_+}$ remains by $t + 1$; $\gamma$ is referred to as the *retention factor*. This models the fact that learned skills can degrade over time and more effort is required to keep them up to date. (2) We assume the agent receives a boost to its attribute in the amount $\delta(l_{t+1} - 1)$; this models an opportunity gain the agent receives via more/better learning and other resources often correlated with higher levels (e.g., a student scoring high on a previous exam may attract the teacher's attention and be called upon to answer questions more frequently, leading to enhanced learning). The constant $\delta \geq 0$ will be referred to as the *leg-up factor*. Note that $l_{t+1}$ is the level the agent attains after its effort at time $t$ and is thus the level that determines the reward it collects during the $t$-th time step as detailed shortly below. The choice of $(l_{t+1} - 1)$ is designed to rule out a boost at the lowest, entry level of $l = 1$ while allowing a constant boost of $\delta$ with each increasing level.[3] The overall transition of the agent's attribute is thus given by $x_{t+1} = \gamma x_{t_+} + \delta(l_{t+1} - 1)$.

**The principal** The principal deploys a sequence of $L$ ternary classifiers, which, in addition to the common binary decisions, can also *abstain* if uncertainty is deemed high. This abstention option can reduce decision error, albeit at the expense of lower coverage (Cortes et al., 2016; Franc et al., 2023). It turns out this classifier choice fits nicely in our sequential setting, where an agent can be promoted, relegated, or remain at the same level in accordance with

[2]By convention, $t_+$ (resp. $t_-$) denotes the time immediately after (resp. before) $\vec{a}_t$. We write $x_{t_+}$ explicitly when needed, but abbreviate $x_t := x_{t_-}$.

[3]This is equivalent to indexing the lowest level by $l = 0$ rather than $l = 1$ as we have done.

the ternary decision outcome. Specifically, the principal designs a sequence $\vec{\mu} := \{\mu_l\}_{l=1}^L$ representing the desired qualifications at each level. These are the thresholds that determine the classifier output $f_l(\cdot)$ and the movement of the agent:

$$f_l(z_t) = \begin{cases} 1 & \text{if } \theta z_t \geq \mu_{l+1} & \text{(pass/promotion)} \\ 0 & \text{if } \mu_l \leq \theta z_t < \mu_{l+1} & \text{(abstain/staying)} \\ -1 & \text{if } \theta z_t \leq \mu_l & \text{(fail/relegation)} \end{cases} \quad (1)$$

where $\theta$ is the model weight shared across levels. For agents at the boundary level $1$ or $L$, it remains at that level whenever $f_1(z_t) < 1$ or $f_L(z_t) > -1$, respectively. This transition is illustrated in Figure 2. By default, we set $\mu_1 \equiv 0$.

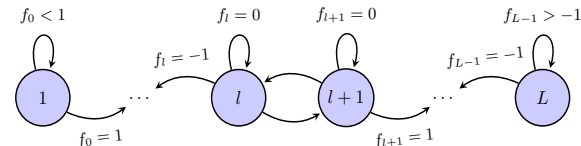

*Figure 2.* The evolution of the agent's level.

**The agent's problem** Each level $l$ is associated with a reward $R_l := r(l - 1)$ with $r > 0$. The agent receives a reward of $R_{l_{t+1}}$ following its action $\vec{a}_t$ and attaining the level $l_{t+1}$. It is easy to see that $\{(l_t, x_t)\}_{t \geq 0}$ is a discrete-time Markov decision process (MDP) with continuous state and action spaces. The dynamics of this process are illustrated in Figure 1. The agent's optimal strategy is the sequence of actions $\{\vec{a}_t^*\}_{t \geq 0}$ that maximizes its discounted total reward over an infinite horizon given the state $(l_t, x_t)$:

$$\{\vec{a}_t^*\}_{t \geq 0} \in \underset{\vec{a}_0, \vec{a}_1, \dots}{\arg\max} \sum_{t=0}^{\infty} \beta^t \left( R_{l_{t+1}} - \vec{c}^\top \vec{a}_t \right), \quad (2)$$

where $\beta \in (0, 1)$ is the discount factor. We will assume ties are broken in favor of improvement actions.

**The principal's problem** Given a target attribute $M \geq 0$ and the reward rate $r > 0$, the principal aims to design the shortest possible sequence $\vec{\mu}$ that satisfies three goals: (1) the agent acts honestly (never games); (2) the agent's long-term attribute exceeds the target $M$; and (3) the agent eventually attains the highest level. This is formulated as

the following constrained optimization problem $\mathbf{P}(M, r)$:

$$\min_{L, \vec{\mu}} \quad L$$
$$\text{s.t.} \quad (a_t^-)^* \equiv 0, \quad \forall t \geq 0$$
$$\liminf_{t \to \infty} x_{t_+} \geq M, \quad \forall (l_0, x_0) \in \{0\} \times \mathbb{R}_{\geq 0} \quad (3)$$
$$\lim_{t \to \infty} l_t = L, \quad \forall (l_0, x_0) \in \{0\} \times \mathbb{R}_{\geq 0}$$

where the limits are defined under the agent's optimal strategy, best-responding to the principal's design variable $(L, \vec{\mu})$ according to Equation (2). The solution $(L^*, \vec{\mu}^*, \{\vec{a}_t^*\}_{t \geq 0})$ constitutes a *subgame-perfect equilibrium* (SPE).

We note that $\theta$ is not a decision variable in $\mathbf{P}(M, r)$, and is assumed to be pre-determined (e.g., via estimation on non-strategic data). Without loss of generality, we let $\theta = 1$. The threshold inequality $\theta z \geq \mu_l$ is scale-invariant in $\theta$, and since $\theta$ is shared across all levels, any $\theta > 0$ is equivalent up to a common rescaling of thresholds.

Finally, note that the principal need not explicitly/separately optimize for strategic robustness, defined as the consistency of classifier accuracy under manipulation, e.g., $\mathbf{1}\{f_{l_t}(z_t) = f_{l_t}(x_{t_+})\}$. As the constraints of $\mathbf{P}(M, r)$ eliminate gaming actions (i.e., $(a_t^-)^* \equiv 0$), under any solution to this problem the observed feature $z_t$ coincides with the true attribute $x_{t_+}$, thereby guaranteeing the accuracy of the decision.

**An immediate result: a reduction in Non-Incentivizable Region** A direct consequence of our formulation is a critical reduction in the range of problem instances where genuine improvement is impossible to incentivize, compared to a one-shot setting[4].

**Proposition 2.1.** *The agent never chooses an improvement action if* $(1 - \beta\gamma)c^+ > c^-$.

Theorem 2.1 essentially says that with farsightedness and attribute retention, the agent's effective long-term unit cost of improvement is $(1 - \beta\gamma)c^+$, which is lower than $c^+$. So long as this is greater than the unit cost of gaming ($c^-$), it will never choose improvement effort, regardless of the classifier design $\vec{\mu}$. This is the limit of any mechanism in deterring gaming actions. Figure 3a illustrates this "impossibility region" (shaded red). Crucially, since $\beta, \gamma < 1$, this region is strictly smaller than the one-shot impossibility region given by $c^+ > c^-$ (Jin et al., 2021; 2022) (solid orange in Figure 3a). Effectively, the sequential dynamics reduce the true unit cost of improvement, thereby making it possible for the system to incentivize improvement in scenarios where a one-shot formulation would fail. The remainder of this paper will exclusively focus on the incentivizable region and assume that the following holds:

**Assumption 2.2.** $(1 - \beta\gamma)c^+ < c^-$.

---

[4]All proofs can be found in the appendix.

## 3. The Agent's Optimal Strategy in the Two-Level Case

In this section, we characterize the agent's optimal strategy in Equation (2) for $L = 2$, writing $\mu \equiv \mu_2$. Since $\mu_1 \equiv 0$, the agent interacts repeatedly with a single classifier at threshold $\mu$; the optimal Markov strategy is independent of $(l_0, x_0)$. Beyond capturing the one-classifier sub-case, this two-level analysis is the mathematical backbone of Theorem 5.1 (each greedy step reduces to a two-level subproblem) and can also be interpreted as a bounded-rationality model in which the agent plans one step ahead.

### 3.1. Agent Strategy for Small Threshold $\mu$

**Theorem 3.1.** *When* $\mu < \frac{\delta}{1-\gamma}$, $r \geq \frac{(1-\beta)c^- \delta}{(1-\gamma)(1-\beta\gamma)}$, *and* $c^- \geq \max\{\beta\gamma c^+, (1 - \beta\gamma)c^+\}$, *there exists* $x^\circ \in [0, \mu]$ *such that the agent's optimal strategy is the following:*

(1) *(gaming only)* $\vec{a}_t^* = [0, \mu - x_t]^\top$, *if* $x_t \in [x^\circ, \mu]$;
(2) *(mixture of improvement and gaming)* $\vec{a}_t^* = [x^\circ - x_t, \mu - x^\circ]$, *if* $x_t \in [0, x^\circ)$.

The above behavior is illustrated in Figure 3b, lower-left corner. Theorem 3.1 indicates that gaming actions cannot be fully discouraged when the threshold is too small. This is because a small threshold means more gain in attribute from leg-up than loss due to depreciation: $\delta > (1 - \gamma)\mu$. This leads the agent to apply at most an insufficient level of improvement and use (less costly) gaming to cover the remaining gap to the threshold, relying on the guaranteed future leg-up benefit. We will subsequently call this $\mu < \frac{\delta}{1-\gamma}$ region the *leg-up region*. From Figure 3b, this is the only region where a mixture of non-zero improvement and gaming can be optimal. The optimal strategy outside this region dictates only *one* of these actions exclusively.

### 3.2. Agent Strategy for Large Threshold $\mu$

**Theorem 3.2.** *There exists* $\underline{\mu}, \overline{\mu} \geq 0$ *independent of* $\delta$, *and* $\underline{x}, \overline{x} \in [0, \mu]$ *dependent on* $\mu$ *and* $\delta$, *such that*

(1) *if* $\mu \in [\frac{\delta}{1-\gamma}, \underline{\mu} + \frac{\delta}{1-\gamma})$, *then* $\vec{a}_t^* = [\mu - x_t, 0]^\top$ *if* $x_t \in [\underline{x}, \mu)$, *and* $[0, 0]^\top$ *otherwise*;
(2) *if* $\mu \in [\underline{\mu} + \frac{\delta}{1-\gamma}, \overline{\mu} + \frac{\delta}{1-\gamma})$, *then* $\vec{a}_t^* = [0, \mu - x_t]^\top$ *if* $x_t \in [\mu - \frac{r}{c^-}, \overline{x})$, $\vec{a}_t^* = [\mu - x_t, 0]^\top$ *if* $x_t \in [\overline{x}, \mu)$, *and* $[0, 0]^\top$ *otherwise*;
(3) *if* $\mu \in [\overline{\mu} + \frac{\delta}{1-\gamma}, \infty)$, *then* $\vec{a}_t^* = [0, \mu - x_t]^\top$ *if* $x_t \in [\underline{x}, \mu]$, *and* $[0, 0]^\top$ *otherwise*.

The upper portion of Figure 3b (i.e., $\mu \geq \frac{\delta}{1-\gamma}$) visualizes the agent behavior detailed in Theorem 3.2. The three disjoint sub-intervals on the $\mu$-axis in this region (separated by $\underline{\mu} + \frac{\delta}{1-\gamma}$ and $\overline{\mu} + \frac{\delta}{1-\gamma}$), in increasing order, correspond to cases (1)-(3) of Theorem 3.2, respectively.

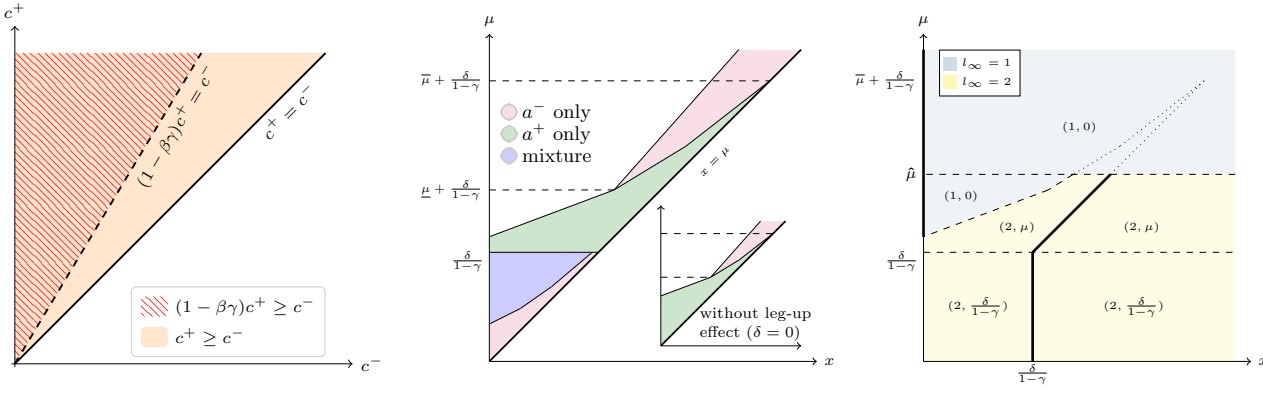

*(a)* Change in impossibility region.     *(b)* Agent's optimal strategy.     *(c)* Agent's steady state $(l_\infty, x_\infty)$ per region.

*Figure 3.* General impossibility region and the agent's optimal long-term strategy in the two-level case. (a) Reduction of the non-incentivizable region under the multilevel mechanism compared to the single-shot problem. (b) The agent's optimal (Markov) strategy for given threshold $\mu$ and attribute $x$. (c) The agent's steady state distribution for given initial state under the optimal strategy.

**Interpreting the Optimal Long-Term Strategy** Theorem 3.2 implies that the agent's optimal action depends on both the threshold and the proximity of its attribute to the threshold. In the low-threshold regime (case (1)), the agent exerts improvement efforts if the threshold is attainable (the green region) or else does nothing. In the intermediate regime (case (2)), the behavior is stepwise: the agent makes an improvement effort if the attribute is already close enough to the threshold, cheats if it's further away, or does nothing if it's too far away. In the higher regime (case (3)), the agent will cheat if the threshold is within reach or else do nothing, indicating the lack of improvement incentives. This occurs when the threshold is set above $\overline{\mu} + \frac{\delta}{1-\gamma}$, where maintenance costs outweigh the benefits of improvement.

Comparing the main figure to the small inlaid (lower right) in Figure 3b, we see that the leg-up effect shifts all these regions up by $\frac{\delta}{1-\gamma}$: as leg-up makes attaining the threshold more rewarding, it allows the principal to set a higher threshold. However, the net region in which effort can be incentivized (height of the green region, $\overline{\mu}$) remains constant.

**Trajectories and Steady State** Figure 3c depicts the agent's steady state (i.e., $(l_\infty, x_\infty)$) under the optimal strategy for each $x_0$ and $\mu$. The solid black lines mark the steady-state attribute, and the gray and yellow colors highlight the regions where $l_\infty = 1$ and $l_\infty = 2$, respectively. Notice that the dotted wedge in Figure 3c is the same as the boundary of the green region of Figure 3b.

When $\mu \geq \frac{\delta}{1-\gamma}$, an agent follows one of two trajectories: improving to $(2, \mu)$ (as agents with $x_0 > \mu$ initially coast without effort, exerting improvement only when depreciation makes the threshold binding), or dropping to $(1, 0)$.

Notably, Figure 3c identifies a threshold $\hat{\mu} < \overline{\mu} + \frac{\delta}{1-\gamma}$, above which the steady-state attribute collapses to 0 even

if the agent may have initially improved (if starting in the green region of Figure 3b). This is because the depreciation dominates: Even when an agent reaches $\mu$, the attribute in the subsequent time step ($\gamma\mu + \delta$) falls outside the improvement region. Conversely, for $\mu < \frac{\delta}{1-\gamma}$, the steady state converges to $(2, \frac{\delta}{1-\gamma})$ because $\mu$ is set low enough for the leg-up effect alone to sustain an attribute above $\mu$. In this sense, $\frac{\delta}{1-\gamma}$ may be viewed as a "natural equilibrium" of the attribute dynamics, independent of the principal's decision. We will see in Section 4 that this equilibrium is critical in designing high-achieving systems (incentivizing very high attributes while deterring gaming).

**Role of Retention and Discounting** $\underline{\mu}$ and $\overline{\mu}$ increase in $\beta$ and $\gamma$ (see Section C.4.2), but with a key asymmetry: $\gamma \to 1$ pushes both to $\infty$, fully eliminating the all-gaming region (case (3)), while $\beta \to 1$ only reaches the finite limit $\frac{r}{(1-\gamma)c^+}$, leaving that region intact.

## 4. The Principal's Optimal Decision

We now consider the principal's problem $\mathbf{P}(M, r)$ as formulated in Equation (3), and examine to what extent it can be rigorously characterized, utilizing analysis similar to that established in Section 3. We begin by examining the *feasibility* of $\mathbf{P}(M, r)$, i.e., whether the problem's constraints can be simultaneously satisfied, and the necessary conditions to achieve the principal's three goals outlined in Section 2.

### 4.1. Without Leg-up Effect

**Theorem 4.1.** *When $\delta = 0$, $\mathbf{P}(M, r)$ is infeasible if $M \geq \frac{r}{(1-\beta)(1-\gamma)^2 c^+}$.*

Theorem 4.1 indicates a limit to how high a level the agent can reach without the leg-up effect. This is because without

the leg-up effect to counter the attribute depreciation, the latter dominates as a multiplicative discount. This means that the loss eventually outweighs the gain when the agent reaches a certain level, above which the mechanism is no longer able to sustain. The bound in Theorem 4.1 increases in both $\beta$ and $\gamma$, and its sensitivity to $\gamma$ is significantly higher (quadratic in $\gamma$ but linear in $\beta$). This implies that the impossibility region shrinks more rapidly in response to rises in attribute retention than the agent's farsightedness. This is consistent with that in Section 3.2, suggesting the retention rate is more effective in incentivizing improvement.

### 4.2. With Leg-up Effect

**Theorem 4.2.** *Suppose $\delta > 0$. Then,*

*(1) if $r < \frac{1-\beta}{1-\gamma}c^+\delta$, $\mathbf{P}(M, r)$ is infeasible for any $M$.*

*(2) if $r \geq \frac{1-\beta}{1-\gamma}c^+\delta$ and $c^- \geq \max\{(1 + \frac{\beta\gamma}{2})(1 - \beta\gamma)c^+,\ \beta\gamma(1 - \beta^2\gamma^2)c^+\}$, $\mathbf{P}(M, r)$ is feasible for all $M$, and a feasible sequence is given by $\mu_l = \frac{\delta l}{1-\gamma}$, $\forall l \in [L]$, with $L = \lceil(1-\gamma)M/\delta\rceil$. Furthermore, this sequence is optimal if $r = \frac{1-\beta}{1-\gamma}c^+\delta$.*

**Feasibility condition**  Theorem 4.2 says that the problem is feasible only if the reward rate is sufficiently high relative to the leg-up effect. Note that the sufficient condition in (2) implies Theorem 2.2; empirically, it is at most $\approx 2\times$ as strict (see Section E.4), so the principal retains 30–50% tolerance for parameter misspecification. The core tension lies in the spacing between thresholds: when the reward rate is low, smaller gaps are required to maintain the agent's incentive to climb the ladder (i.e., if the promotion is minor, do not make people jump through hoops); however, if the gaps become too small, the agent can rely only on the leg-up effect to navigate through the levels without improvement efforts (i.e., promotion begets promotion). The minimum reward rate required for the feasibility of $\mathbf{P}(M, r)$ is decreasing in the discount factor $\beta$, but increasing in both the retention rate $\gamma$ and the leg-up effect $\delta$. Intuitively, a less farsighted agent requires larger future rewards to sustain the incentive for improvement, thereby raising the minimum required reward rate.

**Optimal threshold sequence**  To understand the feasible threshold sequence identified in Theorem 4.2-(2), note that each $\mu_l$ in this sequence equals the steady state of the attribute if the agent were to stay in level $l$ indefinitely without further effort. At this threshold, attribute depreciation and the leg-up effect cancel out. Once the agent reaches $\mu_l$ from a previous level, the attribute transition guarantees that its next-state attribute remains the same: $\gamma\mu_l + \delta(l-1) = \gamma\frac{\delta(l-1)}{1-\gamma} + \delta(l-1) = \frac{\delta(l-1)}{1-\gamma} = \mu_l$. That is, reaching a level by improvement from a lower level effectively resets the agent's "intrinsic endowment" to $\mu_l$, as

its attribute will not fall below this value in the future.

By setting the threshold at the steady state, the principal harnesses the system's natural equilibrium rather than fighting against it. In fact, such a "natural" solution is very close to the optimal sequence as $M$ grows large. If the threshold were set such that $\mu_l > \frac{\delta(l-1)}{1-\gamma}$ (an inevitable situation if $L < \lceil(1-\gamma)M/\delta\rceil$), this level is not "stable" enough because the attribute dynamics would exert a downward pressure toward the steady state $\frac{\delta(l-1)}{1-\gamma}$, forcing the agent to keep exerting efforts to maintain their current level. This backward drift becomes particularly detrimental at large $M$, where the multiplicative depreciation dominates. Conversely, if the threshold were set lower than $\frac{\delta(l-1)}{1-\gamma}$, an agent with attribute $x_t \in ((\mu_l - \delta(l-1))/\gamma,\ \mu_l]$ at level $l_t = l - 1$ would find it optimal to just game up for a promotion, where the attribute dynamic from the next level ($l_{t+1} = l$) would then push their attribute upward above $\mu_l$ (i.e., $x_{t+1} \geq \gamma x_t + \delta(l-1) \geq \mu_l$).

While Theorem 4.2-(2) offers a powerful result, it requires a higher threshold for the minimum gaming cost, $c^-$. This reflects a critical trade-off: If gaming is too inexpensive, the agent will tend to continuously rely on cheating to gain promotion. Although cheating does not improve attributes directly, the "leg-up" effect provided by higher levels facilitates rapid attribute cumulation, even in the absence of genuine improvement efforts. This type of gaming behavior is more thoroughly discussed in Section 5.2.

## 5. Numerical Results

As shown in Section 4, the principal's problem $\mathbf{P}(M, r)$ is not always feasible. In this section we propose and analyze a greedy heuristic algorithm and numerically solve a relaxed principal's problem. In all experiments, we employ a subroutine to compute the agent's optimal response to the principal's decisions. To this end, we develop VALUEITERATE, a value iteration algorithm that utilizes attribute-space discretization and linear interpolation of the Bellman equation. We show that the algorithm exhibits linear convergence with rate $O\left(\frac{\log(1/\varepsilon)}{|\log\beta|}\right)$ and approximates the true value function with an error bound of $\frac{c^+\Delta x}{2(1-\beta)}$. The details and theoretical proofs are provided in Section E.1.

### 5.1. The Greedy Heuristic

We propose a greedy algorithm that iteratively builds the threshold sequence via searching for the largest next threshold such that the first and third conditions of $\mathbf{P}(M, r)$ are satisfied under the current number of thresholds. More details of the algorithm are provided in Section E.2.

**Theorem 5.1.** *Let Algorithm 1 return $\vec{\mu}$ of length $L$. Then, $\vec{\mu}$ is a feasible solution to $\mathbf{P}(M, r)$ for $M \leq \mu_L$.*

**Algorithm 1** Greedy Heuristic for Optimal Thresholds
___

**Require:** target $M$, params $\Theta = \{\beta, \gamma, c^+, c^-, r, \delta\}$, discretization step $\Delta x$, max attribute $X_{\max}$, precision $\varepsilon$
**Require:** VALUEITERATE that approximates the agent's optimal strategy on the discretized attribute space $\mathcal{X}$
1: $\mathcal{X} \leftarrow$ discretized $[0, X_{\max}]$ with gap $\Delta x$
2: $\mu_1 \leftarrow 0, x_0 \leftarrow 0$ {post-response attribute}, $l \leftarrow 2$
3: **while** $x_{l-1} < M$ **do**
4:   $a \leftarrow \max\{\mu_{l-1}, \frac{\delta(l-1)}{1-\gamma}\}, b \leftarrow X_{\max}$
5:   $m \leftarrow \mu_{l-1}$ {candidate for $\mu_l$}
6:   **while** $(b - a) > \varepsilon$ **do**
7:     $m \leftarrow (a + b)/2$
8:     $(a^+, a^-) \leftarrow$ VALUEITERATE$(\vec{\mu} \cup \{m\}, \mathcal{X}|\Theta)$
9:     $x_l \leftarrow x_{l-1} + a^+(l - 1, \gamma x_{l-1} + \delta(l - 2))$
10:     $z = x_l + a^-(l - 1, \gamma x_{l-1} + \delta(l - 2))$
11:     **if** $x_l = z, z \geq m$, and $a^+(l, \gamma m + \delta l) > 0$ **then**
12:       $a \leftarrow m$
13:     **else**
14:       $b \leftarrow m$
15:     **end if**
16:   **end while**
17:   **if** $(a - \mu_{l-1}) < \varepsilon$ **then**
18:     **break** {not found within numerical tolerance}
19:   **else**
20:     $\mu_l \leftarrow a, l \leftarrow l + 1$
21:   **end if**
22: **end while**
23: Return $\vec{\mu}$
___

While the analytical bounds in Theorems 4.1 and 4.2(2) guarantee feasibility, Algorithm 1 provides an efficient numerical tool, consistently recovering a feasible sequence, especially when $\delta = 0$ and Theorem 4.2 does not apply. In practice, the search algorithm consistently recovers a feasible sequence whenever the problem parameters satisfy the sufficiency conditions derived in Section 4.

Theorem 5.1 enables a complementary analysis of the theoretical results established in Section 4. In Sections E.2.2 and E.2.3, we use this greedy heuristic to assess how problem parameters affect the highest threshold of a feasible sequence. Our results indicate that higher retention rates ($\gamma$) and agent farsightedness ($\beta$) can expand the range of incentivizable attributes (i.e., the range of $M$), allowing the principal to induce a significantly higher attribute from the agent (see Figures 9 and 10). Furthermore, Section E.2.4 empirically validates the bound in Theorem 2.1, demonstrating a sharp phase transition where incentivizability vanishes if the gaming cost falls below the critical threshold $(1 - \beta\gamma)c^+$ (see Figure 11).

## 5.2. A Relaxed Objective Function and Experiment

To evaluate the practical implications of our model, we simulate a population of agents using the FICO credit score dataset (Hardt et al., 2016). We model exclusively the initial attribute $x_0$ using the FICO raw score distribution normalized to $[0, 10]$. We simulate a multi-level credit building system where agents seek to move from entry-level credit products to high-reward loans (e.g., mortgages). By designing a progression of thresholds, our model demonstrates how the principal can incentivize customers from subprime to prime status through genuine financial/credit posture improvement rather than temporary "gaming" of credit score formulas.

For the principal's utility, we introduce the following utility function by relaxing the hard constraint in $\mathbf{P}(M, r)$:

$$U(r, L, \vec{\mu}) := \tag{4}$$
$$\mathbb{E}_{x_0} \sum_{t=0}^{\infty} \alpha^t \big( \mathbf{1}\{f_{l_t}(z_t^*) = f_{l_t}(x_{t_+}^*)\} + \lambda x_{t+1}^* - \xi r l_{t+1} \big)$$

where $\alpha$ is the principal's discount factor and $\lambda, \xi > 0$ are constants. This utility represents a weighted trade-off among the three most desired properties for the principal: classifier robustness against strategic manipulation, agent qualification (as reflected by the limiting attribute), and the cost of implementation. The reward rate, $r$, is explicitly included as a design variable and added as the cost of implementation.

Similar to the derivation presented in Section 3, the features ($z_t^*$) and attributes ($x_{t_+}^*$ and $x_{t+1}^*$) are solutions to the agent's optimal strategy in response to a given design $(r, L, \vec{\mu})$. The solution similarly constitutes a subgame perfect equilibrium and is inherently non-analytical and non-convex. Thus, we employ the Covariance Matrix Adaptation Evolution Strategy (CMA-ES) (Hansen et al., 2025) for black-box optimization of the principal's problem (see Section E.3.1 for details).

**Optimal Sequence Design** In this experiment, we determine the principal's optimal classifier sequence across four cost cases $(c^+, c^-)$, shown in Table 5.1. We fix the remaining parameters at $\beta = 0.8, \gamma = 0.8, \delta = 0.01, \alpha = 0.95, \xi = 0.01$, and $\lambda = 5$. We test levels between $L = 2$ and $L = 8$ and choose the one yielding the highest principal's utility. Case I represents the ideal case where $c^+$ is low, and $c^-$ is high. This could be for a task like learning a language, where genuine effort is accessible, but faking fluency is difficult. In this case, the principal successfully incentivizes with 6 levels and moderate reward $r^* = 1.80$, achieving a high long-term agent attribute. Case II is when both $c^+$ and $c^-$ are high. This is analogous to a task where both qualifying and faking are difficult. To compensate for the high cost of effort, the principal must provide significant rewards $r^* = 2.51$, to sustain long-term motivation. Case

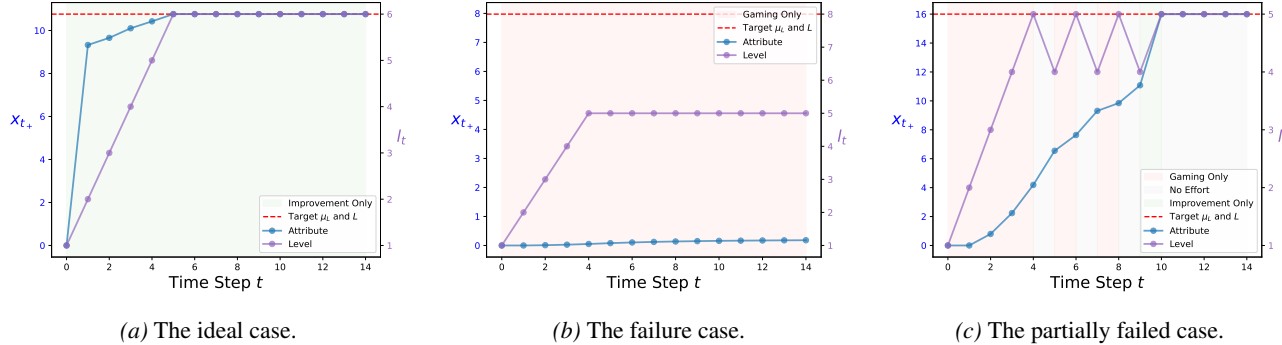

*(a)* The ideal case.      *(b)* The failure case.      *(c)* The partially failed case.

*Figure 4.* Agent state trajectories and strategy types. The dashed red lines mark the largest level ($L$) and threshold ($\mu_L$). (a) Successful incentive alignment leads to monotonic honest improvement. (b) Total failure, where high costs and low rewards prevent genuine improvement efforts despite advancement in levels. (c) Partial failure where the agent achieves the target, yet mainly through gaming.

III is when $c^+$ is low, but $c^-$ is much lower, typical of simple tasks that can be easily manipulated. In this case, the principal collapses the levels into two and sets a very high threshold to discourage gaming over repeated time steps. Finally, Case IV represents a worst-case scenario where learning is hard, and cheating is cheap. In this case, the condition in Theorem 2.2 is not met, and the mechanism becomes highly inefficient, with the agent best responding by cheating, yielding the lowest utility (107.9), and requiring a long 8-level structure.

*Table 5.1.* Optimal design under different cost conditions.

| **Case** | $(c^+, c^-)$ | $L^*$ | $r^*$ | $\mu_L^*$ | $U^*$ |
|---|---|---|---|---|---|
| I | (0.8, 0.7) | 6 | 1.80 | 10.76 | 630.4 |
| II | (1.5, 1.2) | 7 | 2.51 | 11.92 | 629.9 |
| III | (0.8, 0.4) | 2 | 4.48 | 11.98 | 628.8 |
| IV | (1.5, 0.4) | 8 | 0.63 | 7.98 | 107.9 |

**Agent's Strategic Behavior under the Optimal Design**
We now examine the agent's state trajectory in the practical scenario where the principal strategically maximizes Equation (4). Figure 4 summarizes the trajectories for an agent with initial state $(l_0, x_0) = (1, 0)$ across different parameter settings (details see Section E.3.2).

Figure 4a illustrates the ideal scenario (Case I in Table 5.1), where the agent consistently exerts genuine effort, advancing both its attribute and level at every time step. Here, a non-equally spaced threshold sequence is used, corresponding to the optimization results for Case I in Table 5.1. Notably, trajectories in Cases II and III exhibit patterns similar to Figure 4a (see Section E.3.2), suggesting that the principal maximizes utility by inducing consistent improvement up to the highest level and attribute. These outcomes satisfies the hard constraints of $\mathbf{P}(M, r)$ in Equation (2).

In contrast, Figure 4b illustrates Case IV in the low gaming cost setting. Here, the agent fails to reach the target $L = 8$, instead plateauing at $l = 5$. Because the principal

strategically sets the final threshold significantly higher (i.e., $\mu_L \gg \frac{\delta(L-1)}{1-\gamma}$), the attribute stabilizes at $\frac{\delta(L-1)}{1-\gamma}$, far below $\mu_L$. Consequently, the plateau in this case persists indefinitely without any improvement effort. Recall that Case IV of Table 5.1 corresponds to the impossibility regime where $c^- < (1 - \beta\gamma)c^+$. This implies that when the system cannot sustain improvement at all (e.g., due to extremely low gaming costs), a strategic principal accepts an "inferior" equilibrium where improvement incentives are absent.

Finally, Figure 4c presents an instructive yet suboptimal (to the principal's relaxed utility) example where the agent reaches the target state but relies heavily on gaming. Here, the low gaming cost condition in Theorem 4.2-(2) is violated, but Theorem 2.2 holds. Using the "natural" threshold sequence $\mu_l = \frac{\delta(l-1)}{1-\gamma}$ for $l \in [5]$ in Theorem 4.2-(2), the agent exclusively utilizes low-cost gaming to ascend rapidly, reaching the ceiling ($L = 5$) at $t = 4$. During this ascent, the agent's attribute accumulates at an accelerated rate due to the stronger leg-up effect at higher levels. However, this proves insufficient, and thus during $t \in [4, 9]$, the agent enters an oscillatory pattern: dropping to $L - 1$ (via inaction) and immediately gaming back to $L$. This cycle persists until $t = 9$, when the attribute has accumulated sufficiently via the leg-up effect to make genuine improvement viable. The agent then exerts a single improvement effort, stabilizing indefinitely at the maximum state $(L, \mu_L)$.

**Ablation on the Agent Behavior** We investigate how system parameters influence the agent's trajectory and strategic choice between improvement and gaming (see Section E.3.3). In short, while both retention ($\gamma$) and farsightedness ($\beta$) raise steady-state attributes, retention proves to be the dominant factor (Figure 13). Notably, the leg-up parameter ($\delta$) has a two-sided impact: While a moderate leg-up encourages improvement, a strong leg-up ($\delta = 0.5$) reduces the fraction of honest effort, as the natural boost provided by the level itself becomes sufficient to sustain the agent's position without active investment (Figure 14).

*Table 5.2.* Univariate misspecification: utility loss (%) relative to oracle ($U^* = 630.4$) and terminal gaming fraction (in parentheses).

| Parameter | $\varepsilon = -30\%$ | $\varepsilon = -10\%$ | $\varepsilon = 0\%$ | $\varepsilon = +10\%$ | $\varepsilon = +30\%$ |
|---|---|---|---|---|---|
| $\delta$ (leg-up) | 0.0% (0%) | 0.0% (0%) | 0.0% (0%) | 0.0% (0%) | 0.0% (0%) |
| $\gamma$ (retention) | 0.2% (0%) | 0.4% (0%) | 0.0% (0%) | 82.4% (100%) | 81.3% (100%) |
| $c^-$ (gaming cost) | 0.1% (0%) | 0.0% (0%) | 0.0% (0%) | 2.6% (0%) | 12.0% (0%) |
| $c^+$ (impr. cost) | 15.2% (99%) | 1.3% (0%) | 0.0% (0%) | 0.0% (0%) | 0.0% (0%) |
| $\beta$ (farsight.) | 0.1% (0%) | 0.0% (0%) | 0.0% (0%) | 32.8% (28%) | 50.9% (100%) |

**Robustness to parameter misspecification** A practical concern is the principal's potential uncertainty regarding the agent's true parameters $(\beta, \gamma, c^+, c^-, \delta)$. We examine the mechanism's robustness when the principal's belief about each parameter $\kappa \in (\beta, \gamma, c^+, c^-, \delta)$ deviates by a relative error of $\varepsilon \in (-1, 1)$, i.e., $\hat{\kappa} = \kappa \cdot (1+\varepsilon)$. The agent responds according to the true parameters (Case I: $\beta = 0.8$, $\gamma = 0.8$, $c^+ = 0.8$, $c^- = 0.7$, $\delta = 0.01$). For each (parameter, $\varepsilon$) pair, we (i) use CMA-ES under $\hat{\kappa}$ over $L \in \{2, \ldots, 8\}$ to obtain $(r^*, \boldsymbol{\mu}^*)$, and (ii) simulate the agent population under the true parameters to measure the realized utility loss and terminal gaming fraction. The results are reported in Table 5.2.

The oracle at $\varepsilon = 0$ achieves $U = 630.4$ with $L = 6$ levels and 0% gaming. First, $\delta$ misspecification is irrelevant. The mechanism is immune across the full $\pm 30\%$ range. Second, $c^-$ exhibits moderate one-sided sensitivity: overestimating gaming cost by up to 30% incurs at most 12% utility loss with 0% gaming, while underestimation is harmless. Third, three directions are dangerous: (a) overestimating $\gamma$ by as little as 10% causes 82% utility loss and 100% gaming; (b) underestimating $c^+$ by 30% causes the principal to set thresholds unreachable by honest effort at the true cost, collapsing to 99% gaming; (c) overestimating $\beta$ by as little as 10% induces 32.8%–50.9% utility loss and near-full gaming at 30%. All three failures share the same root cause, which is that the principal designs an overly ambitious mechanism calibrated to an (imaginary) agent who, compared to the real agent, is better at retention, cheaper to improve, or more patient; such a mechanism causes the real agent to revert to gaming. These results suggest a conservative bias: underestimating $\gamma$, overestimating $c^+$, and underestimating $\beta$ is robust, while the opposite directions cause harm.

## 6. Discussion

**More General Attribute Dynamics** Consider a general model $x_{t+1} = g(x_{t_+}, l_{t+1})$, where $g$ captures the combination of depreciation and the leg-up effect. Let $H_l$ denote the Lipschitz constant of $g(\cdot, l)$. This constant represents the *sensitivity* of the attribute dynamics: $H_l < 1$ implies contractive dynamics dominated by depreciation, while $H_l > 1$ implies dominance of the leg-up effect. Under this setup,

Theorem 2.1 generalizes as follows:

**Proposition 6.1.** *The agent never chooses an improvement action at level $l$ if $(1 - \beta H_l)c^+ > c^-$.*

Theorem 6.1 reveals that the "impossibility region" depends fundamentally on the sensitivity of the attribute within its temporal transition rule $g$, which depends on both depreciation and the leg-up effect—the latter influences $H_l$ through the level index $l$. This is in contrast to Theorem 2.1 where the leg-up factor does not appear in the condition. Theorem 6.1 suggests that higher sensitivity implies higher chances of a system being improvement incentivizable.

This sensitivity derives from both depreciation and the leg-up effect, the latter of which influences $H_l$ only through the level $l$ and is absent in Theorem 2.1.

Consider, for instance, a transition rule where the leg-up effect is multiplicative, and depreciation is additive, i.e., $g(x, l) = \max\{0, (1 + \sqrt{\delta l})x - \gamma\}$. The resulting Lipschitz constant is $H_l = 1 + \sqrt{\delta l}$, which increases in the level $l$ with diminishing returns and is independent of the depreciation. Thus, as the leg-up effect becomes stronger (i.e., larger $\delta$), the system is more likely to be improvement incentivizable.

Specializing to a level-dependent leg-up $\delta_l$, one can derive infeasibility and feasibility conditions analogous to Theorem 4.2 (with a closed-form optimal threshold), as well as a concave/saturating $\delta_l$ analysis (Section D.3).

**Scalar vs. Vector Settings** We have so far focused on the one-dimensional (scalar) setting. However, many of the insights extend naturally to the multi-dimensional (vector) case, in which the attribute becomes a vector $\mathbf{x}_t \in \mathbb{R}^d$, and the transition function $g$ operates component-wise or jointly across dimensions. The sensitivity of the system can be characterized by the largest singular value of the Jacobian of $g$ with respect to $\mathbf{x}_t$. Thus, key concepts such as the leg-up effect and the impossibility region can be similarly analyzed. In particular, the impossibility condition can be generalized by replacing the scalar Lipschitz constant $H_l$ with its vector analogue, capturing the maximal amplification across all directions. This suggests that systems with higher inter-dimensional sensitivity are more likely to be improvement incentivizable, paralleling the scalar case.

## Impact Statement

Our findings reveal how human or societal factors influence strategic classification outcomes, aiding the design of robust algorithms that account for long-term strategic adaptation. Similar to most strategic classification systems, our model carries the risks of inadvertently disadvantaging groups with high effort costs. However, our definition of "feasibility" in Equation (2) enforces equal long-term outcomes across groups with disparate endowment (i.e., initial attribute $x_0$).

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

## A. Related Works

**Strategic classification.** Strategic classification aims at enhancing the robustness of machine learning models under data providers' strategic manipulation (Hardt et al., 2015; Kleinberg & Raghavan, 2019; Milli et al., 2019; Braverman & Garg, 2020; Chen et al., 2020; Haghtalab et al., 2020; Miller et al., 2020; Bechavod et al., 2021; Jin et al., 2022). Hardt et al. (2015) pioneered the research of strategic classification and formulated the problem as a Stackelberg game, where the decision maker leads by publishing a classifier and the agents best respond by strategically manipulating their features. Derivative works have included optimal classifier design in anticipation of the agent's strategic response (Braverman & Garg, 2020; Miller et al., 2020; Zrnic et al., 2022), discussions on social welfare (Milli et al., 2019; Haghtalab et al., 2020), mechanism design that limits manipulations (Kleinberg & Raghavan, 2019; Jin et al., 2022), and learning algorithms under imperfect information (Dong et al., 2017; Chen et al., 2020; Bechavod et al., 2021).

**Improvement incentivization.** While a large body of the literature has focused on malicious manipulation/gaming actions (Hardt et al., 2015; Braverman & Garg, 2020), a comparably large amount of works also consider the existence of both improvement (causal to both true attribute and observable feature) and gaming (causal only to the observable feature) actions (Miller et al., 2020; Bechavod et al., 2021; Ahmadi et al., 2022; Jin et al., 2022). Jin et al. (2022) points out that improvement can only be incentivized when there exists an improvement direction that is more cost efficient than all gaming directions and Bechavod et al. (2021) shows the possibility of learning such direction when it exists. Improvement incentivization becomes much more challenging in a s practical situation when such direction does not exist. Jin et al. (2022; 2023) propose to leverage external instruments, such as transfer mechanisms, collaborations, or recourse policies, to address this problem. In contrast, the present paper shows the possibility of incentivizing improvement in the long run, through classifier design only.

**Long-term strategic classification.** Long-term impacts of machine learning systems have been extensively studied in the literature. Besides Harris et al. (2021) discussed in Section 1, a very different type of sequentiality is considered in Cohen et al. (2023), where an agent can bypass a set of classifiers (half planes) sequentially without entering the intersecting region. This study does not invoke inter-temporal incentives in the conventional sense, and more importantly, it does not distinguish between honest and dishonest efforts. Zhang et al. (2019; 2020) examine the distribution dynamics of an agent under indefinite and repeated interactions with fair classifiers, where agents' attribute distribution is updated each time as a result of responding to the classification outcome. Performative prediction (Perdomo et al., 2021; Jin et al., 2024a;b) further abstracts the agent's response as the sensitivity of the population distribution and studies the model and population dynamics in the long run. However, long-term properties with explicit best-response actions are less commonly studied, with the exception of Xie & Zhang (2024) that examines the evolving dynamics between machine learning models and the population where human best responses are taken as annotated samples in the next round of interaction and Zrnic et al. (2022) that characterizes the equilibrium outcome of the repeated Stackelberg game between the decision maker and the strategic agent. All works discussed above assume a singular effect of the agent's actions (i.e., either causal/improvement or non-causal/gaming to the agent's attribute), while our work models both improvement and gaming actions explicitly in the study of long-term properties. In a companion work, Huang et al. (2026) extend the present model to a stochastic setting where classification outcomes are random.

**Self-selection in a sequence of classifiers.** This stream of literature also involves the notion of sequencing and level advancement, but focuses primarily on the voluntary participation or "self-selection" problem: whether the agent is incentivized to complete the sequence of tests rather than opting out in the middle of the process. In particular, Zhang et al. (2021) study the optimal design of test sequences and acceptance rules to incentivize agents to participate. In their model, the agent's actions are binary choices—taking the next test or dropping out—rather than the exertion of effort to change their attributes. Horowitz et al. (2024) focus on the learning perspective of this self-selection problem, while the agent's actions remain binary.

## B. Preliminary: Formulation of the Bellman Equation

Let $V_l(x) = V(l, x)$ denote the agent's minimum *negative utility* (i.e., the negative of the objective in Equation (2)) if it starts at level $l$ with attribute $x$. By the Markov property, $V(l, x)$ must satisfy the following Bellman equation

$$V(l, x) = \min_{\vec{a} \geq \vec{0}} \ \vec{c}^\top \vec{a} - r\big(l + f_l(x + a^+ + a^-) - 1\big) + \beta V\Big(l + f_l(x + a^+ + a^-), \ \gamma(x + a^+) + \delta\big(l + f_l(x + a^+ + a^-) - 1\big)\Big)$$

where $\vec{a} := [a^+, a^-]^\top$ is the action vector. We introduce the following reparameterization: $\tilde{x} := x + a^+$ and $u := a^-$, and thus rewrite the Bellman equation as

$$V(l, x) = -c^+ x + \min_{\tilde{x} \geq x, \, u \geq 0} c^+ \tilde{x} + c^- u - r\big(l + f_l(\tilde{x} + u) - 1\big) + \beta V\Big(l + f_l(\tilde{x} + u), \, \gamma \tilde{x} + \delta\big(l + f_l(\tilde{x} + u) - 1\big)\Big). \quad (5)$$

Define $W_l(x) := V(l, x) + c^+ x$. The Bellman equation is equivalent to the following fixed-point problem in $W$.

$$W_l(x) = \min_{\tilde{x} \geq x, \, u \geq 0} (1 - \beta\gamma)c^+ \tilde{x} + c^- u - (r + \beta c^+ \delta)\big(l + f_l(\tilde{x} + u) - 1\big) + \beta W_{l+f_l(\tilde{x}+u)}\Big(\gamma \tilde{x} + \delta\big(l + f_l(\tilde{x} + u) - 1\big)\Big). \quad (6)$$

It is easy to verify that $\{W_l\}_{l \in [L]}$ are **continuous** and **non-decreasing**.

Inserting the expression of $f_l$ and expand Equation (6), we obtain the detailed expression for $W_l$ as follows:

$$W_l(x) = \begin{cases} \min\limits_{\substack{\tilde{x} \geq x, u \geq 0 \\ \tilde{x} + u < \mu_l}} (1 - \beta\gamma)c^+ \tilde{x} + c^- u - (r + \beta c^+ \delta)(l - 2) + \beta W_{l-1}(\gamma x + \delta(l - 2)) \\ \min\limits_{\substack{\tilde{x} \geq x, u \geq 0 \\ \mu_l \leq \tilde{x} + u < \mu_{l+1}}} (1 - \beta\gamma)c^+ \tilde{x} + c^- u - (r + \beta c^+ \delta)(l - 1) + \beta W_l(\gamma x + \delta(l - 1)) \\ \min\limits_{\substack{\tilde{x} \geq x, u \geq 0 \\ \tilde{x} + u \geq \mu_{l+1}}} (1 - \beta\gamma)c^+ \tilde{x} + c^- u - (r + \beta c^+ \delta)l + \beta W_{l+1}(\gamma x + \delta l) \end{cases} \quad (7)$$

where the first and third cases do not exist for $l = 1$ and $l = L$, respectively.

We first minimize each case in Equation (7) w.r.t. the gaming action $u \geq 0$ and obtain $W_l(x) = \min_{\tilde{x} \geq x} \Phi_l(\tilde{x} \mid W)$ where

$\Phi_l(\tilde{x} \mid W) = $ for each case of $\tilde{x}$, the minimum of the corresponding block

$$\begin{cases} \Phi_l^{(1)}(\tilde{x}) := & (1 - \beta\gamma)c^+ \tilde{x} - (r + \beta c^+ \delta)(l - 2) + \beta W_{l-1}(\gamma \tilde{x} + \delta(l - 2)) \\ \\ \Phi_l^{(2)}(\tilde{x}) := & [(1 - \beta\gamma)c^+ - c^-]\tilde{x} + c^- \mu_l - (r + \beta c^+ \delta)(l - 1) + \beta W_l(\gamma \tilde{x} + \delta(l - 1)) \quad \text{if } \tilde{x} < \mu_l \\ \\ \Phi_l^{(3)}(\tilde{x}) := & [(1 - \beta\gamma)c^+ - c^-]\tilde{x} + c^- \mu_{l+1} - (r + \beta c^+ \delta)l + \beta W_{l+1}(\gamma \tilde{x} + \delta l) \\ \text{- - - - - - - - - - - - - - - - - - - - - - - - - - - - - - - - - - - - - - - - - - - - -} \\ \Phi_l^{(4)}(\tilde{x}) := & (1 - \beta\gamma)c^+ \tilde{x} - (r + \beta c^+ \delta)(l - 1) + \beta W_l(\gamma \tilde{x} + \delta(l - 1)) \\ & \hspace{9cm} \text{if } \mu_l \leq \tilde{x} < \mu_{l+1} \\ \Phi_l^{(5)}(\tilde{x}) := & [(1 - \beta\gamma)c^+ - c^-]\tilde{x} + c^- \mu_{l+1} - (r + \beta c^+ \delta)l + \beta W_{l+1}(\gamma \tilde{x} + \delta l) \\ \text{- - - - - - - - - - - - - - - - - - - - - - - - - - - - - - - - - - - - - - - - - - - - -} \\ \Phi_l^{(6)}(\tilde{x}) := & (1 - \beta\gamma)c^+ \tilde{x} - (r + \beta c^+ \delta)l + \beta W_{l+1}(\gamma \tilde{x} + \delta l) \quad \text{if } \tilde{x} \geq \mu_{l+1} \end{cases} \quad (8)$$

where $\Phi_1$ does not have $\Phi_1^{(1)}$, $\Phi_1^{(2)}$, and $\Phi_1^{(3)}$, and $\Phi_L$ does not have $\Phi_l^{(3)}$, $\Phi_l^{(5)}$, and $\Phi_l^{(6)}$. Each of the functions $\Phi_l^{(1)}$ to $\Phi_l^{(6)}$ represents a behavior pattern of the agent. For example, for $x < \mu_l$, $\Phi_l^{(2)}(x)$ represents the agent's value if it improves (insufficiently) to a point $\tilde{x}$ less than $\mu_l$, and uses gaming actions to bridge the remaining gap to the threshold $\mu_l$. In some cases, we omit the dependence of $\Phi$ on $W$ without ambiguity.

For $x < \mu_l$, we sometimes exchange the two minimum operators in $W_l(x)$ ($\Phi_l^{(3)}$ and $\Phi_l^{(5)}$ are combined as they are the same expression):

$$W_l(x) = \min\left\{ \min_{x \leq \tilde{x} < \mu_l} \Phi_l^{(1)}(\tilde{x}), \min_{x \leq \tilde{x} < \mu_l} \Phi_l^{(2)}(\tilde{x}), \min_{x \leq \tilde{x} < \mu_{l+1}} \Phi_l^{(3)}(\tilde{x}), \min_{\mu_l \leq \tilde{x} < \mu_{l+1}} \Phi_l^{(4)}(\tilde{x}), \min_{\tilde{x} \geq \mu_{l+1}} \Phi_l^{(6)}(\tilde{x}) \right\}. \quad (9)$$

The optimal improvement action is determined by

$$(a^+)^*(l, x) = \sup\left( \operatorname*{arg\,min}_{\tilde{x} \geq x} W_l(\tilde{x}) - x \right) = \sup\left( \operatorname*{arg\,min}_{\tilde{x} \geq x} \Phi_l(\tilde{x} \mid W) - x \right). \quad (10)$$

The optimal gaming action depends on which one of $\Phi_l^{(1)}$ to $\Phi_l^{(6)}$ equals $W_l$. For instance, if $W_l(x) = \Phi_l^{(5)}(\mu_{l+1} - \varepsilon)$ for some $\varepsilon > 0$, the optimal improvement action is $\mu_{l+1} - \varepsilon - x$ and the gaming action is $\varepsilon$. In this scenario, the agent always reaches the promotion threshold $\mu_{l+1}$, partitioning the total effort between $\mu_{l+1} - \varepsilon - x$ in improvement and $\varepsilon$ in gaming.

### B.1. Proof of Theorem 2.1

When $(1 - \beta\gamma)c^+ > c^-$, it directly follows that all functions from $\Phi_l^{(1)}$ to $\Phi_l^{(6)}$ in Equation (8) are strictly increasing. This implies $\arg\min_{\tilde{x} \geq x} W_l(\tilde{x}) = x$ for any $x$, and, according to Equation (10), $(a^+)^*(l, x) \equiv 0$ for any $l$ and $x$.

## C. The Agent's Optimal Strategy in the Two-Level Case

### C.1. Notation Simplification

For $L = 2$, we have $\Phi_1(x \mid W) = \Phi_2(x \mid W)$ for all $x \geq 0$, implying $W_1 = W_2$, $a^+(1, \cdot) = a^+(2, \cdot)$, and $a^-(1, \cdot) = a^-(2, \cdot)$. For brevity, we drop the dependence on the level in this section and adopt the notations $W(x) := W_1(x)$, $a^+(x) := a^+(1, x)$, and $a^-(x) := a^-(1, x)$. From Section B, we obtain the fixed-point relation regarding $W$ as follows

$$W(x) = \min_{\tilde{x} \geq x, \, u \geq 0} (1 - \beta\gamma)c^+\tilde{x} + c^-u - (r + \beta c^+\delta)\mathbf{1}\{\tilde{x} + u \geq \mu\} + \beta W(\gamma\tilde{x} + \delta\mathbf{1}\{\tilde{x} + u \geq \mu\}), \tag{11}$$

and the $\Phi$ function can be written as

$$\Phi(\tilde{x}) = (1 - \beta\gamma)c^+\tilde{x} + \begin{cases} -(r + \beta c^+\delta) + \beta W(\gamma\tilde{x} + \delta) & \tilde{x} \geq \mu \\ c^-(\mu - \tilde{x}) - (r + \beta c^+\delta) + \beta W(\gamma\tilde{x} + \delta) & \tilde{x} < \mu \, \& \, (\star) \\ \beta W(\gamma\tilde{x}) & \tilde{x} < \mu \, \& \, \neg(\star), \end{cases} \tag{12}$$

where $(\star)$ stands for the condition

$$c^-(\mu - \tilde{x}) + \beta\left[W(\gamma\tilde{x} + \delta) - W(\gamma\tilde{x})\right] \leq r + \beta c^+\delta. \tag{13}$$

Notice that Equation (12) also implies that

$$u^* = (a^-)^* = (\mu - \tilde{x})\mathbf{1}\{\tilde{x} < \mu \, \& \, (\star)\}, \tag{14}$$

meaning only in the second case, the agent would select a non-zero gaming action whose magnitude equals $\mu - \tilde{x}$.

### C.2. Overview of the Proofs of Theorems 3.1 and 3.2

Both proofs of Theorems 3.1 and 3.2 involve two steps: first identifying the closed-form expression for $W$, and then finding the optimal action from the expression of $W$.

Define

$$G(x) := \frac{1 - \beta\gamma}{1 - \beta}c^+x + \frac{c^-(\mu - x) - (r + \beta c^+\delta)}{1 - \beta}\mathbf{1}\{(\star)\}, \quad x \in [0, \mu]. \tag{15}$$

Then, it is easy to see that $\Phi(x) = (1 - \beta)G(x) + \beta\widetilde{W}(x)$, where $\widetilde{W}(x) := W(\gamma x + \delta)\mathbf{1}\{(\star)\} + W(\gamma x)\mathbf{1}\{\neg(\star)\}$. The function $\widetilde{W}$ can be viewed as a transformation of $W$ whose domain is **first scaled by $\gamma$ and then partially shifted by $\delta/\gamma$** — the shifted part corresponds to the region $x \in \{(\star)\}$. **If the running minimum $x \mapsto \min_{\tilde{x} \geq x} \Phi(x)$ coincides with the curve of $W$, then $W$ must be the solution to the fixed point equation.** Figure 5 illustrates the steps to verify whether a function $W$ satisfies the fixed-point equation.

The "scaled-and-shift" perspective divides our analysis into two disjoint cases: $\mu < \frac{\delta}{1-\gamma}$ and $\mu \geq \frac{\delta}{1-\gamma}$. In the former case, corresponding to Theorem 3.1, there exists a small $\varepsilon > 0$ such that $\gamma x + \delta > \mu$, $\forall x \in (\mu - \varepsilon, \mu)$. This means $\widetilde{W}$ on $(\mu - \varepsilon, \mu)$ equals to values of $W$ outside of $[0, \mu]$. The other case $\mu \geq \frac{\delta}{1-\gamma}$, corresponding to Theorem 3.2, however, does not have this issue.

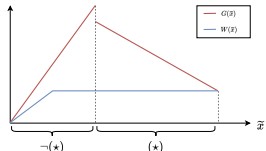 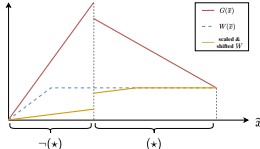 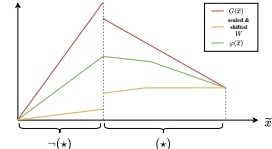 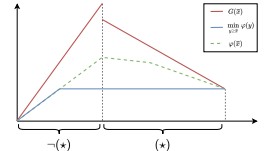

*(a)* $G(x)$ and $W(x)$ over $[0, \mu]$; the regions partitioned by the condition $(\star)$ are marked as disjoint intervals.

*(b)* Scale $W(x)$ by $\gamma$ and the curve corresponding to $(\star)$ is shifted to the left by $\delta/\gamma$, yielding the yellow curve.

*(c)* Compute the weighted average between $G$ and scaled & shifted $W$, yielding the green curve.

*(d)* Compute the running minimum of the green curve $(\min_{y \geq x} \Phi(y))$, yielding a curve that coincides with $W(x)$.

*Figure 5.* Pictorial interpretation of $W$ that satisfies the fixed-point equation.

### C.3. Proof of Theorem 3.1

#### C.3.1. IDENTIFYING THE FIXED-POINT SOLUTION

**Claim C.1.** *Define the sequence* $\{s_k, \xi_k\}_{k=0}^{K-1}$ *where* $K := \left\lceil \log_{\beta\gamma}(1 - (1 - \beta\gamma)c^+/c^-\right\rceil$ *by* $s_0 = \mu$, $\xi_0 = c^+\mu - \frac{r}{1-\beta}$, $s_k = \frac{s_{k-1} - \delta}{\gamma}$, *and* $\xi_k = \left(c^+ - \frac{1 - \beta^k\gamma^k}{1 - \beta\gamma}c^-\right)(s_k - s_{k-1}) + \xi_{k-1}, \forall k = 1, \ldots, K-1$. *Then, we have*

$$W(x) = \xi_{K-1}\mathbf{1}\{x < s_{K-1}\} + \sum_{k=1}^{K-1}\left[\left(c^+ - \frac{1 - \beta^k\gamma^k}{1 - \beta\gamma}c^-\right)(x - s_k) + \xi_k\right]\mathbf{1}\{s_k \leq x < s_{k-1}\}$$

$$+ \left[c^+(x - s_0) + \xi_0\right]\mathbf{1}\{x \geq s_0\}. \quad (16)$$

*Proof.* Given $\mu < \frac{\delta}{1-\gamma}$, it is easy to see that $\{s_k\}_k$ is a decreasing sequence.

Let $h(x) := c^-(\mu - x) + \beta\left[W(\gamma x + \delta) - W(\gamma x)\right]$ be the LHS of Equation (13). With a slight abuse of notation, we use $h'$ and $W'$ to denote the *right* derivative of the respective functions. Then, we have $h'(x) = -c^- + \beta\gamma\left[W'(\gamma x + \delta) - W(\gamma x)\right]$. Using Equation (16), we obtain

$$\sup_{x \in [0, \mu]} h'(x) \leq -c^- + \beta\gamma c^+ < 0$$

where the inequality results from the condition $c^- > \max\{\beta\gamma c^+, (1 - \beta\gamma)c^+\}$. Thus, the LHS of Equation (13) is strictly decreasing, which implies the existence of $\omega \in \mathbb{R}$ such that $\{x \leq \mu \ \& \ \neg(\star)\} = [0, \omega)$ and $\{x \leq \mu \ \& \ (\star)\} = [\omega, \mu]$.

Next, we establish that the region where $W$ is strictly increasing (or "non-flat") is contained in the second case of Equation (12), specifically, $[s_{K-1}, \mu] \cap \mathbb{R}_{\geq 0} \subseteq [\omega, \mu]$. Let $\tilde{K}$ be the **first index** in $[K-1]$ such that $s_{\tilde{K}} \leq 0$. If $s_k > 0$ for all $k \in [K-1]$, we set $\tilde{K} = K - 1$.

$$h(\max\{0, s_{\tilde{K}}\}) \leq c^-(\mu - s_{\tilde{K}}) + \beta(\xi_{\tilde{K}-1} - \xi_{\tilde{K}})$$

$$= c^-(\mu - s_{\tilde{K}}) + \beta\left(c^+ - \frac{1 - \beta^{\tilde{K}}\gamma^{\tilde{K}}}{1 - \beta\gamma}c^-\right)(s_{\tilde{K}-1} - s_{\tilde{K}})$$

$$= c^-\frac{1 - \gamma^{\tilde{K}}}{\gamma^{\tilde{K}}}\left(\frac{\delta}{1-\gamma} - \mu\right) + \beta\left(c^+ - \frac{1 - \beta^{\tilde{K}}\gamma^{\tilde{K}}}{1 - \beta\gamma}c^-\right)\frac{1-\gamma}{\gamma^{\tilde{K}}}\left(\frac{\delta}{1-\gamma} - \mu\right) \quad (17)$$

$$= \underbrace{\left[\frac{\beta(1-\gamma)}{\gamma^{\tilde{K}}}c^+ + \frac{(1-\beta)(1-\gamma^{\tilde{K}}) - \beta\gamma^{\tilde{K}}(1-\gamma)(1-\beta^{\tilde{K}})}{\gamma^{\tilde{K}}(1 - \beta\gamma)}c^-\right]}_{=: (A)}\left(\frac{\delta}{1-\gamma} - \mu\right)$$

We claim that the term $(A)$ is positive. To see that, it is sufficient to show that $(1 - \beta)(1 - \gamma^{\tilde{K}}) - \beta\gamma^{\tilde{K}}(1 - \gamma)(1 - \beta^{\tilde{K}}) > 0$.

$$(1 - \beta)(1 - \gamma^{\tilde{K}}) - \beta\gamma^{\tilde{K}}(1 - \gamma)(1 - \beta^{\tilde{K}}) = \gamma^{\tilde{K}}(1 - \beta)(1 - \gamma)\left(\frac{1 - \gamma^{\tilde{K}}}{\gamma^{\tilde{K}}(1 - \gamma)} - \frac{\beta(1 - \beta^{\tilde{K}})}{1 - \beta}\right)$$

$$= \gamma^{\tilde{K}}(1 - \beta)(1 - \gamma)\left(\sum_{i=1}^{\tilde{K}}\frac{1}{\gamma^i} - \sum_{i=1}^{\tilde{K}}\beta^i\right)$$

$$= \gamma^{\tilde{K}}(1 - \beta)(1 - \gamma)\left(S(1/\gamma) - S(\beta)\right)$$

where $S(x) = \sum_{i=1}^{\tilde{K}} x^i$ the partial sum function. Since the partial sum is increasing in $x$, we have $S(1/\gamma) > S(\beta)$ as $\beta, \gamma \in (0, 1)$. This implies that $(A) > 0$ for any $\tilde{K}$.

By definition, $s_{\tilde{K}-1} > 0$, which is equivalent to $\mu > \frac{1-\gamma^{\tilde{K}-1}}{1-\gamma}\delta$. Then, as $(A)$ is a positive coefficient, we have

$$h(\max\{0, s_{\tilde{K}}\}) < \left[\beta c^+ + \frac{(1 - \beta)(1 - \gamma^{\tilde{K}}) - \beta\gamma^{\tilde{K}}(1 - \gamma)(1 - \beta^{\tilde{K}})}{(1 - \gamma)(1 - \beta\gamma)}c^-\right]\delta$$

$$< \left[\beta c^+ + \frac{1 - \beta}{(1 - \gamma)(1 - \beta\gamma)}c^-\right]\delta$$

$$\leq r + \beta c^+\delta$$

where the last inequality follows from Theorem 3.1's condition. Then, with the monotonicity of $h$ we established at the beginning of this proof, we conclude that the region where $W$ is strictly increasing lies entirely in $\{x \leq \mu \ \& \ (\star)\}$.

Lastly, we show that the running minimum of $\Phi(x)$ in Equation (12) coincides exactly with $W$.

**Analysis of the case when $x \in [\mu, \infty)$** For $x \geq \mu$, we know $\gamma x + \delta \geq \mu$ due to $\mu < \frac{\delta}{1-\gamma}$. Then,

$$\Phi(x) = (1 - \beta\gamma)c^+ x - (r + \beta c^+\delta) + \beta c^+(\gamma x + \delta - s_0) + \beta\xi_0 = c^+ x - \frac{r}{1 - \beta} = W(x).$$

As $\Phi(x)$ is strictly increasing on $[\mu, \infty)$, we immediately have $\min_{\tilde{x} \geq x} \Phi(\tilde{x}) = \Phi(x) = W(x), \forall x \geq \mu$.

**Analysis of the case when $x \in [s_{K-1}, \mu] \cap \mathbb{R}_{\geq 0}$** For $x \in [s_{K-1}, \mu] \cap \mathbb{R}_{\geq 0}$, we have $x \in [\omega, \mu]$ as we established just now. Then,

$$\Phi(x) = (1 - \beta)G(x) + \beta W(\gamma x + \delta)$$

$$= \beta\sum_{k=1}^{K-1}\left[\left(c^+ - \frac{1 - \beta^k\gamma^k}{1 - \beta\gamma}c^-\right)(\gamma x + \delta - s_k) + \xi_k\right]\mathbf{1}\{s_k \leq \gamma x + \delta < s_{k-1}\}$$

$$+ \beta\left[c^+(\gamma x + \delta - s_0) + \xi_0\right]\mathbf{1}\{\gamma x + \delta \geq s_0\} + \left[(1 - \beta\gamma)c^+ - c^-\right]x + c^-\mu - (r + \beta c^+\delta)$$

$$\overset{(a)}{=} \sum_{k=0}^{K-1}\left[\left(\beta\gamma c^+ - \beta\gamma\frac{1 - \beta^k\gamma^k}{1 - \beta\gamma}c^-\right)(x - s_{k+1}) + \beta\xi_k + \left[(1 - \beta\gamma)c^+ - c^-\right](x - s_{k+1})\right.$$

$$\left. + \left[(1 - \beta\gamma)c^+ - c^-\right]s_{k+1} + c^-\mu - (r + \beta c^+\delta)\right]\mathbf{1}\{s_{k+1} \leq x < s_k\}$$

$$\overset{(b)}{=} \sum_{k=0}^{K-1}\left[\left(c^+ - \frac{1 - \beta^{k+1}\gamma^{k+1}}{1 - \beta\gamma}c^-\right)(x - s_{k+1}) + \xi_{k+1}\right]\mathbf{1}\{s_{k+1} \leq x < s_k\}, \tag{18}$$

where in (a) and (b) we set temporary variables $s_K := \frac{s_{K-1}-\delta}{\gamma}$ and $\xi_K := \left(c^+ - \frac{1-\beta^K\gamma^K}{1-\beta\gamma}c^-\right)(s_K - s_{K-1}) + \xi_{K-1}$. In (b), we utilized the identity

$$\beta\xi_k + \left[(1 - \beta\gamma)c^+ - c^-\right]s_{k+1} + c^-\mu - r - \beta c^+\delta = \xi_{k+1}, \quad \forall k = 0, 1, \ldots, K - 1. \tag{19}$$

To see this, we apply an induction argument. For $k = 0$, Equation (19) already holds because

$$\beta\xi_0 + [(1 - \beta\gamma)c^+ - c^-]s_1 + c^-\mu - r - \beta c^+\delta$$

$$= \beta c^+ s_0 - \frac{\beta r}{1 - \beta} + [(1 - \beta\gamma)c^+ - c^-]s_1 + c^-\mu - r - \beta c^+\delta$$

$$= \beta c^+(s_0 - \delta) + [(1 - \beta\gamma)c^+ - c^-]s_1 + c^-\mu - \frac{r}{1 - \beta}$$

$$= \beta\gamma c^+ s_1 + [(1 - \beta\gamma)c^+ - c^-]s_1 - (c^+ - c^-)\mu + c^+\mu - \frac{r}{1 - \beta}$$

$$= (c^+ - c^-)(s_1 - s_0) + \xi_0 = \xi_1.$$

For the induction step, assume Equation (19) holds for 0 to $k$. Then, the induction shows

$$\beta\xi_{k+1} + [(1 - \beta\gamma)c^+ - c^-]s_{k+2} + c^-\mu - r - \beta c^+\delta$$

$$= \beta\xi_k + \left(\beta c^+ - \beta\frac{1 - \beta^{k+1}\gamma^{k+1}}{1 - \beta\gamma}c^-\right)(s_{k+1} - s_k) + [(1 - \beta\gamma)c^+ - c^-]s_{k+2} + c^-\mu - r - \beta c^+\delta$$

$$= \beta\xi_k + \left(\beta c^+ - \beta\frac{1 - \beta^{k+1}\gamma^{k+1}}{1 - \beta\gamma}c^-\right)\frac{(1 - \gamma)\mu - \delta}{\gamma^{k+1}} + [(1 - \beta\gamma)c^+ - c^-]\left(s_{k+1} + \frac{(1 - \gamma)\mu - \delta}{\gamma^{k+2}}\right) + c^-\mu - r - \beta c^+\delta$$

$$= \xi_{k+1} + \left(\beta\gamma c^+ - \beta\gamma\frac{1 - \beta^{k+1}\gamma^{k+1}}{1 - \beta\gamma}c^- + (1 - \beta\gamma)c^+ - c^-\right)\frac{(1 - \gamma)\mu - \delta}{\gamma^{k+2}}$$

$$= \xi_{k+1} + \left(c^+ - \frac{1 - \beta^{k+2}\gamma^{k+2}}{1 - \beta\gamma}c^-\right)\frac{(1 - \gamma)\mu - \delta}{\gamma^{k+2}}$$

$$= \xi_{k+1} + \left(c^+ - \frac{1 - \beta^{k+2}\gamma^{k+2}}{1 - \beta\gamma}c^-\right)(s_{k+2} - s_{k+1}) = \xi_{k+2}.$$

Back to Equation (18), we immediately see that $\Phi$ coincides with $W$ on $[s_{K-1}, \mu] \cap \mathbb{R}_{\geq 0}$, which is also strictly increasing.

**Analysis of the case when $x \in [0, s_{K-1}) \cap \mathbb{R}_{\geq 0}$**   This case is nonempty only when $s_{K-1} > 0$. In this case, the function $W(x)$ is constant: $W(x) = \xi_{K-1}$ for all $x \in [0, s_{K-1})$. First, notice that $c^+ - \frac{1-\beta^K\gamma^K}{1-\beta\gamma}c^- < 0$ by the definition of $K$. Thus, $\Phi$ is strictly decreasing on $[\omega, s_{K-1}) \cap \mathbb{R}_{\geq 0}$.

- If $\omega \leq 0$, this implies $\Phi$ is strictly decreasing on $[0, s_{K-1})$ with the minimum attained at $x = s_{K-1}$.

- If $\omega > 0$, $\Phi$ is first increasing on $[0, \omega)$ and then decreasing on $[\omega, s_{K-1})$. Using the conditions of Theorem 3.1, we notice

$$\xi_{K-1} < \xi_0 = c^+\mu - \frac{r}{1 - \beta} < c^+\mu - \frac{c^-\delta}{(1-\gamma)(1-\beta\gamma)} < c^+\left(\mu - \frac{\delta}{1-\gamma}\right) < 0.$$

  Thus, $\Phi(0) = \beta\xi_{K-1} > \xi_{K-1} = \Phi(s_{K-1})$.

Consequently, the minimum of $\Phi$ on $[0, s_{K-1})$ is $s_{K-1}$, regardless of the positivity of $\omega$. Therefore, for all $x \in [0, s_{K-1})$, we conclude that $\min_{\tilde{x} \geq x} \Phi(\tilde{x}) = \Phi(s_{K-1}) = \xi_{K-1} = W(s_{K-1})$. $\square$

The fixed point $W$ identified by Claim C.1 is flat on $[0, \max\{0, s_{K-1}\})$ and strictly increasing on $[\max\{0, s_{K-1}\}, \mu)$. The general shape of this function is illustrated in Figure 6.

C.3.2. DERIVING THE AGENT'S OPTIMAL ACTION

Recall $\tilde{x}^*$ is the optimizing point to Equation (11), which is also the post-response attribute of the agent. The proof of Claim C.1 indicates

$$\tilde{x}^* = \begin{cases} s_{K-1} & x \in [0, \max\{0, s_{K-1}\}) \\ 0 & x \in [\max\{0, s_{K-1}\}, \mu) \end{cases}$$

Setting $x^\circ = \max\{0, s_{K-1}\}$, Theorem 3.1 can be readily recovered: the optimal improvement action can be obtained by reverting the reparameterization, i.e., $(a^+)^* = \tilde{x}^* - x$, and the optimal gaming action can be obtained from Equation (14). $\square$

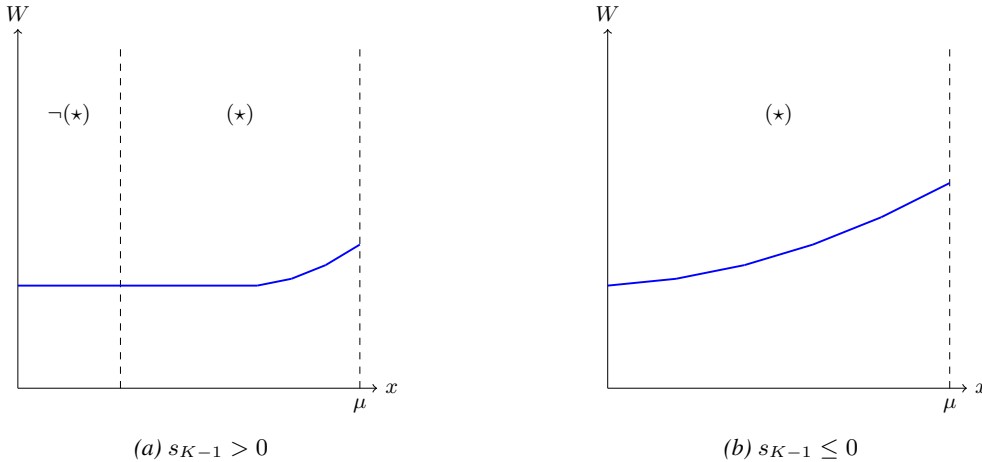

(a) $s_{K-1} > 0$          (b) $s_{K-1} \leq 0$

*Figure 6.* Illustration of the fixed-point $W$ in Theorem 3.1.

## C.4. Proof of Theorem 3.2

All ranges of $\mu$ considered in this theorem satisfy the condition $\mu > \frac{\delta}{1-\gamma}$. By Section C.2, $\Phi$ is strictly increasing on $[\mu, \infty)$, which implies that $(a^+)^* \equiv 0$ for all $x \geq \mu$. Furthermore, this condition on $\mu$ ensures that $\gamma x + \delta \in [0, \mu]$ for all $x \in [0, \mu]$. Consequently, it suffices to restrict the analysis in this proof to the interval $[0, \mu]$.

Table C.3 summarizes different cases to be discussed, conditions of each case, and the corresponding solutions.

### C.4.1. IDENTIFYING THE FIX-POINT SOLUTIONS

**Claim C.2.** *When* $\mu \in \left[\frac{\delta}{1-\gamma}, \frac{r+\beta c^+ \delta}{(1-\beta\gamma)c^+}\right)$, *we have* $W(x) \equiv G(\mu)$.

*Proof.* For $\mu \geq \frac{\delta}{1-\gamma}$, we have $\gamma x + \delta \in [0, \mu], \forall x \in [0, \mu]$. Thus, $\widetilde{W}(x) \equiv G(\mu)$. This implies $\Phi(\mu)$ is a local minimum of $\Phi$, while the other local minimum, if it exists, would be $\Phi(0)$. When $\mu < \frac{r}{(1-\beta\gamma)}c^+ + \frac{\beta\delta}{1-\beta\gamma}$, we have $\Phi(\mu) < \Phi(0)$, implying $\Phi(\mu)$ is the global minimum of $\Phi$ on $[0, \mu]$. Therefore, $W(x) = \min_{\tilde{x} \geq x} \Phi(\tilde{x}) = \Phi(\mu) = G(\mu)$. By uniqueness of the fixed point, we have $W(x) \equiv G(\mu)$. $\square$

**Claim C.3.** *When* $\mu \in \left[\max\left\{\frac{\delta}{1-\gamma}, \frac{r+\beta c^+ \delta}{(1-\beta\gamma)c^+}\right\}, \frac{c^- - (1-\beta)c^+}{\beta(1-\gamma)c^+ c^-}r + \frac{\delta}{1-\gamma}\right)$, *we have*

$$W(x) = c^+ x \mathbf{1}\left\{0 \leq x < \frac{G(\mu)}{c^+}\right\} + G(\mu)\mathbf{1}\left\{\frac{G(\mu)}{c^+} \leq x \leq \mu\right\}.$$

*Proof.* We first show that the "kink", attained at $x = \omega := \frac{G(\mu)}{c^+}$, lies in the region $\neg(\star)$. This is further divided into two cases: (1) when $\gamma\omega + \delta \geq \omega$, we notice $\mu \leq \frac{r}{(1-\beta\gamma)c^+} + \frac{\delta}{1-\gamma} < \frac{r+\beta c^+ \delta}{\beta(1-\gamma)c^+}$. We then plug it into the LHS of Equation (13) at $\omega$:

$$c^-(\mu - \omega) + \beta(G(\mu) - c^+ \gamma\omega) = \frac{\beta(1-\gamma)\left[(1-\beta\gamma)c^+ - c^-\right]\mu}{1-\beta} - \frac{\left[\beta(1-\gamma)c^+ - c^-\right](r + \beta c^+ \delta)}{(1-\beta)c^+}$$

$$\underset{\substack{> \\ \text{since } (1-\beta\gamma)c^+ < c^-}}{} \frac{\beta(1-\gamma)\left[(1-\beta\gamma)c^+ - c^-\right]}{1-\beta} \cdot \frac{r + \beta c^+ \delta}{\beta(1-\gamma)c^+} - \frac{\left[\beta(1-\gamma)c^+ - c^-\right](r + \beta c^+ \delta)}{(1-\beta)c^+} = r + \beta c^+ \delta;$$

*Table C.3.* Summary of different cases and the solutions of Theorem 3.2, where $K := \left\lceil \log_{\beta\gamma}\left(1 - (1-\beta\gamma)c^+/c^-\right) \right\rceil$.

| Claim | $\mu$ | "kink" | $W$ has plateau | shape of $W$ |
|---|---|---|---|---|
| Claim C.2 | $\left[\frac{\delta}{1-\gamma},\ \frac{r+\beta c^+\delta}{(1-\beta\gamma)c^+}\right)$ | not exist | yes |  |
| Claim C.3 | $\left[\max\left\{\frac{\delta}{1-\gamma},\ \frac{r+\beta c^+\delta}{(1-\beta\gamma)c^+}\right\},\ \frac{c^--(1-\beta)c^+}{\beta(1-\gamma)c^+c^-}r + \frac{\delta}{1-\gamma}\right)$ | in $\neg(\star)$ | yes |  |
| Claim C.4 | $\left[\frac{c^--(1-\beta)c^+}{\beta(1-\gamma)c^+c^-}r + \frac{\delta}{1-\gamma},\ \frac{r}{(1-\gamma^{K-1})c^-} + \frac{\delta}{1-\gamma}\right)$ | in $(\star)$ | yes |  |
| Claim C.5 | $\left[\max\left\{\frac{c^--(1-\beta)c^+}{\beta(1-\gamma)c^+c^-}r,\ \frac{r}{(1-\gamma^{K-1})c^-}\right\} + \frac{\delta}{1-\gamma},\ \infty\right)$ | not exist | no |  |

and (2) when $\gamma w + \delta < \omega$, we apply the condition of Case B to the LHS of Equation (13) at $\omega$:

$$c^-(\mu - \omega) + \beta c^+\delta = c^- \cdot \frac{r + \beta c^+\delta}{(1-\beta)c^+} - c^- \cdot \frac{\beta(1-\gamma)c^+\mu}{(1-\beta)c^+} + \beta c^+\delta$$

$$\underset{\text{Case B condition}}{>} c^- \cdot \frac{r + \beta c^+\delta}{(1-\beta)c^+} - c^- \cdot \frac{\beta(1-\gamma)c^+}{(1-\beta)c^+}\left(\frac{c^- - (1-\beta)c^+}{\beta(1-\gamma)c^+c^-}r + \frac{\delta}{1-\gamma}\right) + \beta c^+\delta$$

$$= \frac{c^-r + \beta c^+c^-\delta - [(c^- - (1-\beta)c^+)r + \beta c^+c^-\delta]}{(1-\beta)c^+} + \beta c^+\delta = r + \beta c^+\delta.$$

Second, notice that the LHS of Equation (13) is strictly decreasing. So, $\exists x^\circ \in [0,\mu]$ such that $\{x \le \mu \ \& \ \neg(\star)\} = [0, x^\circ)$ and $\{x \le \mu \ \& \ (\star)\} = [x^\circ, \mu]$. The above analysis shows $x^\circ > \frac{G(\mu)}{c^+}$. This implies $\Phi(x) = c^+x, \forall x \in [0, \frac{G(\mu)}{c^+})$.

We next show that $\Phi(x) \ge G(\mu), \forall x \in \left[\frac{G(\mu)}{c^+}, \mu\right]$. Notice $\Phi$ is increasing in $\left[\frac{G(\mu)}{c^+}, \max\left\{\frac{G(\mu)-c^+\delta}{\gamma c^+}, x^\circ\right\}\right)$ (e.g., if $\frac{G(\mu)-c^+\delta}{\gamma c^+} > x^\circ$, then, according to Equation (12), $\Phi(x) = \begin{cases} c^+x & \frac{G(\mu)}{c^+} \le x < x^\circ \\ (c^+ - c^-)x + c^-\mu - r & x^\circ \le x < \frac{G(\mu)-c^+\delta}{\gamma c^+} \end{cases}$ is strictly increasing). Besides, it is easy to verify that $\Phi$ is linearly decreasing in $\left[\max\left\{\frac{G(\mu)-c^+\delta}{\gamma c^+}, x^\circ\right\}, \mu\right]$. Therefore, we have, for $x \in \left[\frac{G(\mu)}{c^+}, \mu\right]$,

$$\Phi(x) = \Phi(x)\mathbf{1}\left\{x \in \left[\frac{G(\mu)}{c^+}, \max\left\{\frac{G(\mu)-c^+\delta}{\gamma c^+}, x^\circ\right\}\right)\right\} + \Phi(x)\mathbf{1}\left\{x \in \left[\max\left\{\frac{G(\mu)-c^+\delta}{\gamma c^+}, x^\circ\right\}, \mu\right]\right\}$$

$$\ge \Phi(\frac{G(\mu)}{c^+})\mathbf{1}\left\{x \in \left[\frac{G(\mu)}{c^+}, \max\left\{\frac{G(\mu)-c^+\delta}{\gamma c^+}, x^\circ\right\}\right)\right\} + \Phi(\mu)\mathbf{1}\left\{x \in \left[\max\left\{\frac{G(\mu)-c^+\delta}{\gamma c^+}, x^\circ\right\}, \mu\right]\right\} = G(\mu).$$

Then, it is easy to see $W(x) = c^+x\mathbf{1}\left\{0 \le x < \frac{G(\mu)}{c^+}\right\} + G(\mu)\mathbf{1}\left\{\frac{G(\mu)}{c^+} \le x \le \mu\right\}$. $\qquad \square$

**Claim C.4.** *Suppose* $\mu \in \left[\frac{c^- - (1-\beta)c^+}{\beta(1-\gamma)c^+c^-}r + \frac{\delta}{1-\gamma}, \frac{r}{(1-\gamma^{K-1})c^-} + \frac{\delta}{1-\gamma}\right)$, *where* $K := \left\lceil \log_{\beta\gamma}\left(1 - (1-\beta\gamma)c^+/c^-\right)\right\rceil$. *Define the sequence* $\{s_k, \xi_k\}_{k=0}^K$ *such that* $s_0 = \xi_0 = 0$, $s_k = \frac{(1-\gamma)(c^-\mu-r)-(1-\gamma^{k-1})c^-\delta}{\gamma^{k-1}(1-\gamma)c^-}$, $\xi_k = \left(c^+ - \frac{1-\beta^{k-1}\gamma^{k-1}}{1-\beta\gamma}c^-\right)(s_k - s_{k-1}) + \xi_{k-1}, \forall k = 1, 2, \ldots, K-1$, *and the last elements determined by*

$$\xi_K = \min\left\{G(\mu), \ \left(c^+ - \frac{\beta\gamma(1-\beta^{K-1}\gamma^{K-1})}{1-\beta\gamma}c^-\right)\mu - \left(r + \frac{\beta(1-\beta^{K-1}\gamma^{K-1})}{1-\beta\gamma}c^-\delta\right)\right.$$

$$\left. + \beta\left(\xi_{K-1} - \left(c^+ - \frac{1-\beta^{K-1}\gamma^{K-1}}{1-\beta\gamma}c^-\right)s_{K-1}\right)\right\}$$

*and* $s_K = \frac{(1-\beta\gamma)(\xi_K - \xi_{K-1})}{(1-\beta\gamma)c^+ - (1-\beta^{K-1}\gamma^{K-1})c^-} + s_{K-1}$. *Then,*

$$W(x) = \sum_{k=0}^{K-1}\left[\left(c^+ - \frac{1-\beta^k\gamma^k}{1-\beta\gamma}c^-\right)(x - s_k) + \xi_k\right]\mathbf{1}\{s_k \le x < s_{k+1}\} + \xi_K\mathbf{1}\{s_K \le x \le \mu\}. \tag{20}$$

*Proof.* Remark that $W$ has decreasing slope and thus the LHS of Equation (13) is strictly decreasing. This implies the existence of $x^\circ \in [0,\mu]$ such that $\{x \le \mu \ \& \ \neg(\star)\} = [0, x^\circ)$ and $\{x \le \mu \ \& \ (\star)\} = [x^\circ, \mu]$. We first show that $x^\circ = s_1$.

When Case C holds, $s_1 = \mu - \frac{r}{c^-} \ge \frac{c^- - (1-\beta\gamma)c^+}{\beta(1-\gamma)c^+c^-}r + \frac{\delta}{1-\gamma} \ge \frac{\delta}{1-\gamma}$. This implies $\gamma s_1 + \delta \le s_1$. Plugging this into the LHS of Equation (13) at $s_1$, we obtain $c^-(\mu - s_1) + \beta c^+\delta = r + \beta c^+\delta$, where the equality is attained. Thus, we conclude $x^\circ = s_1$. This implies $s_1, \ldots, s_K$ are all on the $(\star)$ region on which $G(x)$ is decreasing.

**Analysis of the case when** $x \in [0, s_1)$  For $x \in [0, s_1)$, it is easy to see that $\Phi(x) = W(x) = c^+x$.

**Analysis of the case when $x \in [s_1, \mu]$.** For $x \in [s_1, \mu]$, we start with the following identity

$$\beta\xi_k + [(1 - \beta\gamma)c^+ - c^-]s_{k+1} + c^-\mu - r - \beta c^+\delta = \xi_{k+1}, \quad \forall k = 1, \ldots, K - 2. \tag{21}$$

Note that this identity has the same form as Equation (19), differing only in the initial condition. For $k = 1$, we have:

$$\beta\xi_1 + [(1 - \beta\gamma)c^+ - c^-]s_2 + c^-\mu - r - \beta c^+\delta = c^+s_1 + (c^+ - c^-)\frac{(1 - \gamma)s_1 - \delta}{\gamma} = \xi_2.$$

We omit the full induction as it can be readily reproduced using the steps in Equation (19).

Then, using the identity $\gamma s_{k+1} + \delta = s_k, \forall k = 1, \ldots, K - 2$, and Equation (21), we can derive, for $x \in [s_1, \mu]$,

$$\begin{aligned}
\Phi(x) &= (1 - \beta)G(x) + \beta W(\gamma x + \delta) \\
&= \beta\sum_{k=0}^{K-1}\left[\left(c^+ - \frac{1 - \beta^k\gamma^k}{1 - \beta\gamma}c^-\right)(\gamma x + \delta - s_k) + \xi_k\right]\mathbf{1}\{s_k \le \gamma x + \delta < s_{k+1}\} + \beta\xi_K\mathbf{1}\{s_K \le \gamma x + \delta \le \mu\} \\
&\quad + [(1 - \beta\gamma)c^+ - c^-]x + c^-\mu - (r + \beta c^+\delta) \\
&= \sum_{k=0}^{K-2}\left[\left(\beta\gamma c^+ - \beta\gamma\frac{1 - \beta^k\gamma^k}{1 - \beta\gamma}c^-\right)(x - s_{k+1}) + [(1 - \beta\gamma)c^+ - c^-](x - s_{k+1})\right. \\
&\quad \left. + \beta\xi_k + [(1 - \beta\gamma)c^+ - c^-]s_{k+1} + c^-\mu - (r + \beta c^+\delta)\right]\mathbf{1}\{s_{k+1} \le x < \tilde{s}_{k+2}\} + \Gamma(x) \\
&= \sum_{k=0}^{K-2}\left[\left(c^+ - \frac{1 - \beta^{k+1}\gamma^{k+1}}{1 - \beta\gamma}c^-\right)(x - s_{k+1}) + \xi_{k+1}\right]\mathbf{1}\{s_{k+1} \le x < \tilde{s}_{k+2}\} + \Gamma(x),
\end{aligned}$$

where $\Gamma(x)$ hides the remaining terms, $\tilde{s}_k = s_k, \forall k = 1, \ldots, K - 1$, and $\tilde{s}_K = \frac{s_{K-1} - \delta}{\gamma}$.

Next, we show that $s_K < \tilde{s}_K$. Let $\tilde{\xi}_K = \xi_{K-1} + \left(c^+ - \frac{1 - \beta^{K-1}\gamma^{K-1}}{1 - \beta\gamma}c^-\right)(\tilde{s}_K - s_{K-1})$. Then, $(\xi_{K-1}, \tilde{s}_K, \tilde{\xi}_K)$ satisfies Equation (19). Notice $(s_K, \xi_K)$ and $(\tilde{s}_K, \tilde{\xi}_K)$ lie on the same increasing linear segment. Thus, it is sufficient to show $\xi_K < \tilde{\xi}_K$.

$$\begin{aligned}
\xi_K - \tilde{\xi}_K &\le \left(c^+ - \frac{\beta\gamma(1 - \beta^{K-1}\gamma^{K-1})}{1 - \beta\gamma}c^-\right)\mu - \left(r + \frac{\beta(1 - \beta^{K-1}\gamma^{K-1})}{1 - \beta\gamma}c^-\delta\right) + \\
&\quad \beta\left(\xi_{K-1} - \left(c^+ - \frac{1 - \beta^{K-1}\gamma^{K-1}}{1 - \beta\gamma}c^-\right)s_{K-1}\right) - \tilde{\xi}_K \\
&\overset{\text{by Equation (19)}}{=} \left(c^+ - \frac{\beta\gamma(1 - \beta^{K-1}\gamma^{K-1})}{1 - \beta\gamma}c^-\right)\mu - \left(r + \frac{\beta(1 - \beta^{K-1}\gamma^{K-1})}{1 - \beta\gamma}c^-\delta\right) + \\
&\quad \beta\left(\xi_{K-1} - \left(c^+ - \frac{1 - \beta^{K-1}\gamma^{K-1}}{1 - \beta\gamma}c^-\right)s_{K-1}\right) - \beta\xi_{K-1} - [(1 - \beta\gamma)c^+ - c^-]\tilde{s}_K - c^-\mu + r + \beta c^+\delta \\
&= \left(c^+ - \frac{1 - \beta^K\gamma^K}{1 - \beta\gamma}c^-\right)\mu + \beta\left(c^+ - \frac{1 - \beta^{K-1}\gamma^{K-1}}{1 - \beta\gamma}c^-\right)(\delta - s_{K-1}) - [(1 - \beta\gamma)c^+ - c^-]\tilde{s}_K \\
&= \left(c^+ - \frac{1 - \beta^K\gamma^K}{1 - \beta\gamma}c^-\right)(\mu - \tilde{s}_K).
\end{aligned}$$

By definition of $K$, we have $c^+ - \frac{1 - \beta^K\gamma^K}{1 - \beta\gamma}c^- \le 0$. Besides, since $\mu < \frac{r}{(1 - \gamma^{K-1})c^-} + \frac{\delta}{1 - \gamma}$ for Case C, we have

$$\mu - \tilde{s}_K = \frac{-(1 - \gamma)(1 - \gamma^{K-1})c^-\mu + (1 - \gamma)r + (1 - \gamma^{K-1})c^-\delta}{\gamma^{K-1}(1 - \gamma)c^-} > 0, \tag{22}$$

where the inequality is obtained by plugging in $\mu < \frac{r}{(1 - \gamma^{K-1})c^-} + \frac{\delta}{1 - \gamma}$. Thus, we have $\xi_K < \tilde{\xi}_K$ and $s_K < \tilde{s}_K = \frac{s_{K-1} - \delta}{\gamma}$.

The analysis so far implies that $\Phi(x)$ coincides with $W(x)$ on $[0, s_K)$, where both functions are strictly increasing: by definition of $K$, $c^+ - \frac{1 - \beta^k\gamma^k}{1 - \beta\gamma}c^- > 0, \forall k \le K - 1$. However, $\Phi(x)$ keeps increasing in $[s_K, \tilde{s}_K)$, while $W(x)$ is flat on

this interval. Using this fact, it is easy to see that $\Phi(x)$ decreases with the same rate as $(1-\beta)G(x)$ over $[\tilde{s}_K, \mu]$. As long as $\Phi(\mu) = \Phi(s_K)$, we know $\Phi(x) \geq \Phi(\mu)$, $\forall x \in [s_K, \mu]$, which further concludes $\min_{\tilde{x} \geq x} \Phi(\tilde{x}) = W(x)$. Then, $W$ being the fixed point follows from the uniqueness theorem.

To show $\Phi(\mu) = \Phi(s_K) = W(s_K)$. first notice $\Phi(s_K) = W(s_K) = \xi_K$. Denote

$$g(x) := \left(c^+ - \frac{1 - \beta^{K-1}\gamma^{K-1}}{1 - \beta\gamma}c^-\right)(x - s_{K-1}) + \xi_{K-1}$$

(i.e., the linear segment of $W$ on $[s_{K-1}, s_K)$) and it follows $g(s_K) = \xi_K$. Observe that

$$\xi_K = \min\left\{G(\mu), (1-\beta)G(\mu) + \beta g(\gamma\mu + \delta)\right\}.$$

When $\gamma\mu + \delta \geq s_K$, by monotonicity of $g$, $\xi_K \geq (1-\beta)G(\mu) + \beta g(\gamma\mu + \delta) \geq (1-\beta)G(\mu) + \beta\xi_K \implies \xi_K = G(\mu)$. In this case, it is easy to see that $\Phi(\mu) = (1-\beta)G(\mu) + \beta W(\gamma x + \delta) = (1-\beta)G(\mu) + \beta\xi_K = \xi_K$.

When $\gamma\mu + \delta < s_K$, by monotonicity of $g$, $(1-\beta)G(\mu) + \beta g(\gamma\mu + \delta) < (1-\beta)G(\mu) + \beta g(s_K) = (1-\beta)G(\mu) + \beta\xi_K$. This can only occur when $\xi_K = (1-\beta)G(\mu) + \beta g(\gamma\mu + \delta) < G(\mu)$. Additionally, we notice $\gamma\mu + \delta - s_{K-1} = \gamma(\mu - \tilde{s}_K) > 0$ by Equation (22). This means $\gamma\mu + \delta \in [s_{K-1}, s_K)$ and we have $\Phi(\mu) = (1-\beta)G(\mu) + \beta W(\gamma\mu + \delta) = (1-\beta)G(\mu) + \beta g(\gamma\mu + \delta) = \xi_K$, which completes the proof. $\square$

**Claim C.5.** *Suppose* $\mu \in \left[\max\left\{\frac{c^- - (1-\beta)c^+}{\beta(1-\gamma)c^+c^-}r, \frac{r}{(1-\gamma^{K-1})c^-}\right\} + \frac{\delta}{1-\gamma}, \infty\right)$, *where $K$ is defined in* Claim C.4. *Recall the sequence* $\{s_k, \xi_k\}_{k=0}^{K-1}$ *constructed in* Claim C.4 *(ignoring the last element). Then,*

$$W(x) = \sum_{k=0}^{K-1}\left[\left(c^+ - \frac{1 - \beta^k\gamma^k}{1 - \beta\gamma}c^-\right)(x - s_k) + \xi_k\right]\mathbf{1}\{s_k \leq x < \min\{s_{k+1}, \mu\}\}. \tag{23}$$

*Proof.* Denote $\tilde{K} := \max\{k : s_k < \mu\}$. Notice that $\tilde{K} \leq K - 1$ because, when Case D holds, the inequality of Equation (22) is reversed, implying $\mu \leq \tilde{s}_K$. In addition, the following results in the proof of Case C still applies: (1) $\{x \leq \mu \& \neg(\star)\} = [0, s_1)$ and $\{x \leq \mu \& (\star)\} = [s_1, \mu]$; and (2) for $x \in [0, s_1)$, $\Phi(x) = c^+x$ (which is exactly $W(x)$ on $[0, s_1)$), and for $x \in [s_1, \mu]$,

$$\Phi(x) = \sum_{k=0}^{\tilde{K}-2}\left[\left(c^+ - \frac{1 - \beta^{k+1}\gamma^{k+1}}{1 - \beta\gamma}c^-\right)(x - s_{k+1}) + \xi_{k+1}\right]\mathbf{1}\{s_{k+1} \leq x < s_{k+2}\}$$

$$+ \left[\left(c^+ - \frac{1 - \beta^{\tilde{K}}\gamma^{\tilde{K}}}{1 - \beta\gamma}c^-\right)(x - s_{\tilde{K}}) + \xi_{\tilde{K}}\right]\mathbf{1}\{s_{\tilde{K}} \leq x < \mu\}. \tag{24}$$

By definition of $K$, $c^+ - \frac{1 - \beta^k\gamma^k}{1 - \beta\gamma}c^- > 0$, $\forall k \leq K - 1$. Thus, $\Phi(x)$ is strictly increasing on $[0, \mu]$, and thus $W(x) = \Phi(x)$, $\forall x \in [0, \mu]$. The proof is completed by noticing Equation (24) is exactly the same as Equation (23) on $[s_1, \mu]$. $\square$

### C.4.2. DERIVING THE AGENT'S OPTIMAL ACTION

- Theorem 3.2-(1) is a combination of Case A and B, with $\underline{\mu} := \frac{c^- - (1-\beta)c^+}{\beta(1-\gamma)c^+c^-}r$ and $\underline{x} := \max\left\{0, \frac{G(\mu)}{c^+}\right\}$;

- Theorem 3.2-(2) corresponds to Case C, with $\overline{\mu} := \max\left\{\frac{c^- - (1-\beta)c^+}{\beta(1-\gamma)c^+c^-}r, \frac{r}{(1-\gamma^{K-1})c^-}\right\}$ and $\overline{x} = s_K$;

- Theorem 3.2-(3) corresponds to Case D.

Similar to Section C.3, the optimal improvement action and gaming action can be easily obtained by inverting the reparameterization of $\tilde{x}$ and Equation (14), respectively. $\square$

## D. Principal's Optimal Design

We start with the necessary conditions for a threshold sequence $\vec{\mu}$ to be feasible to the problem $\mathbf{P}(M, r)$:

**Lemma D.1.** *If $\vec{\mu}$ is feasible to $\mathbf{P}(M, r)$, then*

(1) *(Large Final Threshold) $\mu_L \geq \frac{\delta(L-1)}{1-\gamma}$.*

(2) *(Promotion Region) $W_l(x) \equiv \Phi_l^{(5)}(\mu_{l+1})$, $\forall x \in [\max\{\mu_l, \frac{\delta(l-1)}{1-\gamma}\}, \mu_{l+1}]$, $\forall l \in [L-1]$.*

(3) *(Relegation Region) For all $l \in [L-1] \setminus \{1\}$, if $\mu_l \geq \frac{\delta(l-1)}{1-\gamma}$, then*

$$W_l(x) \equiv \Phi_l^{(5)}(\mu_{l+1}), \quad \forall x \in \big[\min\{\gamma\mu_l + \delta(l-1), \max\{\mu_{l-1}, \frac{\delta(l-2)}{1-\gamma}\}\}, \mu_{l+1}\big].$$

*In addition, if $\mu_L \geq \frac{\delta(L-1)}{1-\gamma}$, then $W_L(x) \equiv \Phi_l^{(2)}(\mu_L)$, $\forall x \in [\min\{\gamma\mu_{L-1} + \delta(L-2), \max\{\mu_{L-1}, \frac{\delta(L-2)}{1-\gamma}\}\}, \mu_L]$.*

**Implications** Lemma D.1-(1) implies that for $\vec{\mu}$ to be feasible, the last threshold cannot be smaller than $\frac{\delta(L-1)}{1-\gamma}$. Lemma D.1-(2) implies that, under a feasible threshold sequence $\vec{\mu}$, an agent whose current attribute is higher than the threshold of the current level should have an incentive to honestly improve its attribute to the next threshold. Lemma D.1 further requires that the agent whose attribute is larger than the previous level should always improve.

*Proof.* For Lemma D.1-(1), suppose the opposite (i.e., $\mu_L < \frac{\delta(L-1)}{1-\gamma}$) holds. Then, there exist $x_0 < \mu_L$ such that $\gamma x_0 + \delta(L-1) \geq \mu_L$. That means, for an agent with attribute $x_0$, the attribute dynamic of level $L$ will keep it in the same level, even though no actions are taken. Thus, the optimal strategy of an agent with initial level $l_0 = L-1$ and attribute $x_0$ is to choose $(a_0^+)^* = 0$ and $(a_0^-)^* = \mu_L - x_0$ at $t = 0$, and exert no efforts in all subsequent time steps, since the attribute dynamics will maintain its attribute above $\mu_L$ in all future time steps. This violates the no-gaming constraint of $\mathbf{P}(M, r)$, and therefore we must have $\mu_L \geq \frac{\delta(L-1)}{1-\gamma}$.

For Lemma D.1-(2), let $\mathcal{D} := [\max\{\mu_l, \frac{\delta(l-1)}{1-\gamma}\}, \mu_{l+1}]$. We first show that $W_l(x) \equiv W_l(\mu_{l+1})$, $\forall x \in \mathcal{D}$ by contradiction. Suppose there exists $x_0 \in \mathcal{D}$ such that $W_l(x_0) < W_l(\mu_{l+1})$ (notice $W_l$ is non-decreasing). We observe that an agent with state $l_t \leq l$ and $x_t \leq x_0$ will never improve its attribute beyond $x_0$ due to two separate cases:

- If $l_t = l$, this is directly implied by $W_l(x_0) < W_l(\mu_{l+1})$ and Equation (10).

- If $l_t < l$, we know by the non-decreasing monotonicity of $\Phi_l^{(6)}$ in Equation (8) that the agent never improves its attribute beyond $\max\{x_t, \mu_{l_t}\}$, which is also no greater than $x_0$.

Given $\vec{\mu}$ is feasible, this optimal strategy does not involve gaming actions. Consequently, its next state satisfies $l_{t+1} \leq l$ and $x_{t+1} = \gamma x_{t_+} + \delta(l-1) \leq \gamma x_0 + \delta(l-1)$. Since $x_0 > \frac{\delta(l-1)}{1-\gamma}$, we have $x_{t+1} \leq x_0$. Thus, the agent is trapped in the region $l_t \leq l$, violating the constraint $\lim_{t \to \infty} l_t = L$ of $\mathbf{P}(M, r)$.

Next, we show that $W_l(\mu_{l+1}) = \Phi_l^{(5)}(\mu_{l+1})$. Since $\Phi_l$ is strictly increasing on $[\mu_{l+1}, \infty)$, we have $W_l(\mu_{l+1}) = \Phi_l(\mu_{l+1})$. Combining this with the fact that $W_l(x)$ is constant on $\mathcal{D}$, we obtain:

$$W_l(x) \equiv \Phi_l(\mu_{l+1}) = \min\{\Phi_l^{(4)}(\mu_{l+1}), \Phi_l^{(5)}(\mu_{l+1})\}, \quad \forall x \in \mathcal{D}.$$

This implies that $\mu_{l+1}$ minimizes $\Phi_l(x)$ over the interval $\mathcal{D}$. Finally, since $\Phi_l^{(4)}$ is strictly increasing, the minimum must be determined by $\Phi_l^{(5)}(\mu_{l+1})$. If it were determined by $\Phi_l^{(4)}$, then $\Phi_l$ would take smaller values at the left end of $\mathcal{D}$, contradicting the constancy of $W_l$. Thus, $W_l(\mu_{l+1}) = \Phi_l^{(5)}(\mu_{l+1})$.

For Lemma D.1-(3), we first consider $l \in [L-1] \setminus \{1\}$. From Equation (8), we observe that $\Phi_{l-1}^{(4)} = \Phi_l^{(1)}$ and $\Phi_{l-1}^{(5)} = \Phi_l^{(2)}$. Besides, Lemma D.1-(2) on level $l-1$ implies

$$\min\big\{\min_{\tilde{x} \in \mathcal{D}} \Phi_{l-1}^{(4)}(\tilde{x}), \min_{\tilde{x} \in \mathcal{D}} \Phi_{l-1}^{(5)}(\tilde{x})\big\} = \Phi_{l-1}^{(5)}(\mu_l).$$

This allows us to simplify Equation (9) for $W_l$ into

$$W_l(x) = \min\left\{\Phi_{l-1}^{(5)}(\mu_l), \min_{\tilde{x} \in [x, \mu_{l+1}]} \Phi_l^{(5)}(\tilde{x})\right\}, \quad \forall x \in \left[\max\left\{\mu_{l-1}, \frac{\delta(l-2)}{1-\gamma}\right\}, \mu_l\right]. \tag{25}$$

As $\mu_l \geq \frac{\delta(l-1)}{1-\gamma}$, Lemma D.1-(2) on level $l$ implies $\Phi_l^{(4)}(\mu_l) \geq W_l(\mu_l) = \Phi_l^{(5)}(\mu_{l+1})$. Observe $\Phi_l^{(2)}(\mu_l) = \Phi_l^{(4)}(\mu_l)$. We thus obtain

$$\Phi_{l-1}^{(5)}(\mu_l) = \Phi_l^{(2)}(\mu_l) \geq \Phi_l^{(5)}(\mu_{l+1}). \tag{26}$$

Plugging this into Equation (25), we derive $W_l(x) \equiv \Phi_l^{(5)}(\mu_{l+1}), \forall x \in [\max\{\mu_{l-1}, \frac{\delta(l-2)}{1-\gamma}\}, \mu_{l+1}]$ and $l \in [L-1] \setminus \{1\}$.

For the last level, observe that $\Phi_{L-1}(x) = \Phi_L(x), \forall x \in [\mu_{L-1}, \mu_L)$. Thus, by Lemma D.1-(2), we have $W_L(x) = W_{L-1}(x) = \Phi_{l-1}^{(5)}(\mu_L) = \Phi_l^{(2)}(\mu_L), \forall x \in [\max\{\mu_{L-1}, \frac{\delta(L-2)}{1-\gamma}\}, \mu_L)$.

Finally, we show that $W_l(x) = W_l(\mu_{l+1})$ for $l \in [\gamma\mu_l + \delta(l-1), \mu_{l+1}]$ for all $l \geq 2$ (we treat $\mu_{L+1} = \mu_L$ here to simplify notations). From the discussion above, we know that an agent who reaches $l_t = l$ from $l_{t-1} = l-1$ would have an attribute $x_t \leq \gamma\mu_l + \delta(l-1)$, since its post-response attribute in the previous timestep (i.e., $x_{(t-1)_+}$) never exceeds $\mu_l$. If $W_l(\gamma\mu_l + \delta(l-1)) < W_l(\mu_{l+1})$, Lemma D.1-(2) implies $W_l(\gamma\mu_l + \delta(l-1)) < W_l(\mu_l)$ as $\mu_l \geq \frac{\delta(l-1)}{1-\gamma}$. Then, by the no-gaming constraint of $\mathbf{P}(M, r)$, the agent's state variables would be $z_t = x_{t_+} < \mu_l$, meaning the agent would be demoted back to $l_{t+1} = l-1$ with the next-state attribute $x_{t+1} \leq \gamma\mu_l + \delta(l-1)$. Thus, the level $l-1$ is recurrent, which violates the constraint $\lim_{t \to \infty} l_t = L$ of $\mathbf{P}(M, r)$. $\qquad \square$

### D.1. Proof of Theorem 4.1

A direct consequence of $\vec{\mu}$ being feasible to $\mathbf{P}(M, r)$ is that $(L, \gamma\mu_L + \delta)$ is an *absorbing state* for the state process $\{(l_t, x_t)\}_{t \geq 0}$. In other words, once the agent is in this state, the optimal strategy ensures the agent remains at the final level $L$ indefinitely. In this proof, we introduce the vector-form notation of the action $\vec{a} := [a^+, a^-]^\top$.

We proceed by contradiction. Suppose the state $(L, \gamma\mu_L)$ is an absorbing state under the optimal strategy $\vec{a}^*$. Then, it must satisfy $\vec{a}^*(L, \gamma\mu_L) = [(1-\gamma)\mu_L, 0]^\top$: the agent has to restore its depreciated attribute $(\gamma\mu_L)$ to $\mu_L$ in the subsequent timestep. Now, consider the alternative strategy $\vec{a}'$ such that

$$\vec{a}'(l, x) = \begin{cases} [\gamma\mu_L - x, 0]^\top & \text{if } x < \gamma\mu_L \\ [0, 0]^\top & \text{if } x \geq \gamma\mu_L \end{cases}, \quad \forall l \in [L].$$

This strategy always restores the agent's attribute to $\gamma\mu_L$ regardless of the level. It is straightforward to see that $\vec{a}' \neq \vec{a}^*$. Let $U'(l, x)$ and $U^*(l, x)$ denote the agent's utilities with the **initial state** $(l, x)$ under the strategies $\vec{a}'$ and $\vec{a}^*$, respectively. It is easy to see that $U^*(L, \gamma\mu_L) = \sum_{t=0}^\infty \beta^t(rL - c^+(1-\gamma)\mu_L) = \frac{rL - c^+(1-\gamma)\mu_L}{1-\beta}$.

For $U'(L, \gamma\mu_L)$, define $K \in [L-2]$ as the integer such that $\mu_{L-K-1} \leq \gamma\mu_L < \mu_{L-K}$. Thus, we have

$$U'(L, \gamma\mu_L) = r\left[\sum_{t=0}^{K-1} \beta^t(L-t-1) + (L-K-1)\sum_{t=K}^\infty \beta^t\right] - \sum_{t=0}^\infty \beta^t c^+(1-\gamma)\gamma\mu_L$$

$$\geq r\left[\frac{L}{1-\beta} - \sum_{t=0}^\infty \beta^t(t+1)\right] - \frac{c^+(1-\gamma)\gamma\mu_L}{1-\beta}.$$

Then, we observe

$$U^*(L, \gamma\mu_L) - U'(L, \gamma\mu_L) \leq r\sum_{t=0}^\infty \beta^t(t+1) - \frac{c^+(1-\gamma)^2}{1-\beta}\mu_L = \frac{r}{(1-\beta)^2} - \frac{c^+(1-\gamma)^2}{1-\beta}\mu_L < 0$$

where the equality applies $\sum_{t=0}^\infty \beta^t(t+1) = \frac{d}{d\beta}\left(\sum_{t=0}^\infty \beta^{t+1}\right) = \frac{d[\beta(1-\beta)^{-1}]}{d\beta} = \frac{1}{(1-\beta)^2}$, and the last inequality utilized the condition of Theorem 4.1. This is a contradiction to $\vec{a}^*$ being an optimal Markov policy, implying $(L, \gamma\mu_L)$ can never be an absorbing state if $\mu_L > \frac{r}{(1-\beta)(1-\gamma)^2 c^+}$. Therefore, the problem $\mathbf{P}(M, r)$ with $M \geq \frac{r}{(1-\beta)(1-\gamma)^2 c^+}$ can never be feasible. $\qquad \square$

## D.2. Proof of Theorem 4.2

### D.2.1. PROOF OF THEOREM 4.2-(1)

A consequence of Lemma D.1 is that each $\mu_l$ in a feasible sequence $\vec{\mu}$ should be sufficiently large, as established below.

**Lemma D.2.** *If $\vec{\mu}$ is feasible to $\mathbf{P}(M, r)$ and $r < \frac{1-\beta}{1-\gamma} c^+ \delta$, then $\mu_l > \frac{\delta(l-1)}{1-\gamma}$, $\forall l \in [L-1]$.*

*Proof.* For simplicity, we use the notation $m_l := W_l(\mu_{l+1})$ in the rest of the proof and $\mu_{L+1} = \mu_L$ to handle the boundary case automatically. By the hypothesis and Equation (26), we have, for all $k \geq l$, $m_{k-1} = \Phi_{k-1}^{(5)}(\mu_k) \geq \Phi_k^{(5)}(\mu_{k+1}) = m_k$. In particular, we have $m_L = m_{L-1}$ as $\Phi_{L-1}(x) = \Phi_L(x)$, $\forall x \in [\mu_{L-1}, \mu_L]$.

We construct the proof by induction from level $L-1$. For the base case, suppose the opposite $\mu_{L-1} \leq \frac{\delta(L-2)}{1-\gamma}$ holds. Then,

$$\Phi_{l-1}^{(5)}(\mu_L) - \Phi_{l-1}^{(4)}\left(\frac{\delta(L-2)}{1-\gamma}\right) =$$
$$(1-\beta\gamma)c^+\left(\mu_L - \frac{\delta(L-2)}{1-\gamma}\right) - (r + \beta c^+ \delta) + \beta\left[W_L(\gamma\mu_L + \delta(L-1)) - W_{L-1}\left(\gamma\frac{\delta(L-2)}{1-\gamma} + \delta(L-2)\right)\right].$$

Since $\vec{\mu}$ is feasible to $\mathbf{P}(M, r)$, Lemma D.1-(1) implies $\mu_L \geq \frac{\delta(L-1)}{1-\gamma}$. Thus, both $\gamma\mu_L + \delta(L-1)$ and $\gamma\frac{\delta(L-2)}{1-\gamma} + \delta(L-2)$ lie in $[\mu_{L-1}, \mu_L]$, on which $W_L$ and $W_{L-1}$ are exactly the same. Therefore, we have

$$\Phi_{L-1}^{(5)}(\mu_L) - \Phi_{L-1}^{(4)}\left(\frac{\delta(L-2)}{1-\gamma}\right) \geq (1-\beta\gamma)c^+\frac{\delta}{1-\gamma} - (r + \beta c^+ \delta) = \frac{1-\beta}{1-\gamma}c^+\delta - r > 0.$$

This implies $W_{L-1}\left(\frac{\delta(L-2)}{1-\gamma}\right) \leq \Phi_{L-1}^{(4)}\left(\frac{\delta(L-2)}{1-\gamma}\right) < \Phi_{L-1}^{(5)}(\mu_L)$, which is a contradiction to Lemma D.1-(2). Therefore, we must have $\mu_{L-1} > \frac{\delta(L-2)}{1-\gamma}$.

For the hypothesis, suppose $\mu_k > \frac{\delta(k-1)}{1-\gamma}$, $\forall k \geq l$, for some $l \in [L-1] \setminus \{1\}$. For the induction step, suppose the opposite (i.e., $\mu_{l-1} \leq \frac{\delta(l-2)}{1-\gamma}$) holds. Consider an agent with state $l_t = l-1$ and $x_t = \frac{\delta(l-2)}{1-\gamma}$. Since $\vec{\mu}$ is feasible, Lemma D.1-(2) implies that the agent will improve its attribute up to $\mu_l$. This results in the agent's value (negative utility) of

$$v^* = c^+(\mu_l - x_t) - r(l-1) + \beta V_l(\gamma\mu_l + \delta(l-1))$$
$$= -c^+x_t + (1-\beta\gamma)c^+\mu_l - (r + \beta c^+ \delta)(l-1) + \beta W_l(\gamma\mu_l + \delta(l-1))$$
$$\geq -c^+x_t + \underbrace{(1-\beta\gamma)c^+\mu_l - (r + \beta c^+ \delta)(l-1)}_{=:T_a} + \beta m_L.$$

where the last inequality utilized Lemma D.1-(3), the hypothesis $\mu_l \geq \frac{\delta(l-1)}{1-\gamma}$, and the monotonicity of $m_l$.

To determine the value of $m_L$, notice that the expression of $\Phi_l^{(2)}$ involves $W_L$ itself and $m_L = \Phi_l^{(2)}(\mu_L)$ by Lemma D.1-(3). Thus, $m_L$ can be obtained by solving the equation

$$m_L = (1-\beta\gamma)c^+\mu_L - (r + \beta c^+ \delta)(L-1) + \beta m_L \implies m_L = \frac{(1-\beta\gamma)c^+\mu_L - (r + \beta c^+ \delta)(L-1)}{1-\beta}.$$

Now consider an alternative strategy where the agent exerts nothing at time $t$ (i.e., $\vec{a}(l-1, \frac{\delta(l-2)}{1-\gamma}) = \vec{0}$). In this case, the agent will remain at level $l-1$ where the attribute dynamic will keep its attribute at the same level: $\gamma \cdot \frac{\delta(l-2)}{1-\gamma} + \delta(l-2) = \frac{\delta(l-2)}{1-\gamma}$. The agent's value under this strategy is $v^{\text{alt}} = -\frac{r(l-2)}{1-\beta}$. Expanding this term, we obtain

$$v^{\text{alt}} = -c^+x_t + \underbrace{\frac{1-\beta}{1-\gamma}c^+\delta(l-2) - r(l-2)}_{=:T_b} + \beta\underbrace{\left(\frac{\frac{1-\beta}{1-\gamma}c^+\delta(l-2) - r(l-2)}{1-\beta}\right)}_{=:\tilde{m}}.$$

By the hypothesis where $\mu_l \geq \frac{\delta(l-1)}{1-\gamma}$, we observe

$$T_a \geq (1-\beta\gamma)c^+ \frac{\delta(l-1)}{1-\gamma} - (r+\beta c^+\delta)(l-1) = \frac{1-\beta}{1-\gamma}c^+\delta(l-1) - r(l-1) > T_b,$$

where the last inequality is due to the condition $r < \frac{1-\beta}{1-\gamma}c^+\delta$. For the last term, consider

$$m_L - \tilde{m} \geq \frac{1}{1-\beta}\left(\frac{1-\beta}{1-\gamma}c^+\delta - r\right)(L-l+1) > 0,$$

where the first inequality applies the hypothesis $\mu_L \geq \frac{\delta(L-1)}{1-\gamma}$. Therefore, we obtain $v^* > v^{\text{alt}}$, which conflicts with the constraints of $\mathbf{P}(M,r)$ because it is not incentive compatible for this agent to improve its attribute for a promotion, who would rather stay at the level $l-1$ indefinitely. Thus, it has to hold $\mu_{l-1} \geq \frac{\delta(l-2)}{1-\gamma}$. $\qquad\square$

Under the conditions of Theorem 4.2-(1), Theorem D.2 implies $\mu_1 > 0$, which conflicts with the setup $\mu_1 = 0$. Therefore, $\vec{\mu}$ can never be feasible to $\mathbf{P}(M,r)$ when $r < \frac{1-\beta}{1-\gamma}c^+\delta$. $\qquad\square$

### D.2.2. PROOF OF THEOREM 4.2-(2)

Notice that each threshold $\mu_l = \frac{\delta(l-1)}{1-\gamma}$ is the steady state of the attribute dynamic when an agent stays in level $l$ forever. This means if an agent reaches level $l$ with post-response attribute $x_{t_+} = \mu_l$, it will not lose any attribute in the next stage, or specifically, $x_{t+1} = x_{t_+}$. Thus, it is sufficient to show that $W_l(x) \equiv W_l(\mu_{l+1}), \forall x \in [\mu_l, \mu_{l+1}]$ for $\vec{\mu}$ to be feasible.

**Designing the complete metric space** Consider the extended threshold sequence $\{\mu_l\}_{l=0}^{L+1}$ where $\mu_l = \frac{\delta(l-1)}{1-\gamma}, \forall l \in [L]$, $\mu_0 = \mu_1$, and $\mu_{L+1} = \mu_L$. Define $\mathcal{F}$ as the set of non-decreasing continuous functions on $[L] \times [0, \mu_L]$ such that, $\forall F \in \mathcal{F}$,

(PA) $F(l, x)$ is $(1-\beta^k\gamma^k)c^+$-Lipschitz on $x \in [\mu_{l-k-1}, \mu_{l-k}], \forall k \in [l-1], \forall l \in [L]$.

(PB) $F(l, \mu_l) \geq F(l+1, \mu_{l+1}), \forall l \in [L-1]$.

We show that $\mathcal{F}$ is closed under supremum norm of functions (i.e., $\|f - g\|_\infty := \sup_{x \in [0, \mu_L]} |f(x) - g(x)|$). Let $\{F_n\}_n$ be a convergent sequence of functions in $\mathcal{F}$, and $F^*$ be the limit of $\{F_n\}_n$ under $\|\cdot\|_\infty$.

(PA) For any $x_1, x_2 \in [\mu_{l-k-1}, \mu_{l-k}]$, as the absolute value is continuous everywhere,

$$|F^*(l, x_1) - F^*(l, x_2)| = \lim_{n\to\infty} |F_n(l, x_1) - F_n(l, x_2)| \leq (1-\beta^k\gamma^k)c^+.$$

Thus, $F^*$ is also $(1-\beta^k\gamma^k)c^+$-Lipschitz on $x \in [\mu_{l-k-1}, \mu_{l-k}]$.

(PB) $F^*(l, \mu_l) \geq F^*(l+1, \mu_{l+1})$ directly follows from the pointwise convergence and that weak inequalities are preserved under limits of real numbers.

Notice that all functions in $\mathcal{F}$ are bounded. Thus, $(\mathcal{F}, \|\cdot\|)$ is a complete metric space.

**Application of the Banach fixed-point theorem** Denote $T$ as the Bellman operator of Equation (7). $T$ is clearly a contraction mapping because $\beta \in (0, 1)$. Let $F_n \in \mathcal{F}$ and $F_{n+1} := TF_n$. We next show that $F_{n+1} \in \mathcal{F}$. The monotonicity and continuity of $F_{n+1}$ are obvious. Observe[5]

$$\Phi_l^{(2)}(\mu_l) - \Phi_l^{(1)}(\mu_{l-1}) = (1-\beta\gamma)c^+(\mu_l - \mu_{l-1}) - (r+\beta c^+\delta) + \beta\left[F_n(l, \mu_l) - F_n(l-1, \mu_{l-1})\right]$$

$$= (1-\beta\gamma)c^+\frac{\delta}{1-\gamma} - (r+\beta c^+\delta) + \beta[F_n(l, \mu_l) - F_n(l-1, \mu_{l-1})] \leq 0,$$

---

[5]We drop the notational dependence of $\Phi_l$ on $n$ (or fundamentally on $F_n$), which should be cleared in the context.

where the inequality is due to the condition $r > \frac{1-\beta}{1-\gamma}c^+\delta$ and (PB). Besides, (PA) implies that $F_n(l, x)$ is constant on $x \in [\mu_{l-1}, \mu_l]$, which further implies that $\Phi_{l-1}^{(2)}(x)$ is strictly decreasing on the same interval. As $\Phi_l^{(1)}$ is strictly increasing, we conclude that $\min\limits_{\tilde{x} \in [x, \mu_l]} \min\{\Phi_l^{(1)}(\tilde{x}), \Phi_l^{(2)}(\tilde{x})\} = \Phi_l^{(2)}(\mu_l), \forall x \in [\mu_{l-1}, \mu_l]$. Noticing that $\Phi_l^{(4)}$ and $\Phi_l^{(5)}$ are the same as $\Phi_{l+1}^{(1)}$ and $\Phi_{l+1}^{(2)}$, respectively, we can directly conclude $F_{n+1}(l, x) \equiv \Phi_l^{(5)}(\mu_{l+1}), \forall x \in [\mu_l, \mu_{l+1}]$.

Besides, using the above discussion and the formulation Equation (9), we can simplify the expression of $F_{n+1}$ on $x \in [\mu_{l-1}, \mu_l]$ into

$$F_{n+1}(l, x) = \min\left\{\Phi_l^{(2)}(\mu_l), \min_{\tilde{x} \in [x, \mu_{l+1}]} \Phi_l^{(5)}\right\}.$$

From the above discussion, we know $\Phi_l^{(5)} = \Phi_{l+1}^{(2)}$ is strictly decreasing on $[\mu_l, \mu_{l+1}]$, so the minimum of $\Phi_l^{(5)}$ on this interval is attained at $x = \mu_{l+1}$. Denote $x^*$ as the minimum of $\Phi_l^{(5)}$ on $[\mu_{l-1}, \mu_l]$. Then,

$$\begin{aligned}
\Phi_l^{(5)}(\mu_{l+1}) - \Phi_l^{(5)}(x^*) &= \Phi_l^{(5)}(\mu_{l+1}) - \Phi_l^{(5)}(\mu_l) + \Phi_l^{(5)}(\mu_l) - \Phi_l^{(5)}(x^*) \\
&\leq \Phi_l^{(5)}(\mu_{l+1}) - \Phi_l^{(5)}(\mu_l) + [(1-\beta\gamma)c^+ - c^- + \beta\gamma(1-\beta\gamma)c^+](\mu_l - \mu_{l-1}) \\
&= [(2+\beta\gamma)(1-\beta\gamma)c^+ - 2c^-]\frac{\delta}{1-\gamma},
\end{aligned}$$

where the inequality utilized the $(1-\beta\gamma)c^+$-Lipschitz property of $F_n(l+1, \cdot)$ on $[\mu_{l-1}, \mu_l]$. Under the condition $c^- \geq (1 + \frac{\beta\gamma}{2})(1-\beta\gamma)c^+$, we have $\Phi_l^{(5)}(\mu_{l+1}) \leq \Phi_l^{(5)}(x^*)$, which implies $F_{n+1}(l, x) = \min\{\Phi_l^{(2)}(\mu_l), \Phi_l^{(5)}(\mu_{l+1})\}$. Observe that $\Phi_l^{(2)}(\mu_l) = \Phi_l^{(4)}(\mu_l)$ and from the discussion above on $\Phi_l^{(1)}$ and $\Phi_l^{(2)}$, we know $\Phi_l^{(4)}(\mu_l) \geq \Phi_l^{(5)}(\mu_{l+1})$. Therefore, we conclude that $F_{n+1}(l, x) \equiv \Phi_l^{(5)}(\mu_{l+1}), \forall x \in [\mu_{l-1}, \mu_{l+1}]$, implying $F_{n+1}(l, \cdot)$ is 0-Lipschitz on $[\mu_{l-1}, \mu_l]$.

Let $H_l$ denote the Lipschitz constant of $F_{n+1}(l, \cdot)$, and $H_l^{(a)}$, $H_l^{(b)}$, and $H_l^{(c)}$ denote the Lipschitz constants of $\Phi_l^{(1)}$, $\Phi_l^{(2)}$, and $\Phi_l^{(3)}$, respectively, the index $n$ and the interval with which this Lipschitz constant is associated are omitted for clarity . Then, it satisfies

$$H_l \leq \max\{H_l^{(a)}, H_l^{(b)}, H_l^{(c)}\}.$$

By (PA), we obtain, for $x \in [\mu_{l-k-1}, \mu_{l-k}], \forall k \in [l-1] \setminus \{1\}$,

$$H_l^{(a)} = (1-\beta\gamma)c^+ + \beta\gamma(1 - \beta^{k-1}\gamma^{k-1})c^+ = (1 - \beta^k\gamma^k)c^+.$$

Using $c^- \geq (1-\beta\gamma)c^+$, we have

$$\begin{aligned}
H_l^{(b)} &= (1-\beta\gamma)c^+ - c^- + \beta\gamma(1 - \beta^k\gamma^k) \\
&= (1 - \beta^{k+1}\gamma^{k+1})c^+ - c^- \\
&= (1 - \beta^k\gamma^k)c^+ + [\beta^k\gamma^k(1-\beta\gamma)c^+ - c^-] \leq H_l^{(a)}.
\end{aligned}$$

When $c^- \geq \beta\gamma(1 - \beta^2\gamma^2)c^+$, we further have

$$H_l^{(c)} = (1 - \beta^k\gamma^k)c^+ + [\beta^k\gamma^k(1 - \beta^2\gamma^2)c^+ - c^-] \leq H_l^{(a)}.$$

Therefore, $F_{n+1}(l, \cdot)$ is also $(1 - \beta^k\gamma^k)c^+$-Lipschitz on $[\mu_{l-k-1}, \mu_{l-k}]$.

For (PB), from the above discussion, we have obtained $F_{n+1}(l, \mu_l) = \Phi_l^{(5)}(\mu_{l+1})$. Since $\Phi_l^{(5)}(\mu_{l+1}) = \Phi_{l+1}^{(2)}(\mu_{l+1}) = \Phi_{l+1}^{(4)}(\mu_{l+1})$, it directly follows that $F_{n+1}(l, \mu_l)$ is non-increasing in $l$.

We have shown that $T$ is a contraction mapping on $\mathcal{F} \to \mathcal{F}$. As $\mathcal{F}$ is a complete metric space, the Banach fixed-point theorem implies that the fixed point, $\{W_l\}_{l \in [L]}$, is also in $\mathcal{F}$, and thereby completes the proof.

**Optimality of the sequence** Theorem D.2 can be modified to show the following: If $\vec{\mu}$ is feasible to $\mathbf{P}(M, r)$ and $r = \frac{1-\beta}{1-\gamma}c^+\delta$, then $\mu_l \geq \frac{\delta(l-1)}{1-\gamma}$, the proof of which is omitted since the reasoning steps are quite similar. Suppose there exists a sequence $\{\mu_l\}_{l \in [L]}$ with $L < \lceil (1-\gamma)M/\delta \rceil$ that is feasible to $\mathbf{P}(M, r)$. Let $k \in [L]$ be the highest level such that

$\mu_k = \frac{\delta(k-1)}{1-\gamma}$. Notice that $k$ always exists and satisfies $k \geq 1$ and $k \leq L - 1$. Then, similar to the proof of Theorem D.2, an agent at level $l_t = k$ with attribute $x_t = \mu_k$ has no incentive to improve up to $\mu_{k+1}$ because exerting no actions in all times yields strictly higher utility (or strictly lower negative utility). This is in contradiction to the fact that $\vec{\mu}$ is feasible. Therefore, the optimal solution to $\mathbf{P}(M, r)$ is $L^* = \lceil (1 - \gamma)M/\delta \rceil$. $\qquad\square$

### D.3. Level-Dependent Leg-Up Effect

We consider a generalization of the attribute dynamics in which the leg-up bonus depends arbitrarily on the level attained:

$$x_{t+1} = \gamma\, x_{t_+} + \delta_{l_{t+1}}, \tag{27}$$

where $\{\delta_l\}_{l \geq 1}$ is a non-negative, non-decreasing sequence (with $\delta_1 = 0$). The original model is the special case $\delta_l = \delta_{l-1}$. The following conditions, which generalize Theorem 4.2, are obtained by replacing $\delta l$ with $\delta_l$ in the feasibility analysis and applying the same proof technique as in Section D.2.

**Proposition D.3.** *Suppose the attribute dynamics follow Equation (27). Define the average leg-up increment between levels $l$ and $k$ as $\bar{\delta}_{lk} := \frac{\delta_l - \delta_k}{l - k}$ for $l \neq k$.*

(a) **Infeasibility.** *If $r < \frac{1-\beta}{1-\gamma} \min_{l > k \geq 1} \bar{\delta}_{lk}$, then $\mathbf{P}(M, r)$ is infeasible for all $M$.*

(b) **Feasibility.** *If $r \geq \frac{1-\beta}{1-\gamma} \max_{l > k \geq 1} \bar{\delta}_{lk}$ and the cost condition of Theorem 4.2-(2) holds($c^- \geq \max\{(1 + \frac{\beta\gamma}{2})(1 - \beta\gamma)c^+,\ \beta\gamma(1 - \beta^2\gamma^2)c^+\}$), then $\mathbf{P}(M, r)$ is feasible for all $M < \frac{\delta_\infty}{1-\gamma}$, where $\delta_\infty := \sup_{l \geq 1} \delta_l$. A feasible threshold sequence is $\mu_l = \frac{\delta_l}{1-\gamma}$ for $l \in [L]$.*

**Connection to original model.** When $\delta_l = \delta_{l-1}$, we have $\bar{\delta}_{lk} = \delta$ for all $l > k$, so $\min = \max = \delta$, conditions (a)/(b) reduce to Theorem 4.2-(1)/(2), and the feasible threshold sequence $\mu_l = \frac{\delta_{l-1}}{1-\gamma}$ matches Theorem 4.2-(2).

**Concave and saturating leg-up.** The key implication of Theorem D.3 depends on whether $\delta_l$ saturates:

- **Non-saturating** ($\delta_\infty = \infty$, e.g., $\delta_l = \delta\sqrt{l}$ or $\delta\log(1 + l)$): Theorem D.3-(b) gives feasibility for all $M$, matching the original linear case. Although threshold gaps $\mu_{l+1} - \mu_l = \frac{\delta_{l+1} - \delta_l}{1-\gamma}$ shrink as $l$ grows (reflecting diminishing leg-up increments), the principal can always design an arbitrarily long feasible ladder.
- **Saturating** ($\delta_\infty < \infty$, e.g., $\delta_l = \delta(1 - e^{-l})$, or $\delta_l = \delta$ constant): the achievable target attribute is bounded by $\frac{\delta_\infty}{1-\gamma}$. Once the threshold sequence reaches the saturation ceiling, no additional level can provide meaningful leg-up, and the mechanism cannot survive future improvement beyond that point.

**Numerical illustration.** Figure 8 illustrates these theoretical predictions under four leg-up shapes: constant ($\delta_l = \delta$), linear ($\delta_l = \delta l$), square root ($\delta_l = \delta\sqrt{l}$), and log ($\delta_l = \delta\log(1 + l)$). The base parameters are $\beta = 0.8$, $\gamma = 0.8$, $c^+ = 0.8$, $c^- = 0.7$, $\delta = 0.01$, and thresholds are optimized via CMA-ES for each shape separately. Utilities across all shapes remain within a narrow range (643–650), confirming that the mechanism is robust to the specific form of the leg-up function.

## E. Numerical Results

### E.1. Interpolated Value Iteration: Computing Optimal Agent Strategy

We use value iteration to solve the fixed-point problem of Equation (7) and derive the optimal strategy via Equation (10). As the attribute space is continuous, we truncate the attribute space by a sufficiently large maximum attribute $\bar{X} > 0$ and discretize the attribute space with a grid step $\Delta x$. In each iteration, we linearly interpolate the $W$ functions on the grid. The details of the algorithm are provided in Algorithm 2.

**Theorem E.1.** *Let $\{W^{(n)}\}_{n \geq 0}$ be the sequence generated by Algorithm 2. The following properties hold:*

(a) **Convergence:** *The sequence converges linearly to a unique fixed point $W^\dagger$. The number of iterations $n$ required to reach a tolerance $\varepsilon$ is bounded by:*

$$n = O\left(\frac{\log(\|W^{(0)} - W^\dagger\|_\infty/\varepsilon)}{|\log\beta|}\right).$$

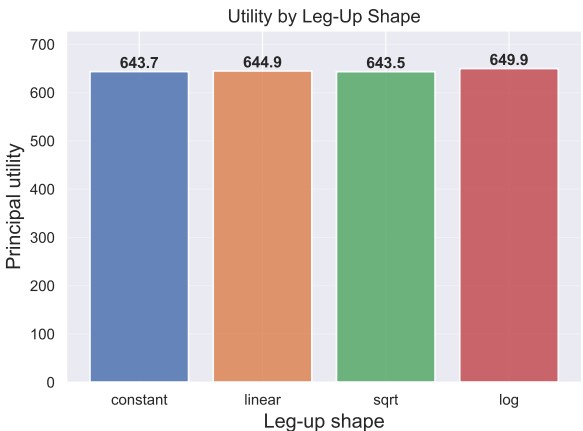

*Figure 7.* Principal utility by leg-up shape.

*Figure 8.* Robustness of the mechanism to alternative leg-up shapes.

---

**Algorithm 2** Interpolated Value Iteration (VALUEITERATE)

---

1: **Input:** thresholds $\vec{\mu}$, discrete attribute space $\mathcal{X}$, and tolerance $\epsilon$
2: **Output:** discretized optimal impr action $a^+(l,x)$ and gaming action $a^-(l,x)$
3: $L \leftarrow |\vec{\mu}|$
4: $\tilde{r} \leftarrow r + \beta c^+ \delta$
5: $\tilde{c}^+ \leftarrow (1 - \beta\gamma)c^+$
6: $W^{(0)}(l,x) \leftarrow 0, \ \forall l \in [L], x \in \mathcal{X}$
7: **for** $n = 1, 2, \ldots$ **until** $\|W^{(n)} - W^{(n-1)}\| \leq \epsilon$ **do**
8:    **for** $l \in [L], \ x \in \mathcal{X}$ **do**
9:       $\Omega^{(n)}(l,\cdot) \leftarrow$ linear interpolation of $W^{(n-1)}(l,\cdot)$
10:       $v_{\text{rel}}^{(n)} \leftarrow \tilde{c}^+ x - \tilde{r}\max\{1, l-1\} + \beta\Omega^{(n)}(\max\{1, l-1\}, \gamma x + \delta\max\{1, l-1\})$
11:       $v_{\text{stay}}^{(n)} \leftarrow \tilde{c}^+ x + c^-(\mu_l - x)_+ - \tilde{r}l + \beta\Omega^{(n)}(l, \gamma x + \delta l)$
12:       $v_{\text{pr}}^{(n)} \leftarrow \tilde{c}^+ x + c^-(\mu_{\min\{l+1,L\}} - x)_+ - \tilde{r}\min\{l+1, L\} + \beta\Omega^{(n)}(\min\{l+1, L\}, \gamma x + \delta\min\{l+1, L\})$
13:       $\Phi^{(n)}(l,x) \leftarrow \min\{v_{\text{rel}}^{(n)}, v_{\text{stay}}^{(n)}, v_{\text{pr}}^{(n)}\}$
14:    **end for**
15:    $W^{(n+1)}(l,x) \leftarrow \min_{\tilde{x} \in \mathcal{X}, \tilde{x} \geq x} \Phi(l, \tilde{x}), \forall l \in [L], x \in \mathcal{X}$
16: **end for**
17: $a^+(l,x) \leftarrow \max\{\tilde{x} \in \mathcal{X} : W^{(n)}(l,\tilde{x}) = W^{(n)}(l,x)\}, \forall l \in [L], x \in \mathcal{X}$
18: **for** $l \in [N], \ x \in \mathcal{X}$ **do**
19:    $\tilde{x} \leftarrow x + a^+(l,x)$ {Post-response attribute}
20:    **if** $\Phi^{(n)}(l,x) = v_{\text{rel}}^{(n)}$ **then**
21:       $a^-(l,x) \leftarrow 0$ {No effort is made if the result is relegation}
22:    **else if** $\Phi^{(n)}(l,x) = v_{\text{stay}}^{(n)}$ **then**
23:       $a^-(l,x) = \max\{0, \ \mu_l - \tilde{x}\}$
24:    **else**
25:       $a^-(l,x) = \max\{0, \ \mu_{l+1} - \tilde{x}\}$
26:    **end if**
27: **end for**
28: Return $a^+$ and $a^-$

---

*(b)* ***Approximation Accuracy:*** *Let $W^*$ be the true fixed point of Equation (7) and $\Omega^\dagger$ be the linear interpolation of the*

*solution $W^\dagger$ produced by Algorithm 2. Then, $\Omega^\dagger$ approximates $W^*$ with the following error bound:*

$$\|W^* - \Omega^\dagger\|_\infty \leq \frac{c^+ \Delta x}{2(1-\beta)}.$$

**Implication** Theorem E.1 implies a **linear convergence rate** of Algorithm 2 and the approximation accuracy of the solution found by Algorithm 2 is increasing in the improvement cost $c^+$, grid step $\Delta x$, and the agent's farsightedness $\beta$.

*Proof.* According to the monotonicity of $\Phi_l^{(6)}$ in Equation (8), there is no effort involved if $x \geq \mu_L$. Thus, we can set $X$ large enough such that the minimum of Equation (6) is attained within $\tilde{x} \in [0, \bar{X}]$, i.e., $W(l,x) = \min_{\tilde{x} \in [x, \bar{X}]} \Phi_l(\tilde{x} \mid W)$.

By discretization, each iterate $W^{(n)}$ can be viewed as a function on the grid $[L] \times \mathcal{X}$. Let $D$ be the linear interpolation operator such that $D \circ W^{(n)}$ is a function that maps $[L] \times [0, \bar{X}]$ to $\mathbb{R}$, obtained by linearly interpolating $W^{(n)}$. With a slight abuse of notation, let $\Phi$ denote the operator of Equation (8) $(\Phi \circ F)(l,x) = \Phi_l(x \mid F)$, $\forall l \in [L]$, $x \in [0, \bar{X}]$. Let $B$ be the running minimum operator such that

$$(B \circ F)(l,x) = \min_{\tilde{x} \in [x, \bar{X}]} F(l, \tilde{x}), \quad \forall (l,x) \in [L] \times [0, \bar{X}],$$

and $\tilde{B}$ be its discretized version on the grid $\mathcal{X}$, i.e.,

$$(\tilde{B} \circ F)(l,x) = \min_{\tilde{x} \in \mathcal{X}, \tilde{x} \geq x} F(l,x), \quad \forall (l, \tilde{x}) \in [L] \times \mathcal{X}.$$

It follows that the original (non-discretized) Bellman operator is $T = B \circ \Phi$. The iteration in Algorithm 2 can be written as $W^{(n+1)} = \tilde{B} \circ \Phi \circ D \circ W^{(n)}$. For the running minimum operator, observe

$$|\min_x f(x) - \min_x g(x)| \leq \max_x |f(x) - g(x)|$$

because otherwise the distance $|f(z) - g(z)|$ where $z := \arg\min_x f(x)$ would be strictly larger than $\max_x |f(x) - g(x)|$. This implies that the $B$ operators are non-expansive:

$$\|B \circ F - B \circ G\|_\infty \leq \|F - G\|_\infty, \quad \forall F, G : [L] \times [0, \bar{X}] \to \mathbb{R},$$

and so does the $\tilde{B}$ operator. It is easy to verify that the linear interpolation operator is also non-expansive because linear interpolation is a convex combination of neighboring grid values. Thus, we obtain

$$\|W^{(n+1)} - W^{(n)}\|_\infty \leq \beta \|W^{(n)} - W^{(n-1)}\|_\infty,$$

due to the $\Phi$ operator being a $\beta$-contraction. Since $0 < \beta < 1$, the sequence $\{W^{(n)}\}_{n \geq 0}$ will converge to a unique limit. Thus,

$$\|W^{(n)} - W^\dagger\|_\infty = \|(\tilde{B} \circ \Phi \circ D)(W^{(n-1)} - W^\dagger)\|_\infty \leq \beta \|W^{(n-1)} - W^\dagger\|_\infty \leq \beta^n \|W^{(0)} - W^\dagger\|_\infty.$$

Thus, to achieve $\varepsilon$ approximation error, we require at least $n \geq \frac{\log \varepsilon - \log \|W^{(0)} - W^\dagger\|_\infty}{\log \beta} = \frac{\log(\|W^{(0)} - W^\dagger\|_\infty / \varepsilon)}{|\log \beta|}$ iterations.

To show the approximation accuracy, we first show that $\Phi_l(\cdot \mid W^*)$ is $c^+$-Lipschitz. We claim that $W^*(l, \cdot)$ is Lipschitz without proof, which should be obvious following the induction argument of Section D.2.2. Let this Lipschitz constant be $K$. Then, by Equation (8), we have the left derivative of $\Phi_l$, denoted by $\Phi_l'$, is bounded by

$$\Phi_l' \leq (1 - \beta\gamma)c^+ + \beta\gamma K. \tag{28}$$

As $W^*(l,x) = \min_{\tilde{x} \geq x} \Phi_l(\tilde{x})$, we have $K \leq (1 - \beta\gamma)c^+ + \beta\gamma K \implies K \leq c^+$. Thus, $W^*(l, \cdot)$ is $c^+$-Lipschitz. It follows from Equation (28) that $\Phi_l(\cdot \mid W^*)$ is also Lipschitz with constant $(1 - \beta\gamma)c^+ + \beta\gamma c^+ = c^+$.

Let $\tilde{T} := D \circ \tilde{B} \circ \Phi$ be the iteration operator used in Algorithm 2 such that $\Omega^{(n+1)} = \tilde{T}\Omega^{(n)}$. Then, we can decompose the difference between the true fixed point $W^*$ and the interpolation of the convergence limit $\Omega^\dagger$ as follows:

$$\|W^* - \Omega^\dagger\|_\infty = \|TW^* - \tilde{T}\Omega^\dagger\|_\infty \leq \underbrace{\|TW^* - \tilde{T}W^*\|_\infty}_{=:\varepsilon_A} + \underbrace{\|\tilde{T}W^* - \tilde{T}\Omega^\dagger\|_\infty}_{=:\varepsilon_B}, \tag{29}$$

Thus, using the contraction property of $\tilde{T}$, we have $\varepsilon_B \leq \beta \|W^* - \Omega^\dagger\|_\infty$. Plugging this back to Equation (29), we have

$$\|W^* - \Omega^\dagger\|_\infty \leq \frac{1}{1-\beta} \|TW^* - \tilde{T}W^*\|_\infty \tag{30}$$

For $\varepsilon_A$, let $i, j \in \mathcal{X}$ denotes any two neighboring points in the grid $\mathcal{X}$ such that $j > x$. For $p \in [0,1]$, denote $(i,j)_p := pi + (1-p)j \in [i,j]$ as a point in the interval. Then, we have

$$\varepsilon_A = \|(B - D \circ \tilde{B})(\Phi W^*)\|_\infty$$
$$= \max_{l \in [L]} \max_{i,j \in \mathcal{X}} \max_{p \in [0,1]} \left| \min_{\tilde{x} \in [(i,j)_p, \bar{X}]} \Phi_l(\tilde{x} \mid W^*) - \left( p \min_{\tilde{x} \in \mathcal{X}, \tilde{x} \geq i} \Phi_l(\tilde{x} \mid W^*) + (1-p) \min_{\tilde{x} \in \mathcal{X}, \tilde{x} \geq j} \Phi_l(\tilde{x} \mid W^*) \right) \right|$$

Let $z \in [0, \bar{X}]$ be the minimizer of $\min_{\tilde{x} \in [(i,j)_p, \bar{X}]} \Phi_l(\tilde{x} \mid W^*)$ and $i_z$ and $j_z$ be two neighbors in $\mathcal{X}$ such that $i_z \leq z < j_z$. Then, we observe

$$\left| \min_{\tilde{x} \in [(i,j)_p, \bar{X}]} \Phi_l(\tilde{x} \mid W^*) - \min_{\tilde{x} \in \mathcal{X}, \tilde{x} \geq i} \Phi_l(\tilde{x} \mid W^*) \right| \leq \min_{\tilde{x} \in \{i_z, j_z\}} |\Phi_l(\tilde{x} \mid W^*) - \Phi_l(z \mid W^*)| \leq c^+ \frac{\Delta x}{2},$$

where the last inequality utilized the Lipschitz constant of $\Phi_l(\cdot, W^*)$ and the fact that $\min\{|i_z - z|, |j_z - z|\} \leq \frac{|i-j|}{2}$. Similarly, the distance between $\min_{\tilde{x} \in [(i,j)_p, \bar{X}]} \Phi_l(\tilde{x} \mid W^*)$ and $\min_{\tilde{x} \in \mathcal{X}, \tilde{x} \geq j} \Phi_l(\tilde{x} \mid W^*)$ is also bounded above by $c^+ \frac{\Delta x}{2}$. Therefore, we obtain $\varepsilon_A \leq \frac{c^+ \Delta x}{2}$. Combining this with Equation (30), we obtain $\|W^* - \Omega^\dagger\|_\infty \leq \frac{c^+ \Delta x}{2(1-\beta)}$. $\qquad \square$

### E.2. The Greedy Heuristic

The proposed greedy heuristic is shown in Algorithm 1.

#### E.2.1. PROOF OF THEOREM 5.1

**Notations**  Let $\vec{\mu}^{(l)}$ be the threshold sequence generated in the $(l-1)$-th iteration. Define $\Theta^{(l)}$ as the system with threshold sequence $\vec{\mu}^{(l)}$. We use superscripts to denote quantities defined in each system $\Theta^{(l)}$ (e.g., $W^{(l)}$ and $\Phi^{(l)}$).

**Lemma E.2.** *In $\Theta^{(l)}$, an agent at state $s_0 = (l_0, x_0) = (1, 0)$, under its optimal long-term strategy, takes the shortest path to reach level $l$ using improvement efforts only (i.e., improve to reach level $l$ in exactly $l$ steps).*

*Proof.* We construct the proof by induction.

**Base case**  When $l = 2$, from Section C, we know that $W_1^{(2)} \equiv W_2^{(2)}$. Thus, line 11 ensures that the least competent agent with $x_0 = 0$ has an incentive to improve for staying in level 2.

**Induction step**  Assume the statement holds for $\vec{\mu}^{(k)}$ for all $k \leq l$. Denote the entry state of this agent to level $l$ by $s^*$, and it is easy to see that $s^* = (l, \gamma \mu_l + \delta(l-1))$. Let this strategy be $\pi^{(l)}$. Thus, the agent's optimal value (negative utility) in $\Theta^{(l)}$ can be written as
$$V^{(l),*}(s_0) = V_{\pi^{(l)}}^{(l)}(s_0) = C(\pi^{(l)}) + \beta^l V_{\pi^{(l)}}^{(l)}(s^*),$$

where $C(\pi)$ denotes the *path cost* of the agent from $s_0 = (1, 0)$ to the first time reaching $s^*$ following the strategy $\pi$. In cases when $\pi$ involves levels higher than $l$, $C$ takes them all as staying at level $l$ via improvement only. Notice that this path cost under the same strategy is the same for all systems $\Theta^{(\tilde{l})}$ as long as $\tilde{l} \geq l$.

Consider the sequence $\vec{\mu}^{(l+1)}$ where $\mu_{l+1}^{(l+1)}$ is obtained by Algorithm 1 given prior thresholds $\vec{\mu}^{(l)}$. Define $\pi^{(l+1)}$ as the optimal agent's strategy.

- If $\pi^{(l+1)}$ never leads this agent to level $l$, this is dominated by $\pi^{(l)}$ according to the optimality of $\pi^{(l)}$ in system $\Theta^{(l)}$.

- Suppose $\pi^{(l+1)}$ first leads the agent to level $l$ in $T > l$ steps and its entrance attribute is $x_T \leq \gamma\mu + \delta(l-1)$. Notice that $x_T$ can never be larger because the agent has no incentive to improve its attribute beyond $\mu_l$ in the previous stage, according to Equation (8). Particularly, $x_T < \gamma\mu + \delta(l-1)$ indicates a non-zero gaming action is involved in the

previous stage. In this case, the agent's value satisfies (notice that by construction of $\mu^{(l+1)}$ in Algorithm 1, the agent at level $l$ will get promoted to $l+1$ and remain in $l+1$ via improvement actions indefinitely)

$$V_{\pi^{(l+1)}}^{(l+1)}(s_0) = C(\pi^{(l+1)}) + \beta^T V_{\pi^{(l+1)}}^{(l+1)}(l, x_T). \tag{31}$$

By optimality of $\pi^{(l)}$ in $\Theta^{(l)}$, we notice

$$C(\pi^{(l)}) + \beta^l V_{\pi^{(l)}}^{(l)}(s^*) \leq C(\pi^{(l+1)}) + \beta^T V_{\pi^{(l)}}^{(l)}(s^*) \implies C(\pi^{(l)}) < C(\pi^{(l+1)})$$

because $T > l$. With this observation, Equation (31) implies

$$V_{\pi^{(l+1)}}^{(l+1)}(s_0) \geq C(\pi^{(l+1)}) + \beta^T V_{\pi^{(l+1)}}^{(l+1)}(s^*) \geq C(\pi^{(l)}) + \beta^l V_{\pi^{(l+1)}}^{(l+1)}(s^*).$$

In the first inequality, we utilized the fact that $V$ is decreasing in the attribute $x$ because $W_l$ is $c^+$-Lipschitz in $x$ by the proof of Section E.1 and $V(l, x) = W(l, x) - c^+ x$; in the second inequality, we utilized the fact that, by the optimality of $\pi^{(l+1)}$, $V_{\pi^{(l+1)}}^{(l+1)}(s^*)$ must be negative or else doing nothing at all times would dominate $\pi^{(l+1)}$. The overall inequality means that $\pi^{(l+1)}$ is dominated by an alternative strategy where the agent reaches level $l$ in the minimum number of steps, after which it follows the original $\pi^{(l+1)}$.

Therefore, in $\Theta^{(l+1)}$, the agent starting from $(1, 0)$ will reach level $l$ following the shortest path and via improvement actions only, and then, according to the construction of $\mu_{l+1}$ in Algorithm 1, it exerts a one-step improvement to reach the last level $l+1$ in which it will stay indefinitely. $\qquad\square$

To show that this holds for all agents with $l_0 = 1$ and $x_0 \geq 0$, observe that the attribute dynamics in level 1 involve depreciation only. Thus, either an agent reaches the last level directly without gaming, or its state enters the region $\{0\} \times [0, \mu_2]$ after a finite number of steps, from which the agent would follow the "shortest-promotion path" described above to reach the last level without gaming. $\qquad\square$

### E.2.2. ABLATION TESTS ON THRESHOLDS WITH THE GREEDY ALGORITHM

To evaluate the sensitivity of the multi-level mechanism to environmental parameters, we conduct a series of ablation tests using the greedy heuristic (Algorithm 1). Here, we fix L=5 and evaluate the sensitivity of the optimal threshold sequence $\mu^* = \{\mu_1, \mu_2, \ldots, \mu_5\}$ as we vary the retention rate ($\gamma$), discount factor ($\beta$), improvement cost ($c^+$), and gaming cost ($c^-$). We define a base environment with the following settings: $\beta = 0.8$, $\gamma = 0.9$, $c^+ = 1.0$, $c^- = 0.7$, and $r = 1.0$. Notably, we set the leg-up factor $\delta = 0$ for these tests to isolate the effects of the primary costs. For each sweep, we hold the other four parameters constant and vary the target parameter. At each step, we use the VALUEITERATE subroutine from Section E.1 with a discretized attribute space. The algorithm iteratively identifies the largest possible thresholds that satisfy the incentive compatibility constraints for promotion at each level.

Figure 9 displays the results of these ablation tests. As seen in Figure 9a, as $\gamma$ increases, all thresholds in the $L = 5$ system demonstrate a strictly increasing trend. This behavior aligns with the intuition that higher retention rates effectively reduce the long-term unit cost of honest improvement. This also leads to an increased capacity for higher thresholds for the principal, because when a greater portion of an agent's effort is retained over time, the principal can set significantly higher qualification standards while still maintaining the agent's incentive to choose improvement over gaming. It is also shown that the gap between successive levels also expands as retention rate increases. Because the effort is less wasteful due to slower depreciation, the principal can demand larger jumps between promotions without discouraging the agent.

Figure 9b displays the impact of the agent's farsightedness, represented by the discount factor $\beta$, on the optimal threshold sequence. Similar to the effects of $\gamma$, higher values of $\beta$ allow the principal to maintain higher thresholds across all levels. As $\beta$ increases, the optimal thresholds for all levels grow, indicating that a more patient agent is more willing to undertake costly efforts for future rewards. In Figure 9c, we see that there is a clear downward trend across all thresholds as $c^+$ increases. As honest effort becomes more expensive for the agent, the principal must adjust their thresholds to remain feasible. If thresholds were held constant while improvement cost rose, agents would eventually find gaming to be the only viable economic path to promotion. This suggests that in high improvement cost environments, promotion steps should be more incremental to prevent the total cost of reaching the next level from exceeding the discounted future rewards.

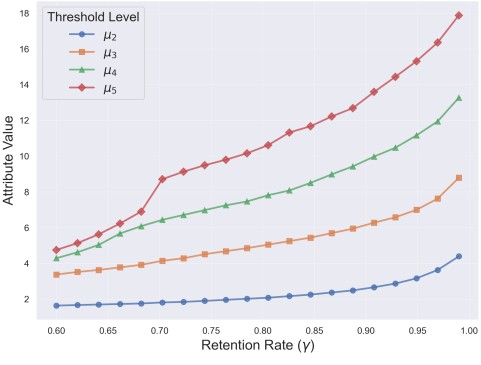

*(a)* Thresholds increase with higher retention ($\gamma$).

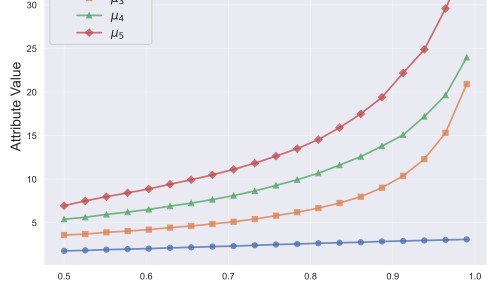

*(b)* Thresholds increase with higher discount factor ($\beta$).

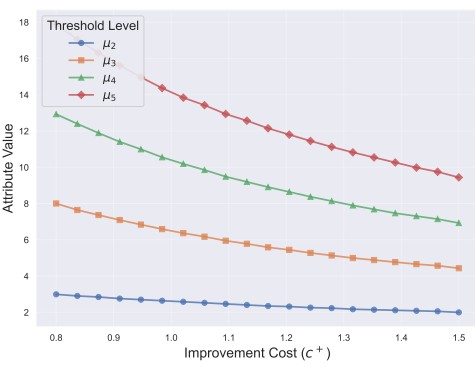

*(c)* Thresholds decrease as improvement becomes costly ($c^+$).

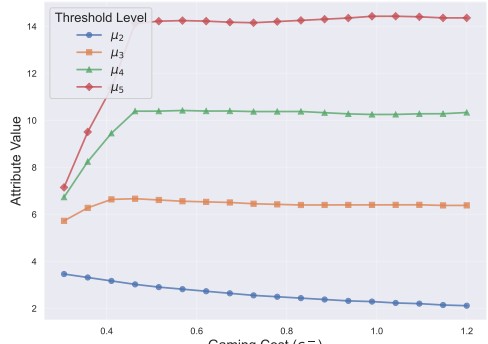

*(d)* Thresholds stabilize as gaming cost ($c^-$) increases.

*Figure 9.* The effect of varying environment parameters on each threshold in the L=5 level case.

Finally, Figure 9d shows the effect of the unit cost of gaming. The results show a divergence in how the principal sets standards for entry-level versus high-level promotions. At higher levels ($\mu_3, \mu_4, \mu_5$), the thresholds initially rise and plateau as $c^-$ increases. A higher gaming cost provides the principal with more room to set higher thresholds because the relative attractiveness of cheating to reach high levels will decrease. Once $c^-$ is sufficiently high, standards are no longer limited by the threat of gaming but by the agent's long-term capacity to sustain their attribute against depreciation. In contrast to higher levels, $\mu_2$ shows a gradual decrease as gaming becomes more costly. This suggests that at the lowest level of promotion, the principal may prioritize keeping the "first step" accessible. As the penalty for gaming increases, the principal can slightly lower the entry bar to encourage early honest effort without fearing that the agent will easily exploit the system through cheap manipulation.

### E.2.3. How the Interplay Between Farsightedness ($\beta$) and Retention ($\gamma$) Determines the Limits of Incentivizability

Next, we investigate the joint influence of an agent's farsightedness ($\beta$) and attribute retention ($\gamma$) on the multi-level mechanism. We perform a 2D grid sweep over $\beta$ and $\gamma$. For each coordinate, we construct an optimal sequence of thresholds using the greedy heuristic (Algorithm 1). The system parameters are fixed at: $c^+ = 1.0, c^- = 0.7, r = 1.0$, and $\delta = 0$. We set a maximum $L = 50$. Across the grid, we measure two primary metrics:

1. **Maximum incentivizable levels:** The total number of levels the algorithm successfully constructs before it is not longer possible to find a next threshold that satisfies the incentive compatibility constraints.

2. **Maximum reachable attribute:** The value of the highest threshold $\mu_L$ achieved.

Figure 10a presents the number of levels ($L$) successfully constructed by the greedy algorithm across varying values of $\gamma$ and $\beta$. The results display the transition in feasibility. A significant portion of the parameter space, particularly where

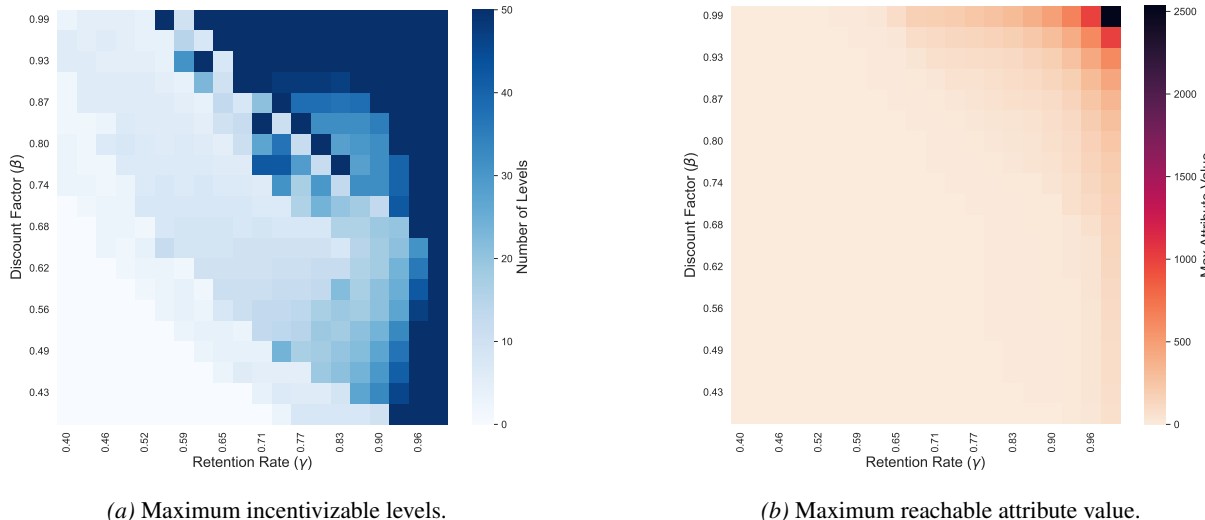

*(a)* Maximum incentivizable levels.                    *(b)* Maximum reachable attribute value.

*Figure 10.* Heatmaps across varying retention ($\gamma$) and discount factors ($\beta$). Darker regions indicate higher values.

both $\gamma$ and $\beta$ are low, remains entirely non-incentivizable (the white region). In this region, even the lowest promotion threshold cannot be sustained because the cost of honest effort, combined with high attribute depreciation outweighs the discounted future rewards. We also note that the transition from infeasibility to feasibility is non-linear. Figure 10a also shows that high retention ($\gamma \to 1$) can compensate for low farsightedness, and vice versa. Figure 10b complements this analysis by visualizing the maximum incentivizable attribute reached by the mechanism across the grid. The heatmap shows a concentrated spike in the top-right corner, where both $\beta$ and $\gamma$ approach unity. In this region, the principal can incentivize arbitrarily high qualification standards. Otherwise, the vast majority of the grid displays a maximum incentivizable attribute below 500. This confirms that without a leg-up effect ($\delta = 0$), the mechanism is capped by the agent's capacity to maintain an attribute against depreciation.

### E.2.4. ON THE MAXIMUM INCENTIVIZABLE ATTRIBUTE

This experiment sweeps the gaming cost ($c^-$) to empirically validate the bound in Theorem 2.1. We configure this experiment with the following parameters: $c^+ = 1.0, \beta = 0.8, \gamma = 0.9, r = 1.0$, and $\delta = 0$. Here, we measure the maximum incentivizable attribute for the agent.

Figure 11 illustrates the transition in the system's ability to incentivize honest effort based on the gaming cost $c^-$. The empirical results align with Theorem 2.1, as we see that the system fails to incentivize any improvement when $c^-$ is below $(1 - \beta\gamma)c^+$. In this region, gaming is so inexpensive that no sequence of thresholds can deter manipulation. Immediately after crossing the threshold $(1 - \beta\gamma)c^+$, we observe that the maximum incentivizable attribute becomes nonzero and begins increasing. Beyond this initial increase, the non-monotonicity observed arises from the sub-optimality of the greedy algorithm. Since the algorithm maximizes each threshold locally, it may select a value that constrains the feasible search space for subsequent levels. However, the greedy algorithm remains a valuable tool due to its computational efficiency.

### E.3. A Relaxed Objective Function and Experiment

### E.3.1. IMPLEMENTATION OF THE CMA-ES ALGORITHM

We maximize the principal's objective using the Covariance Matrix Adaptation Evolution Strategy (CMA-ES), configured with a **population size of 10** and a **30-iteration limit** for computational efficiency, subject to non-negativity constraints on the decision variables. To solve the principal's problem Equation (4), we iterate for each $L = 2$ to $L = 8$, run the CMA-ES algorithm on the fixed-length non-negative vector $\{\mu_1, \ldots, \mu_L\}$, obtain the principal's utility on the solution by CMA-ES, and choose the one (length $L$ and threshold vector $\{\mu_1, \ldots, \mu_L\}$) associated with the highest principal's utility. This gives rise to the solution recorded in Table 5.1.

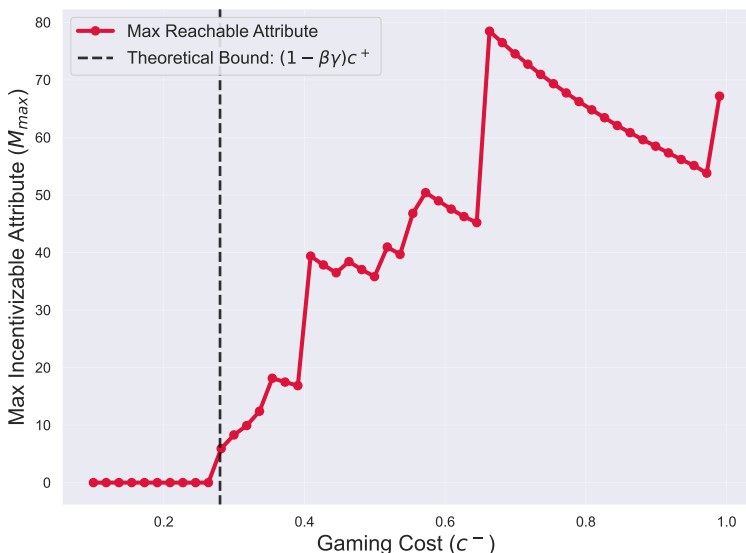

*Figure 11.* Maximum incentivizable attribute for agents versus cost of gaming $c^-$.

### E.3.2. PARAMETER SETUPS AND ADDITIONAL RESULTS FOR FIGURE 4

Figures 4a and 4b utilized the same parameter setups as Cases I and IV in Table 5.1, respectively. Specifically, other than $c^+$, $c^-$, $r^*$, and $\vec{\mu}^*$ (the latter two were solved by the CMA-ES algorithm), the rest parameters are set at $\beta = 0.8, \gamma = 0.8, \delta = 0.01, \alpha = 0.95, \xi = 0.01$, and $\lambda = 5$. For parameter setups of Cases II and III, we show the agent's trajectories in Figures 12a and 12b, respectively. It is easy to see that both cases demonstrate similar patterns as Figure 4a where the agent continuously improves until it stabilizes at the highest level and attribute.

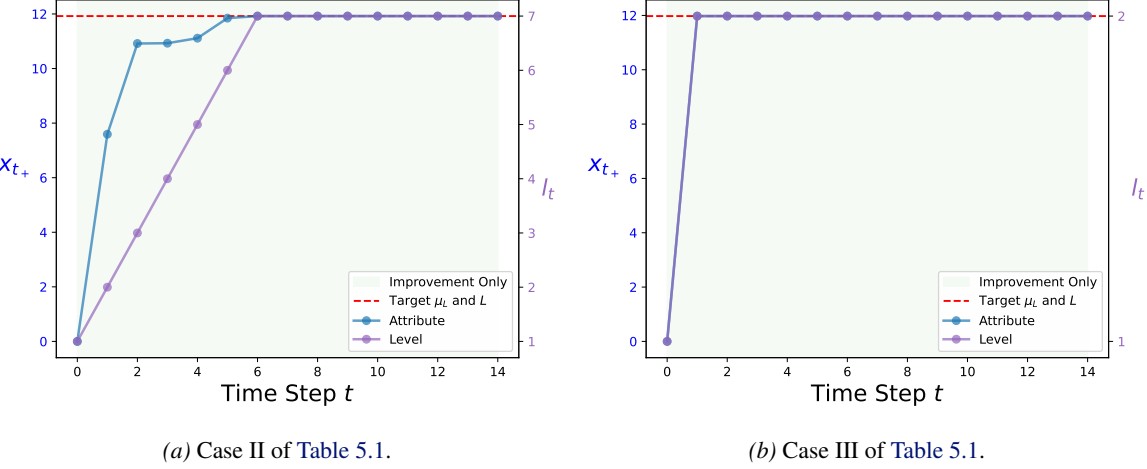

*(a)* Case II of Table 5.1.

*(b)* Case III of Table 5.1.

*Figure 12.* Agent state trajectory and strategy types at the principal's optimal design in Case II (a) and Case III (b) of Table 5.1.

Figure 4c represents the scenario where the threshold sequence defined in Theorem 4.2-(2) is infeasible for $\mathbf{P}(M, r)$ because the cost of gaming is too low. We adopt the following parameters: $\beta = 0.8, \gamma = 0.8, c^+ = 1.0, c^- = 0.365, r = 1$, and $\delta = 0.8$. With these values, one can verify that while the condition $r > \frac{1-\beta}{1-\gamma}c^+\delta$ holds, the condition for $c^-$ fails (specifically, $c^- \not\geq \max\{(1 + \frac{\beta\gamma}{2})(1 - \beta\gamma)c^+, \ \beta\gamma(1 - \beta^2\gamma^2)c^+\}$). However, the global assumption in Theorem 2.2 remains valid as $c^- > (1 - \beta\gamma)c^+$.

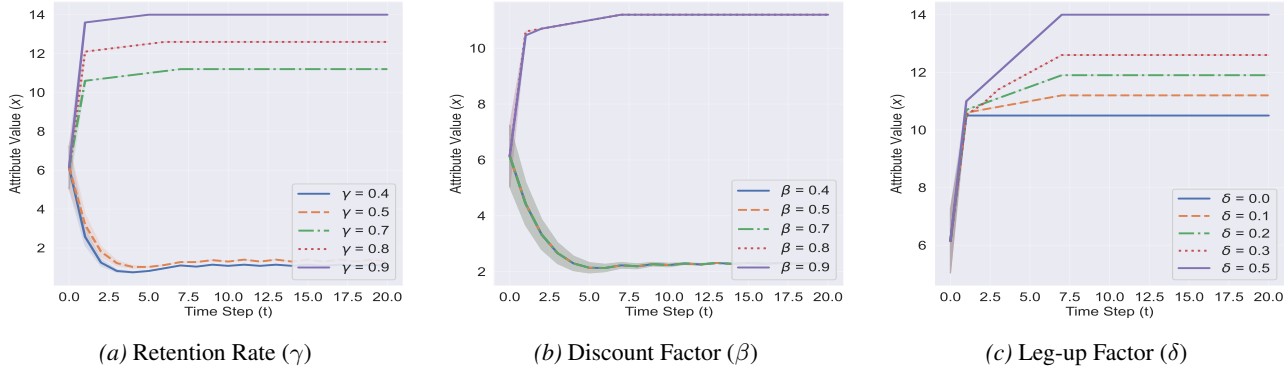

*(a)* Retention Rate ($\gamma$)     *(b)* Discount Factor ($\beta$)     *(c)* Leg-up Factor ($\delta$)

*Figure 13.* Empirical mean attribute trajectory ($\mathbb{E}_{x_0}[x_{t_+}^*]$) under the principal's optimal design with $\pm 1$ standard deviation.

### E.3.3. INFLUENCE OF SYSTEM PARAMETERS ON AGENT BEHAVIOR

We present the agent's trajectory and steady-state under the principal's optimal design in Figure 13. To ensure that the condition in Theorem 2.2 is met, the control variables in each experiment are as follows: $\gamma = 0.7, \beta = 0.8, c^+ = 1.0, c^- = 0.5, \alpha = 0.95, \lambda = 5.0,$ and $\xi = 0.01$. Figures 13a and 13b show that while both higher retention rate and discount factor can incentivize improvement efforts and higher stead-state attributes, the impact of the retention rate is more significant, consistent with the theoretical results in Section 3. Figure 13c shows that moderately higher leg-up effect is more effective in incentivizing improvement efforts than no leg-up effect at all.

We also examine the agent's strategic actions, specifically the improvement fraction. Figure 14 displays the mean improvement fraction, defined as $\frac{a^+}{a^+ + a^-}$, over the 20-step horizon. These plots isolate the behavioral/action response of the agents to the mechanism.

Figure 14a demonstrates that the stability of honest effort greatly depends on skill retention. When agents retain most of their effort ($\gamma = 0.9$), agents maintain a perfect improvement fraction (1.0) throughout the entire horizon. In this case, the maintenance cost of their qualification is low enough that they never find it optimal to resort to gaming. For moderate retention ($\gamma = 0.7, 0.8$), as retention decreases, agents initially frontload honest effort but eventually stabilize at a lower improvement fraction between 0.1 and 0.4. This indicates a shift to a mixture of improvement and gaming as the burden of countering attribute depreciation increases. For low retention ($\gamma = 0.4, 0.5$), the agents exhibit highly volatile behavior, alternating sharply between pure improvement and pure gaming. This oscillatory pattern suggests a recurrent struggle where agents repeatedly lose their qualifications and must "burst" effort to regain status.

A similar trend is seen in Figure 14b, agents who highly value future rewards ($\beta = 0.9$) exert moderate honest effort, which then stabilizes at an improvement fraction of approximately 0.15. As patience decreases, the strategy becomes less stable. For $\beta = 0.8$, the improvement fraction plateaus higher (around 0.2) than at $\beta = 0.9$, but for $\beta = 0.7$, the population eventually collapses into an oscillatory pattern, indicating a failure to maintain consistent honest effort. For low farsightedness ($\beta = 0.4, 0.5$), agents again maintain a pure improvement strategy (1.0) for a short initial period but quickly descend into oscillations between pure gaming and pure improvement. This suggests that without sufficient long-term incentive, agents are unable to sustain the qualifications required by the multi-level progression.

Finally, Figure 14c illustrates the impact of the leg-up factor $\delta$ on the long-term strategic behavior of the agents. This parameter represents the intrinsic attribute boost an agent receives upon reaching a higher level, simulating enhanced learning resources. In the absence of the leg-up effect, agents converge to a relatively high improvement fraction of approximately 0.4. In this case, agents must rely entirely on costly manual improvement to counter the effects of attribute depreciation. When there is moderate leg-up influence ($\delta = 0.1, 0.2, 0.3$), as $\delta$ increases, the steady state improvement fraction decreases, stabilizing between 0.15 and 0.25. Because the level itself contributes to attribute growth, the agent can maintain their qualification with lower ongoing investment of effort. At the high value of $\delta = 0.5$, the improvement effort drops to its lowest steady state level. This suggests that the natural boost provided by the leg-up is nearly sufficient to sustain the agent's position, requiring only minimal supplemental honest effort to offset the remaining depreciation.

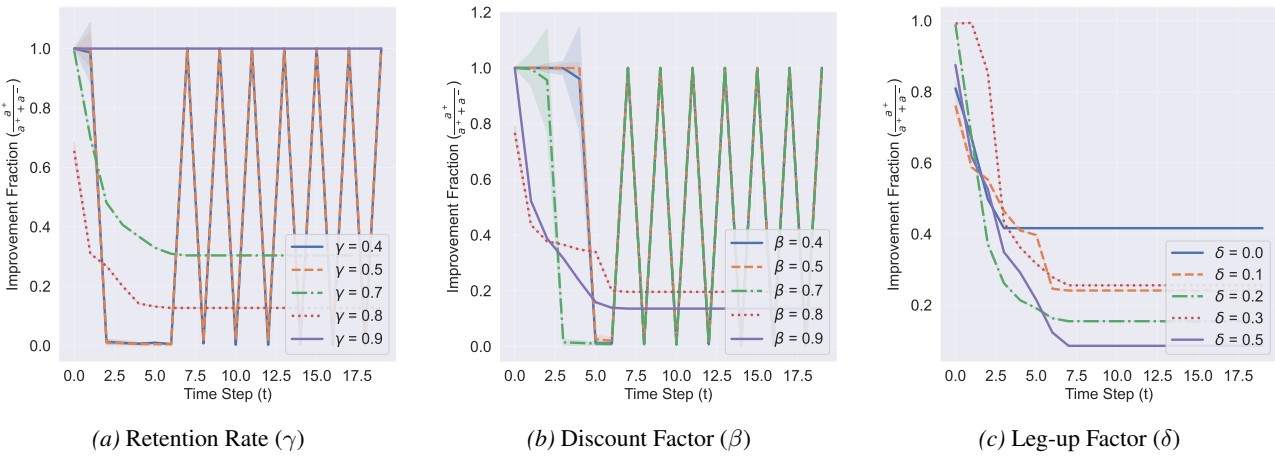

*(a)* Retention Rate ($\gamma$)  *(b)* Discount Factor ($\beta$)  *(c)* Leg-up Factor ($\delta$)

*Figure 14.* Each plot shows improvement fraction with a shaded region representing $\pm 1$ standard deviation.

### E.4. Restrictiveness of Feasibility Conditions

We analyze how restrictive the sufficient condition of Theorem 4.2–(2) is relative to Theorem 2.2 across a range of parameter settings. Both conditions place a lower bound on the gaming cost $c^-$; the question is how much stricter the sufficient condition is in practice.

For a given $(\beta, \gamma)$ and $c^+ = 1$, the two lower bounds on $c^-$ are:

- Theorem 2.2 (necessary): $c^- > (1 - \beta\gamma)c^+$
- Theorem 4.2-(2) (sufficient): $c^- \geq \max\{(1 + \beta\gamma^2)(1 - \beta\gamma),\ \beta\gamma(1 - \beta^2\gamma^2)\} c^+$

**Analytical bound comparison.** Figure 15 plots both the absolute sufficient lower bound (left) and the gap ratio (right), defined as the sufficient bound divided by the necessary bound, over a $30 \times 30$ grid of $(\beta, \gamma) \in [0.4, 0.99]^2$. The gap ratio ranges from 1.06 to 1.97 across the entire grid, remaining strictly below 2. This confirms that the sufficient condition is at most approximately 2 times as strict as necessary, so the principal retains $30 - 50\%$ tolerance for parameter misspecification.

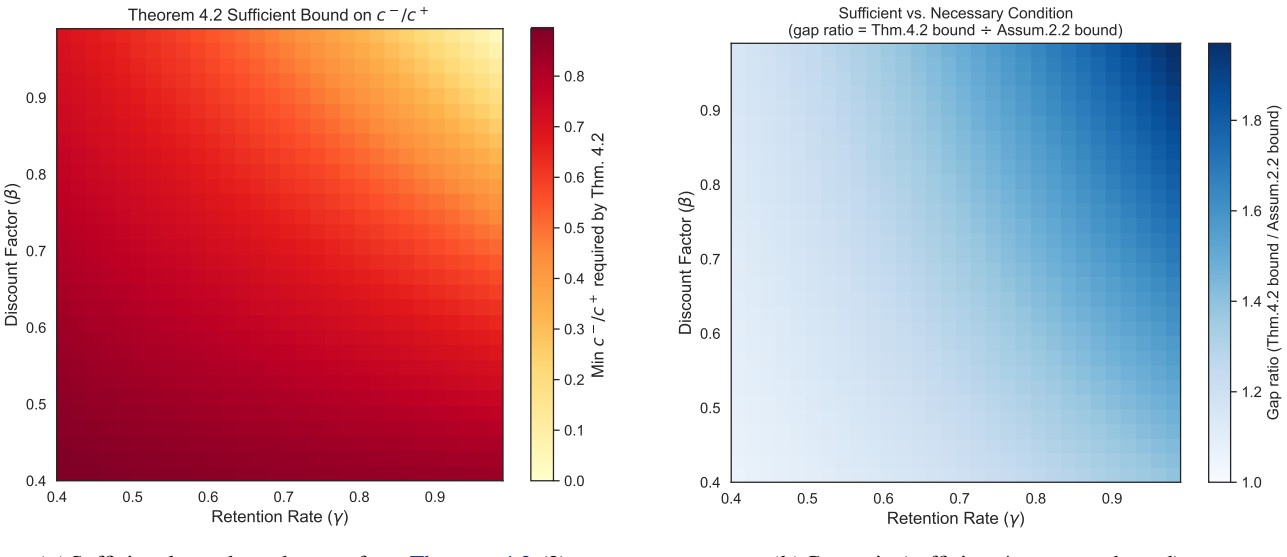

*(a)* Sufficient lower bound on $c^-$ from Theorem 4.2-(2).  *(b)* Gap ratio (sufficient / necessary bound).

*Figure 15.* Analytical comparison of feasibility conditions over $(\beta, \gamma)$ with $c^+ = 1$. The gap ratio is bounded between 1.06 and 1.97 across the full grid.

### E.5. Ternary vs. Binary Decision Rule

Our model uses a ternary selective classifier: promote ($z_t \geq \mu_{l+1}$), stay ($\mu_l \leq z_t < \mu_{l+1}$), or relegate ($z_t < \mu_l$). Here, we consider the following question: What does the abstention/stay option add relative to a purely binary promote-or-relegate design? Alkarmi et al. (2025) show in a one-shot strategic classification setting that abstention can serve as a deterrent to manipulation, making it costlier for less qualified agents to game; here we examine whether this benefit extends to our sequential multi-level setting. We implement a binary mechanism in which a single threshold separates promotion from relegation (no stay zone) and compare this with our ternary mechanism. We optimize both with CMA-ES under identical conditions, and compare outcomes.

The binary decision rule at level $l$ is: promote if $z_t \geq \tau_l$, relegate otherwise. The agent's value function and best-response are re-derived under this rule (removing the stay branch from the Bellman operator). Both mechanisms are optimized with CMA-ES over $L \in \{2, \ldots, 8\}$, using the same FICO-based initial population and principal utility function. Figure 16 sweeps $c^- \in [0.1, 0.9]$ while fixing all other parameters at their Case I values. This reveals a sharp critical regime around $c^- \approx 0.3$–$0.4$ where abstention becomes essential. In the infeasibility region ($c^- < 0.3$), both mechanisms perform similarly. However, once we move into the feasibility region, when $c^-$ is low (0.3–0.4), the binary mechanism achieves 100% gaming, but the ternary mechanism maintains 0% gaming. This indicates that our abstention or stay zone in the ternary classifier provides a buffer, preventing the binary (promote/relegate) pressure that often makes gaming the only rational path. The key takeaway is that abstention is *essential* near the feasibility boundary: when $c^-$ is just above $(1 - \beta\gamma)c^+$, the stay zone provides the critical buffer that discourages gaming without forcing relegation, allowing the mechanism to maintain incentive compatibility where the binary design fails. Thus, these results are consistent with Alkarmi et al. (2025).

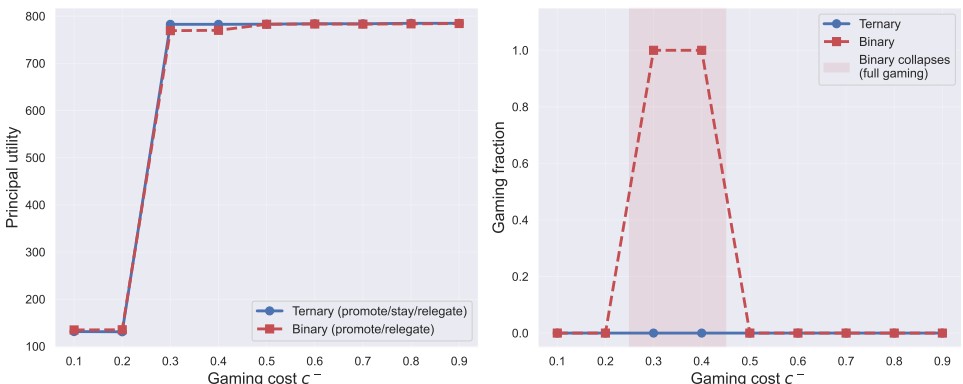

*Figure 16.* Ternary vs. binary utility and gaming fraction as $c^-$ varies. Prior to $c^- = 0.3$, the mechanism is in the infeasibility region. At low but feasible gaming costs, around $c^- \approx 0.3$–$0.4$, the ternary mechanism achieves 0% gaming (utility $\approx 783$) while the binary mechanism collapses to 100% gaming (utility $\approx 769$). Both mechanisms converge as $c^-$ increases above 0.5.

### E.6. Numerical Sensitivity Analysis

Both value iteration and the greedy threshold construction (Algorithm 1) rely on discretizing the attribute space with step size $\Delta x$ and truncation limit $\bar{X}$. We assess robustness to these choices along two axes using parameters $\beta = 0.8$, $\gamma = 0.9$, $c^+ = 1.0$, $c^- = 0.7$, $\delta = 0$.

**Grid resolution ($\Delta x$ sensitivity, $\bar{X} = 30$ fixed).** We vary $\Delta x \in \{0.15, 0.10, 0.075, 0.05, 0.03, 0.02, 0.01\}$. As shown in Figure 17, greedy thresholds $\mu_2$–$\mu_5$ and CMA-ES utility are essentially flat across all resolutions: Utility ranges from 1459.17 to 1459.21 ($< 0.003\%$ variation) and thresholds shift by at most 2.5% between the coarsest and finest grids, well within the binary-search precision $\varepsilon = 0.05$.

**Truncation limit ($\bar{X}$ sensitivity, $\Delta x = 0.05$ fixed).** We vary $\bar{X} \in \{15, 20, 30, 50, 80, 120, 180, 250, 350\}$. As shown in Figure 18, greedy thresholds stabilize for $\bar{X} \geq 30$ (within $\pm 0.15$ across all larger values), confirming that the default $\bar{X} = 30$ captures all relevant agent behavior. For $\bar{X} < 30$, the grid clips $\mu_5 \approx 22$, artificially limiting attainable levels. The monotone growth of CMA-ES utility with $\bar{X}$ reflects a wider simulation horizon, not instability, since a larger grid allows agents to accumulate attribute up to higher values over the finite horizon.

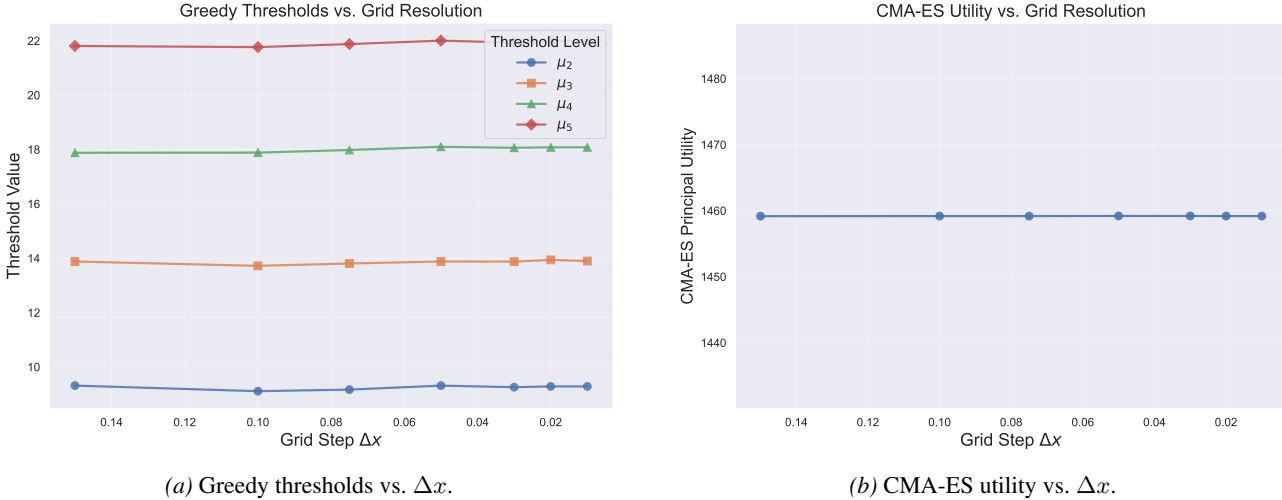

*(a)* Greedy thresholds vs. $\Delta x$.

*(b)* CMA-ES utility vs. $\Delta x$.

*Figure 17.* Greedy thresholds and CMA-ES utility are stable across all grid resolutions ($\bar{X} = 30$ fixed). Utility varies by less than $0.003\%$.

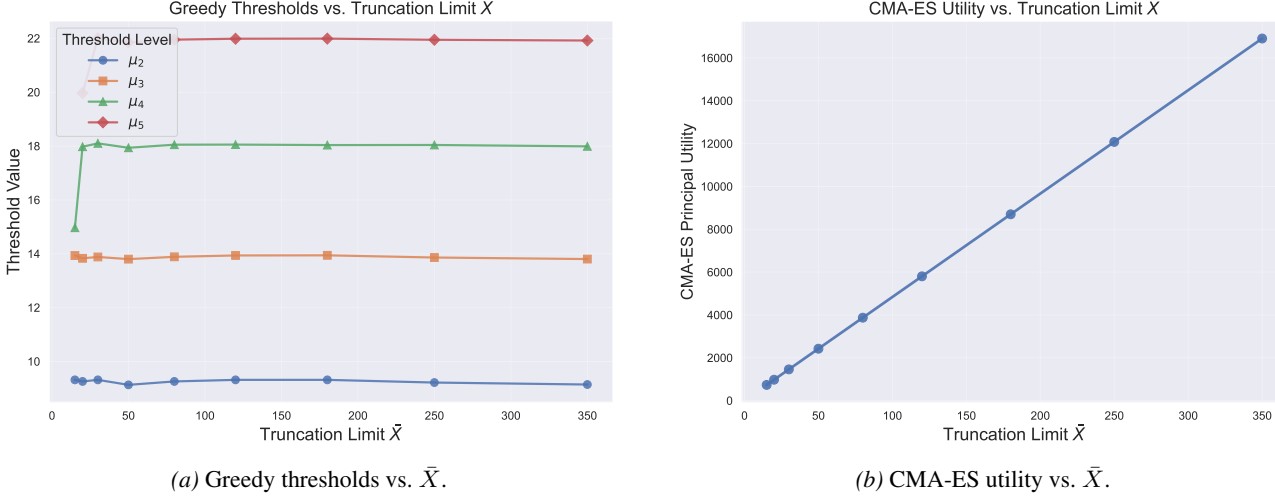

*(a)* Greedy thresholds vs. $\bar{X}$.

*(b)* CMA-ES utility vs. $\bar{X}$.

*Figure 18.* Greedy thresholds stabilize for $\bar{X} \geq 30$ ($\Delta x = 0.05$ fixed). CMA-ES utility grows monotonically with $\bar{X}$ due to the wider simulation horizon, while the threshold structure remains consistent.

