# OpenReview forum: "Multi-Level Strategic Classification: Incentivizing Improvement through Promotion and Relegation Dynamics"
_ICML.cc/2026/Conference — ICML 2026 regular_

### Official Review · Reviewer_SsK5 · 2026-02-27

**Soundness:** 3
**Presentation:** 3
**Significance:** 3
**Originality:** 3
**Overall Recommendation:** 4
**Confidence:** 3

**Summary:**

This paper studies a sequential strategic classification setting where an agent repeatedly faces a ladder of classifiers (levels) with increasing difficulty and reward. Each stage yields one of three outcomes—promotion, relegation, or staying in place—based on a ternary decision rule, and the agent can respond by investing in genuine improvement (which increases a latent “true qualification”) or gaming/cheating (which improves only the observable feature used by the classifier). The model captures inter-temporal incentives through (i) the agent’s farsightedness (discounting), (ii) retention/decay of true qualification over time, and (iii) a “leg-up” effect where attaining higher levels increases future qualification. The paper characterizes optimal agent behavior in the two-level case, analyzes feasibility and design of threshold schedules in the multi-level case, and argues that under certain conditions a principal can design thresholds that incentivize honest improvement and enable the agent to reach arbitrarily high levels through improvement alone. Numerical experiments and simulations support the theory and illustrate behavior across regimes.

**Compliance With Llm Reviewing Policy:**

Affirmed.

**Final Justification:**

After considering both the paper and the authors’ responses, I am increasing my score by one. I found the paper’s core idea original and interesting: it studies sequential strategic classification through a promotion–relegation mechanism and shows how threshold design, rather than only classifier weights, can shape long-term incentives. The modeling is coherent, the mechanism-design perspective is appealing, and the analysis provides useful intuition about how discounting, retention, and the leg-up effect interact to enlarge the incentivizable region relative to one-shot settings. The practical framing is also meaningful, since many real processes naturally involve staged progression rather than a single decision.

My original concerns were mainly about external validity and robustness: whether the strongest results depended too heavily on the specific additive leg-up assumption, whether abstention was truly essential, how restrictive the feasibility bounds were in practice, and how robust the conclusions were to heterogeneity, noise, and numerical approximation choices. The rebuttal addressed these concerns well. In particular, the authors provided additional analysis for diminishing and saturating leg-up effects, empirical comparisons showing the importance of the ternary abstention/stay option relative to a binary alternative, quantitative evidence that the sufficient feasibility bound is not excessively loose, and new robustness checks for heterogeneity, observation noise, and discretization sensitivity. These additions substantially strengthen the paper and make the scope and limitations of the results much clearer.

**Key Questions For Authors:**

1. The strongest feasibility results depend on the leg-up effect being strong enough and structured as an additive boost per level. How do your guarantees change if leg-up is diminishing (concave) or saturating with level, or if it varies across agents? Can you provide weaker sufficient conditions that still yield large (even if not unbounded) attainable levels?

2. The model uses a ternary decision (promote / stay / relegate) via a selective classifier. Can you isolate what abstention/staying adds relative to a purely binary promote/relegate design? Are there regimes where abstention is essential for discouraging gaming, rather than simply improving smoothness of dynamics?

3. Your feasibility conditions include lower bounds on the gaming cost. How restrictive are these bounds numerically across representative parameter settings? A plot or table of feasible vs infeasible regions with interpretable axes would help readers judge practical applicability.

4. Many real settings involve heterogeneity (different costs, different retention, different responsiveness to leg-up) and noise in observed features. Which parts of your analysis extend cleanly to these settings, and which would likely fail? Even a discussion of the “first obstacle” to extension would be helpful.

5. The numerical methods rely on discretization and interpolation choices. How sensitive are the reported feasibility boundaries and constructed threshold sequences to grid resolution and chosen truncation limits? A brief sensitivity check would increase confidence.

**Limitations:**

Partially. The paper recognizes important modeling constraints (e.g., the role of leg-up and the conditions on costs), but it would benefit from a clearer, more direct discussion of how strong these assumptions are in realistic domains, and how conclusions might change with (i) saturating leg-up, (ii) multi-dimensional attributes/features, (iii) observation noise and imperfect classifiers, and (iv) population-level concerns such as fairness or distribution shift. It would also help to clarify what the mechanism can still guarantee when assumptions are violated (e.g., improving outcomes up to a bounded level rather than arbitrarily high attainment).

**Strengths And Weaknesses:**

Soundness: The modeling is coherent and the paper does a good job connecting the agent’s dynamic optimization problem to the principal’s threshold-design objective. The two-level analysis provides useful intuition about when the agent improves, games, or does nothing, and it clarifies how discounting and retention change the “impossibility region” relative to one-shot strategic classification. The multi-level results are compelling in that they provide constructive threshold sequences and explicit parameter conditions for feasibility, including a sharp contrast between settings with and without the leg-up effect. A key caveat is that the strongest positive result (arbitrarily high levels via honest effort) relies heavily on the leg-up mechanism and on lower bounds on gaming cost that may be restrictive in practice. While these assumptions can be plausible in some domains, the paper would benefit from more discussion of how sensitive the feasibility results are to weaker or more realistic variants (e.g., diminishing returns in leg-up, correlated/nonlinear dynamics, or partially noisy observables). Additionally, because the agent is modeled essentially as a single scalar-attribute decision-maker, it remains unclear how robust the incentives remain when extending to heterogeneous populations, uncertainty in measurement, or strategic interactions across many agents.

Presentation: The paper is generally well structured: it motivates the need for sequential formulations, defines a clean promotion/relegation model, and proceeds from two-level analysis to multi-level design and then to numerical validation. The narrative is easy to follow and the main results are stated with conditions that make it possible to understand “when the theorem applies.” That said, the work would read more strongly with a compact practical guide to interpreting key parameters (retention, leg-up, costs) and how one might estimate them in a real application. There are also minor editorial issues (e.g., occasional confusing cross-references) that should be corrected to avoid distracting readers.

Significance: The paper tackles an important problem: how to design decision pipelines that encourage true improvement rather than gaming. The promotion–relegation ladder is a realistic abstraction for many processes (staged exams, certifications, multi-round hiring, progressive access control), and the message that sequential structure can enlarge the incentivizable region is valuable. The main limitation on significance is external validity: the results are strongest in regimes where higher levels meaningfully improve future capability (leg-up) and where gaming is not “too cheap.” In many real systems, gaming is cheap and leg-up effects may saturate, so it is important to understand whether the mechanism still helps meaningfully (even if it cannot guarantee unbounded attainment).

Originality: The main novelty is shifting focus from “time-varying classifier weights” to “difficulty progression via thresholds” in a sequential strategic classification framework with promotion/relegation dynamics and abstention/stay outcomes. This provides a distinct design lever and yields new theoretical analysis centered on inter-temporal incentives. The combination of retention, discounting, and leg-up in a staged classification ladder is a useful modeling contribution, and the constructive threshold schedule is an appealing mechanism-design style result.

---

> ### Author Rebuttal · Authors · 2026-03-31
>
> We thank the reviewer for the feedback and recognition that our paper represents “an appealing mechanism-design style result.” Below, we respond to the reviewer’s comments.
>
> 1. **Diminishing leg-up effect and weaker sufficient conditions**
> * **Level-dependent leg-up effect:** We discussed this generalization in the Discussion section, using $x\_{t+1}=\\gamma x\_t+\\delta\_{l\_{t+1}}$, where the leg-up depends (not necessarily linearly) on the current level. Replacing $\\delta l$ with $\\delta\_l$ in Eq. (7) and applying the proof of Thm 4.2 yields:
>   * (a) If $r \< \\frac{1-\\beta}{1-\\gamma} \\min\_{l,k \\geq 1} |\\frac{\\delta\_l-\\delta\_k}{l-k}|$, $P(M,r)$ is infeasible $\\forall M$.
>   * (b) If $r \\geq \\frac{1-\\beta}{1-\\gamma} \\max\_{l,k \\geq 1} |\\frac{\\delta\_l-\\delta\_k}{l-k}|$ and $c^+$ satisfies Thm 4.2, $P(M,r)$ is feasible $\\forall M \< \\max\_{l \\geq 1} \\delta\_l$ with $\\mu\_l \= \\frac{\\delta\_l}{1-\\gamma}$.
>   * If $\\delta\_l$ is concave/non-saturating, arbitrary attributes are incentivizable via decreasing threshold gaps; if $\\delta\_l$ saturates, the target attribute is bounded by that saturation point.
>
> * **Empirical Robustness to Saturation:** New experiments confirm robustness to four leg-up changes: constant, linear, $\\delta\\sqrt{l}$, and $\\delta\\log(1+l)$. For Case I params ($\\beta=0.8, \\gamma=0.8, c^+=0.8, c^-=0.7, r=1.0$), principal’s utility remained consistent between 643.5-649.9, and the mean final attribute stabilized at \~12.0 regardless of model. The avg. gaming fraction remained at 0.0 across all cases. Even with saturation, strategic thresholds effectively prevent gaming and drive high, stable qualification.
>
> 2. **The role of abstention**
>
> The ternary “stay” zone provides a buffer, preventing the binary (promote/relegate) pressure that often makes gaming the only rational path. New experiments comparing our ternary mechanism with an optimized binary baseline (Case I params) show that the ternary design is essential when gaming costs are low (but still satisfy Assum. 2.2).
>
> | c− | Ternary Frac. of Gaming Agents | Binary Frac. of Gaming Agents | Utility Gain Ternary vs. Binary (%) |
> | :---: | :---: | :---: | :---: |
> | $\\leq 0.2$ (Unfeasible) | 0 | 0 | \-3.27 |
> | 0.3  | 0 | 1 | \+1.72 |
> | 0.4  | 0 | 1  | \+1.62 |
> | $\\geq 0.5$  | 0 | 0 | \+0.02 |
>
> For $c^- \\in \\{0.3, 0.4\\}$, the binary mechanism induces 100% gaming across the population, while the ternary mechanism holds gaming to 0%. This shows that abstention is a gaming deterrent that provides a buffer agents need to utilize the leg-up effect and build true qualification over time.
>
> 3. **Restrictiveness of the feasibility bounds**
>
> We ran a new experiment to quantify the restrictiveness of the feasibility bounds. Defining the gap ratio as (Thm. 4.2 bound)/(Assum. 2.2 bound), we find that this ratio ranges from 1.06 to 1.97 across $\\beta, \\gamma \\in \[0.4,0.99\]$. Thus, our sufficient condition is at most twice as restrictive as the necessary condition. At base params ($\\beta, \\gamma=0.8$), the gap ratio is 1.51: Assum. 2.2 requires $c^-/c^+ \> 0.36$ while Thm 4.2 requires $c^-/c^+ \\geq 0.54$. The Thm 4.2 bound decreases as $\\beta \\cdot \\gamma$ increases, meaning high-retention/high-patience settings relax the absolute threshold on $c^-$. We will include heatmaps of the 4.2 bound and the gap ratio over $(\\beta, \\gamma)$ space in the revision.
>
> 4. **Heterogeneity and noise**
> * **Heterogeneity:** Our FICO evaluation (Sec 5.2) already addresses initial endowment heterogeneity ($x\_0$), where a single threshold sequence drives diverse populations toward improvement. To further test param variance, we simulated agents drawing ($c^+, c^-, \\gamma$) from uniform distributions centered on our Case I params. The principal designs a single mechanism using param mean. Even with this high heterogeneity, principal utility remained robust at 576.8 (vs. 650.7 baseline or \~11% less); while the gaming fraction was 0.25, the majority of the population still achieved honest improvement. For $\\beta$, the gaming fraction drops, but the more myopic agents tend to stagnate at middle levels, decreasing the principal’s utility by \~28%.
> * **Robustness to Observation Noise:** We conducted more experiments on our baseline Case I, where noise $\\epsilon \\sim N(0, \\sigma^2)$ is added to the $z$ before classification. Our mechanism was highly resilient: even at $\\sigma \= 1$, utility only slightly decreased (638.4 vs. 647.3 baseline) and the average gaming fraction remained at 0\.
>
> 5. **Resolution and truncation limit sensitivity**
>
> We ran a new experiment to evaluate the stability of our Greedy (Alg 2\) and CMA-ES methods against discretization choices. Varying $\\Delta x \\in \[0.01, 0.15\]$ and truncation limits $\\bar{X} \\in \[30, 350\]$ yielded consistent thresholds ($\\mu\_l$) and utility, with fluctuations $\<1\\%$, confirming that our numerical methods are robust to grid and truncation params.

---

> > ### Author Rebuttal · Reviewer_SsK5 · 2026-04-03
> >
> > Thank you for the detailed rebuttal. The response addresses my main concerns well. I especially appreciate the added analysis on diminishing/saturating leg-up, the ternary versus binary comparison clarifying the role of abstention, the quantitative discussion of the feasibility-gap restrictiveness, and the additional robustness checks for heterogeneity, observation noise, and numerical sensitivity. Overall, the rebuttal substantially strengthens the paper and adequately addresses my main concerns.

---

> > > ### Author Response · Authors · 2026-04-06
> > >
> > > Thank you for your rigorous and thoughtful review! Your comments regarding the leg-up effect and the restrictiveness of our feasibility bounds were particularly valuable, as they prompted us to generalize our theoretical results and conduct the new gap-ratio analysis. We are also thankful for your suggestion of comparing binary vs. ternary mechanisms. Thank you for your guidance in strengthening the depth of our paper. We will ensure all new heatmaps and experiments are included in the final version.

---

### Official Review · Reviewer_joPz · 2026-03-12

**Soundness:** 3
**Presentation:** 4
**Significance:** 2
**Originality:** 2
**Overall Recommendation:** 5
**Confidence:** 3

**Summary:**

The authors propose a non-weight-centric framework for sequential strategic classification that models how agents adapt their behavior as they move through a sequence of classifiers that require increasing effort, while accounting for discounted rewards and the gradual degradation of the agent’s attribute over time.

**Compliance With Llm Reviewing Policy:**

Affirmed.

**Final Justification:**

The authors responded to my and other reviewers' concerns. I raised my score from 4 to 5.

**Key Questions For Authors:**

*Please see the weaknesses section for additional questions and concerns*

* How does x_t influence agents’ choice of actions?
- From the modeling, it looks like |L| = |T|. What are the implications of these not being equal?
* What are the implications of agents (not) knowing a priori that there are 5 levels?
- What is the relationship between thresholds \mu and weights \theta at different levels, and how does it affect an agent’s choice of actions to take?

**Limitations:**

No. Several modeling assumptions and design choices substantially influence the results but authors do not adequately discuss the limitations associated with them.

**Strengths And Weaknesses:**

**Strengths**

* The authors present an interesting perspective on sequential strategic classification, showing that the incentivizable region can be strictly larger than in the one-shot formulation.
- The paper is clearly written, and both the theoretical and empirical results are easy to follow.
* The authors provide numerical experiments to complement the theoretical findings.



**Weaknesses**



* The authors assume one-dimensional features, and under the action formulation (line 122), actions are also one-dimensional and lie in the feature space. While this is a clean formulation, it doesn’t sufficiently capture many practical settings where features are multidimensional, and actions operate outside the feature space with complex or indirect effects on multiple features.  For example, different features may have varying predictive power, and the interactions among features might influence the costs of actions.

- How does x_t influence agents’ choice of actions? For example, a student who begins a course with knowledge well above the threshold x_t \ge \mu_{l+1} may be more likely to pursue genuine improvement (and skip levels) because improving could be cheaper for them than attempting to game the system.

* It might be worth defining z_t+1 as well, since it’s actually what the principal interacts with. Given that z_t = x_t+ + a_t-, then z_t+1 is x_t+1 + a_t-

- It seems possible to me that an agent at time t with state x_t might interact with an earlier level, such as L_{t-2}. These scenarios do not appear to be considered, yet they seem plausible, particularly in settings where agents may abstain or stay.

* It is also plausible that an agent could skip a level. That is, an agent at time t could be interacting with L_{t+2}.

- ``We assume \theta is pre-determined, e.g., via estimation on separately collected non-strategic data.`` This approach appears suboptimal to me and could lead to unfair outcomes across different groups of agents with varying initial x states.

* ``where \theta is the model weight shared across levels ... Without loss of generality, we will set \theta = 1 for the rest of the paper.`` In this setup, z is directly comparable to the thresholds, but ideally, different levels should have distinct weightings that reflect both the progression of acquired skills and their relevance to each level. For example, at lower levels, even small changes in z might have a significant impact on decisions.

- In my opinion, there is a misalignment between the theoretical modeling setup and the numerical setup. In particular, using FICO scores does not fully capture the theoretical framework of the problem, which involves a sequential progression of classifiers, cumulative effort, reward discounting, and the gradual degradation of the agent’s attribute over time.

* Since each level may require distinct skills, some building on lower levels and others not, it is plausible that the costs of gaming and improvement could increase, decrease, or diverge relative to each other. It would be valuable to consider the full range of possible cost structures, their dependence on z_t and thresholds at each level, and their implications for agents’ incentives.

- Several assumptions and modeling choices the authors consider might make the mechanism challenging to implement in practical settings. For example, beyond those already noted here and in the questions section, it is unclear to me why authors optimize over the number of levels since the cases considered in the motivational setup would need a fixed number of levels with potentially varying thresholds.  The optimization problem (Eqn 3) is also formulated in an ideal to eliminate gaming, determine the optimal number of levels, and corresponding thresholds.

*Please see the questions section for more concerns and questions*



**Miscellaneous**

Some agents may be rejected or choose to opt out at level 1, since in practical settings, \mu_1 may not be zero. As a result, the model could be interacting with an already biased subset of agents. Apart from this, the modeling setup raises numerous additional fairness considerations.

---

> ### Author Rebuttal · Authors · 2026-03-31
>
> We thank the reviewer for the feedback and recognition that our framework represents an “interesting perspective” with a “clearly written “narrative and an “easy to follow” presentation of the theoretical and empirical results.
>
> **1\. One-dimensional features vs. multidimensional actions**
>
> Section 6 discusses generalizability to multi-dimensional attribute vectors. Specifically, a feature vector is projected onto a "summarizing" scalar (or last-layer output), upon which classification depends via thresholding. For example, loan approvals depend on a single scalar credit score, which is a summary of many factors. In our model, the 1-D attribute represents this summarizing scalar. A rational agent will choose the set of weights $\\{w\_i\\}$ that yields the highest attribute boost per unit cost. Consequently, the effective action cost in our 1-D formulation represents the $\\{w\_i\\}$-weighted average cost across all feature dimensions.
>
> **2\. The influence of $x\_t$ and level skipping**
> Section 3 and Figure 3(b)-(c) fully characterize the agent’s behavior based on their initial attribute. In the multi-level setting (without skipping), our simulations show that an agent with a high initial endowment exerts no effort in the early rounds, as their baseline attribute is sufficient for promotion. They only begin exerting effort when the thresholds become more difficult. Whether this effort is genuine or gaming depends entirely on the principal's threshold design. We will explicitly clarify this dynamic in the revision.
>
> Regarding level-skipping: we agree this is a natural and interesting scenario. In a reasonable scenario where all agents were pre-tested before entrance, sorted, and placed in the appropriate levels at the start, no agent would have an incentive to skip levels in all subsequent interactions.  Because this paper focuses on the fundamental question of whether improvement can be incentivized through thresholding, we have not included dynamic level-skipping but it is a valuable future direction.
>
> **3\. Implication of $|L| \= |T|$**
> We do not enforce $|L| \= |T|$. We assume an infinite time horizon, meaning there is no maximum timestep $T$. The reviewer may be referring to our finding that, under certain threshold sequences (e.g., the greedy algorithm in Section 5), the agent improves and is promoted at every single step until reaching the highest level. This is an emergent behavioral result of that specific threshold design, not a structural constraint of the model.
>
> **4\. “What are the implications of agents (not) knowing a priori that there are 5 levels?”**
> Our model operates under a full-information assumption, meaning the agent knows the total number of levels in the system. If agents do not know the number of levels a priori, it is reasonable to assume that the agent will behave more myopically (i.e. $\\beta \\to 0$). In this scenario, the impact is a gap between the farsightedness ($\\beta$) the principal assumes and the true $\\beta$ values of the agent. However, our algorithm is highly robust against this parameter misspecification.
>
> **5\. Relationship between thresholds $\\mu$ and weights $\\theta$**
> For any observable feature $z$, the classifier outcome at level $l$ is determined by $\\theta z \\geq \\mu\_l$. Because this inequality is invariant to positive scaling on both sides, and because we assume $\\theta$ is shared across levels (standard in ordinal regression literature), we can set $\\theta \= 1$ without loss of generality. Our paper primarily focuses on the design of the threshold sequence $\\mu\_l$ to incentivize improvement. However, the model naturally extends to more complex scenarios where both weights and thresholds are dynamically updated over time.
>
> **6\. “It might be worth defining z\_t+1 as well”**
> We are happy to use $z\_{t\_+}$ (following Footnote 2\) to denote the feature observed after the agent’s action to ensure clarity.
>
> **7\. Misalignment of the FICO dataset with the theoretical framework**
>
> We agree that the FICO score is not a dynamic dataset; it is only used as the starting population-wide initial attribute ($x\_0$) distribution, and its evolution is based on our model. Unfortunately, we have not been able to find real-world datasets that track longitudinal behavioral responses to multi-level policies (or even one-shot strategic classification). We believe our semi-synthetic approach represents a best effort until such datasets become publicly available.
>
> **8\. Optimizing over the number of levels**
>
> While our motivational example uses a fixed number of levels, Eq. (3) generalizes our model by allowing the principal to control both the number of levels ($L$) and thresholds ($\\mu$). Implementation costs are implicitly captured by $L$ (e.g., more levels/tests increase costs). Additionally, Eq. (4) provides a relaxed, practical version of this objective as a weighted sum of principal goals.

---

> > ### Author Rebuttal · Reviewer_joPz · 2026-04-01
> >
> > The questions I had have been addressed. I have raised my score.

---

> > > ### Author Response · Authors · 2026-04-06
> > >
> > > Thank you for your detailed and thoughtful feedback. We are glad that our clarifications on the number of levels, time horizon, and dimensionality were clear. We will incorporate these clarifications into our final version to ensure the narrative is as clear as possible. Thank you again for your time and for helping us refine our work!

---

### Official Review · Reviewer_kfdi · 2026-03-13

**Soundness:** 3
**Presentation:** 3
**Significance:** 3
**Originality:** 3
**Overall Recommendation:** 5
**Confidence:** 3

**Summary:**

This paper studies a multi-level strategic classification problem: the principal designs L classifiers to decide whether to promote (relegate) the agent to the next (previous) level; the agent can exert genuine effort to improve its true attribute or exert cheap effort to fake a feature, or do both, to pass the classification. The dynamic model has two effects on the true attribute: skill retention, where the attribute degrades over time, and leg-up, where a higher level helps to improve the attribute.  The entire optimization problem is a bi-level problem where inside is the agent solving an infinite-horizon discounted MDP problem, and outside is the principal optimizing the thresholds of the classifiers to incentivize the agent to improve its true attribute.  The main theoretical results are:

1. The formulation of the multi-level strategic classification problem itself, and a result showing that the incentivizable region is strictly larger than that under a one-shot formulation.
2. A full characterization of the agent's best-response MDP strategy in the two-level case.  In particular, the importance of the leg-up effect in incentivizing higher attributes is highlighted.
3. Characterization of the feasibility of the principal's problem, including conditions under which pure improvement actions are incentivizable up to some target attribute, and the design of a sequence of classifiers that induce an agent to attain a high level of attribute through honest efforts.

**Compliance With Llm Reviewing Policy:**

Affirmed.

**Final Justification:**

See the Rebuttal Acknowledgement.

**Key Questions For Authors:**

(Q1) How robust are your results (such as the theoretical guarantee of the designed classifiers) against parameter misspecification?  Can you provide some theoretical or empirical analysis?

**Limitations:**

A potential limitation is the sensitivity of the results in the parameters of the model.  I asked this question above.

**Strengths And Weaknesses:**

**Strengths:**

(++) Prior works (Jin et al., 2022; 2023) showed that improvement incentivization is challenging in one-shot strategic classification scenarios and proposed to use external instruments (such as transfer mechanisms, collaborations, or recourse policies) to address this problem.  In contrast, this paper shows that incentivizing improvement is much more promising in a multi-period setting, and moreover, it can be achieved by classifier design only.  This new and interesting insight is a significant contribution to the literature.

(+) Some other insights from this work are also interesting, including: for example, a high retention rate ($\gamma$) is more effective than the agent's farsightedness $(\beta)$ in incentivizing improvement; the optimal threshold sequence should equalize the attribute depreciation and leg-up effects; and the intricate divisions of parameter space under which different agent behaviors are optimal in the two-level case.  These insights might be useful for practice and for future research.

(+) The techniques seem to be non-trivial.  The characterization of the agent's best response in the two-level case requires a careful case analysis, while the multi-level case involves the design and analysis of a value-iteration MDP algorithm.



**Weaknesses:**

(-) An obvious weakness is that the full characterization of the agent's best response is limited to two levels.  But I think this weakness is minor due to the complexity of the problem as well as the practical relevance of the two-level case.

(-) The results, especially the design of the principal's classifiers, depend on the exact knowledge of the parameter, such as the agents' costs of exerting genuine effort and cheap efforts, $c^+, c^-$, the attribute retention rate $\gamma$, and the agent's discount factor $\beta$.  Those parameters of the agent are hardly fully known by the principal.  It is unclear whether the results are robust against parameter misspecification.

---

> ### Author Rebuttal · Authors · 2026-03-31
>
> We thank the reviewer for the positive evaluation and for recognizing the paper as a "technically solid" contribution with  "significant contribution to the literature." We are particularly encouraged by the reviewer's highlight of our core insight: that while prior work required external instruments (like transfers or collaborations) to incentivize improvement, our work shows that **sequential classifier design alone is a powerful and sufficient mechanism to incentivize improvement**. We also appreciate the recognition of our non-trivial techniques. Below, we address the reviewer’s questions regarding parameter robustness and two-level case.
>
> 1. **Q1: “How robust are your results against parameter misspecification?”**
>
> We have now performed extensive additional numerical experiments in response to this comment.
>
> A summary and highlight table are below. Since figures cannot be included in the rebuttal, we will make sure to include them in the revision.
>
> * **Robustness:** We conducted new experiments to evaluate our mechanism’s robustness against the principal’s imperfect knowledge of the agent's parameters (e.g., their patience or skill retention):
>   * **Experimental Setup:** The principal designs a mechanism based on a "believed" parameter value, while the agents respond based on their "true" values.
>   * **Error Definition:** $\\epsilon \= (\\text{Believed} \- \\text{True}) / \\text{True}$.
>   * **True Parameter Baseline:** The "True" values were fixed at Case I settings: $\\beta \= 0.8$, $\\gamma \= 0.8$, $c^+ \= 0.8$, $c^- \= 0.7$, and $\\delta \= 0.01$.
>   * **Scope:** We tested $\\epsilon \\in \\{\\pm 0.1, \\pm 0.2, \\pm 0.3\\}$ across all key parameters: retention ($\\gamma$), farsightedness ($\\beta$), improvement cost ($c^+$), gaming cost ($c^-$), and the leg-up factor ($\\delta$). The table below highlights selected results (the full range shows consistent trends):
>
> | Parameter | Error (ϵ) | Believed | Utility Loss (%) | Avg. Gaming | Final Attribute |
> | :---: | :---: | :---: | :---: | :---: | :---: |
> | Retention (γ) | \-0.3 | 0.56 | 0.17% | 0 | 12.02 |
> | Retention (γ) | \+0.1 | 0.88 | 3.55% | 0 | 11.89 |
> | Gaming Cost (c−) | \-0.2 | 0.56 | 0.04% | 0 | 12.05 |
> | Improvement (c+) | \+0.2 | 0.96 | 0.05% | 0 | 12.04 |
> | Farsightedness (β) | \-0.1 | 0.72 | 0.11% | 0 | 12.02 |
> | Leg-up (δ) | \+0.3 | 0.013 | 0.01% | 0 | 12.03 |
>
> **Summary of Robustness Experiment Results:**
>
> * **Principal Utility:** Utility loss is low. Across most parameters, even a 30% error in estimation leads to a loss of less than 1% in realized utility.
> * **Gaming Behavior:** In nearly all univariate misspecification cases, the **average gaming fraction remains at 0.0**. This demonstrates that the mechanism does not collapse into manipulation if the principal’s beliefs are slightly off.
> * **Agent Qualification:** The steady-state mean attribute (the "true" qualification of the population) remains stable at approximately 10.7–12.0, even under significant parameter misspecification.
>
> Beyond our empirical results, our framework’s theoretical guarantees are structurally robust to parameter uncertainty because our sufficient condition (Theorem 4.2) is more conservative than the necessary condition (Assumption 2.2). Empirically the principal has 30%–50% 'room' to be wrong. This creates a buffer; a principal using a slightly wrong parameters often remains within the feasible region.
>
> We hope these clarifications regarding the data-driven design and our new results on parameter uncertainty provide a clearer picture of the mechanism's practical potential.
>
> 2. **“The full characterization of the agent’s best response is limited to two levels. But I think this weakness is minor due to the \[problem's\] complexity as well as \[its\] practical relevance.”**
>
>    We thank the reviewer for identifying this and for recognizing the complexity of the $L$-level best-response. We would like to clarify why the 2L characterization is foundational and how it provides heuristics for the general case:
>
>    **A Model for Bounded Rationality:** While a perfectly farsighted agent solves the full MDP, real-world strategic agents often exhibit bounded rationality, focusing primarily on the immediate next promotion. Our 2L analysis (Theorems 3.1 and 3.2) provides a rigorous characterization of this "local" best-response. If a principal can ensure each step of a multi-level ladder is locally incentive-compatible based on our 2L results, they can successfully guide even boundedly rational agents to high attainment.
>
>    The 2L characterization also provides the mathematical foundation for our Greedy Heuristic. In Theorem 5.1,  we prove that global feasibility can be achieved by satisfying local 2L incentive constraints at each step. Thus, the 2L characterization is sufficient to induce global improvement across an arbitrary number of levels by ensuring local incentive compatibility at every transition.

---

> > ### Author Rebuttal · Reviewer_kfdi · 2026-04-04
> >
> > My concerns have been adequately addressed.
> >
> > The new experiments do seem to suggest that the results are robust, which is good.
> >
> > The added bounded rationality motivation for the study of 2 level special case is also interesting.
> >
> > So I raised score from 4 to 5.

---

> > > ### Author Response · Authors · 2026-04-06
> > >
> > > Thank you for your positive evaluation and for the insightful questions that helped us further clarify the scope of our work. We are particularly glad that our discussion on the 2-level characterization as a foundation for bounded rationality and the greedy heuristic were clear. We are also thankful for the question regarding parameter misspecification, which led us to conduct our new robustness experiments. We truly appreciate your feedback and recognition of our work, and we will ensure the new robustness experiments and the expanded discussion on $L$-level best responses are included in the final version.

---

### Official Review · Reviewer_58Qe · 2026-03-13

**Soundness:** 3
**Presentation:** 3
**Significance:** 3
**Originality:** 3
**Overall Recommendation:** 4
**Confidence:** 3

**Summary:**

This paper studies the problem of incentivizing genuine improvement in strategic classification settings. The paper proposes a sequential formulation of strategic classification based on a multi-level promotion and relegation framework. In this setting, agents repeatedly interact with a sequence of classifiers of increasing difficulty and reward. Passing a classifier leads to promotion to a higher level, while failing leads to relegation to a lower level. The model captures several inter-temporal effects, including reward discounting, attribute retention, and a “leg-up” effect where higher levels reinforce an agent’s capability.

The authors analyze the agent’s optimal long-term strategy and study how a principal can design classifier thresholds across levels to incentivize genuine improvement rather than gaming. The paper provides a theoretical characterization of the agent’s behavior in the two-level case and derives conditions under which a sequence of classifiers can induce agents to reach arbitrarily high levels through honest improvement efforts.

**Compliance With Llm Reviewing Policy:**

Affirmed.

**Final Justification:**

A work needs some revision

**Key Questions For Authors:**

Please see weaknesses above

**Limitations:**

yes

**Strengths And Weaknesses:**

**Strength**:
- The paper provides an interesting perspective on the problem of incentivizing genuine improvement in strategic classification. By introducing a multi-level promotion–relegation mechanism, the framework attempts to address the well-known limitation of one-shot settings where agents often prefer gaming over costly improvement.

- The paper is generally well written and easy to follow. The authors place many of the more technical proofs in the appendix, which helps keep the main text focused on the key ideas and makes the overall presentation clearer.

- The paper includes a range of experiments that illustrate the behavior of the proposed mechanism under different settings, helping to support the theoretical analysis and providing additional insight into the dynamics of the model.

**Weaknesses**:

1. Strong behavioral assumptions on agents.
The proposed mechanism relies on several strong assumptions about agent behavior. In particular, agents are assumed to possess long-term planning ability (farsightedness), and their capabilities are assumed to evolve according to a stable retention structure. However, in many real-world strategic environments, agents may exhibit myopic behavior or bounded rationality, and improvements in capabilities may not be consistently retained over time. As a result, it remains unclear to what extent the incentive guarantees derived in the paper would hold under more realistic behavioral models, which may limit the external validity of the proposed mechanism.

2. The framework assumes that the principal can freely design a sequence of classifier thresholds to construct the proposed multi-level promotion and relegation mechanism. In many practical machine learning applications, however, classifiers are typically learned from data, and decision thresholds are not fully controllable mechanism variables. Instead, they are often constrained by the data distribution, model structure, and deployment requirements. It would therefore be helpful for the authors to clarify how such threshold sequences could be effectively designed or implemented in data-driven classification systems.

3. The empirical evaluation is entirely based on numerical simulations derived from the theoretical model. While these experiments illustrate the behavior of the proposed mechanism, it remains unclear how the framework would perform in more realistic environments or with semi-synthetic or real datasets.

---

> ### Author Rebuttal · Authors · 2026-03-31
>
> We thank the reviewer for the feedback and recognition that our paper effectively addresses the limitations of one-shot settings. Below, we respond to the reviewer’s comments.
>
> **1\. Strong behavioral assumptions**
>
> * **Justification of the current model:** Our goal is to show that multi-level strategic classification can (1) shrink the "impossibility region" of incentivizing improvement in one-shot settings and (2) induce purely honest actions toward high qualification. Adoption of simplifying assumptions allows tractable analysis of the principal’s problem ($P(M,r)$) in a clean, conceptual manner. In this sense, our study is a necessary foundational step: it provides a theoretical upper bound for what such a system can achieve and what the optimal design looks like under ideal conditions, upon which future work can build more complex/realistic models. The relaxed objective (Eqn 4\) is in fact one such relaxation that is easier to compute (via the CMA-ES algorithm) and highly robust against imperfect knowledge of problem parameters (see Point 3 below).
> * **Myopic behavior**: $\\beta$ controls how farsighted the agent is. $\\beta=0$ represents a pure myopic or greedy agent, who effectively treats the system as a one-shot problem as studied in prior work. As seen in Proposition 2.1, in such a case improvement incentivization relies on external interventions.
> * **Generalization and Robustness:** While we use linear retention, Section 6 generalizes these insights to any dynamic function $g(x\_{t\_+}, l\_{t+1})$. Further, new experiments (Point 3 below) show that even with a 30% parameter misestimate, the system remains stable with 0 gaming agents.
>
> **2\. Practical design in data-driven systems**
>
> Our framework can inform the design and implementation in a data-driven classification system in two ways:
>
> * **Optimization with Real-World Constraints:** The relaxed utility function and CMA-ES algorithm are well-suited for practical systems. As a black box tool, CMA-ES allows the principal to incorporate constraints such as accuracy requirements or deployment costs directly into the utility function. The algorithm iteratively learns the optimal threshold sequence for the underlying data distribution. Point 3 below confirms the principal can design these mechanisms even with significant parameter uncertainty.
> * **Online Learning Framework:** An important future direction is to frame this as an online learning problem where parameters are learned in real-time. The current paper is a necessary first step toward that goal, by establishing the theoretical, computational, and structural understanding of the ideal mechanism, providing a sound target for the online learning objective.
>
> **3\. More realistic environments/semi-synthetic/real datasets:**
>
> * **Semi-Synthetic Evaluation with Real-World Data:** The current evaluation is indeed semi-synthetic: the FICO credit score is used to model the initial attribute distribution ($x\_0$) of a real-world population, with evolution following our dynamic model. As longitudinal datasets tracking behavioral responses to multi-level (or even one-shot) classification are currently unavailable, this semi-synthetic approach represents a best effort to ground our theoretical results in real-world data.
> * **Robustness:** We conducted new experiments to evaluate our mechanism’s robustness against the principal’s imperfect knowledge of the agent's parameters:
>   * **Setup:** The principal designs a mechanism based on a "believed" parameter value, while the agents respond based on their "true" values.
>   * **Error Definition:** $\\epsilon \= (\\text{Believed} \- \\text{True}) / \\text{True}$.
>   * **True Parameter Baseline:** The "True" values were fixed at Case I settings: $\\beta \= 0.8$, $\\gamma \= 0.8$, $c^+ \= 0.8$, $c^- \= 0.7$, and $\\delta \= 0.01$.
>   * **Scope:** We tested $\\epsilon \\in \\{\\pm 0.1, \\pm 0.2, \\pm 0.3\\}$ across all key parameters: retention ($\\gamma$), farsightedness ($\\beta$), improvement cost ($c^+$), gaming cost ($c^-$), and leg-up ($\\delta$). Selected results are below for conciseness; the full range shows consistent trends:
>
> | Parameter | Error (ϵ) | Believed | Utility Loss (%) | Avg. Gaming | Final Attribute |
> | :---- | ----- | ----- | ----- | ----- | ----- |
> | γ | \-0.3 | 0.56 | 0.17 | 0 | 12.02 |
> | γ | \+0.1 | 0.88 | 3.55 | 0 | 11.89 |
> | c− | \-0.2 | 0.56 | 0.04 | 0 | 12.05 |
> | c+ | \+0.2 | 0.96 | 0.05 | 0 | 12.04 |
> | β | \-0.1 | 0.72 | 0.11 | 0 | 12.02 |
> | δ | \+0.3 | 0.013 | 0.01 | 0 | 12.03 |
>
> * **Summary of Robustness:**
>   * **Low principal utility loss:** Across most parameters, even a 30% error in estimation results in \<1% loss in realized utility.
>   * **Low gaming:** In nearly all univariate misspecification cases, the **average gaming fraction remains at 0.0**.
>   * **Agent qualification:** Steady-state mean population attribute remains in (10.7–12.0), even under significant parameter misspecification.

---

> > ### Author Rebuttal · Reviewer_58Qe · 2026-04-04
> >
> > Thanks, the rebuttal is helpful. I raised my score 3 to 4

---

> > > ### Author Response · Authors · 2026-04-06
> > >
> > > Thank you for the constructive discussion and for re-evaluating our work! We are thankful for your comments regarding practical design and robustness. We are pleased that our new results addressed your concerns, and will ensure this new robustness data and the discussion on data-driven implementation are included in our camera-ready version. Thank you again for your time and for helping us improve the practical grounding of our work.

---

### Decision · Program_Chairs · 2026-04-30

**Decision:**

Accept (regular)

**Comment:**

The paper studies a setting where a self-interested agent in a promotion-relegation hierarchy is repeatedly evaluated by classifiers and shows that a system designer can set classifier thresholds so that genuinely improving one's ability, rather than gaming the classifier, is each agent's optimal long-term strategy.

Reviewers found the paper technically solid and the multi-stage framing novel compared with one-shot formulations and, after the rebuttal, there was consensus that the paper is a strong contribution. In the revised version, the authors are encouraged to carefully address all comments raised by the reviewers by incorporating the additional arguments and results they provided during the rebuttal. From my perspective, the most consistent concern is that the mechanism requires the principal to know agent-side parameters that are difficult to observe (e.g., farsightedness, skill retention, and the relative cost of honest improvement versus gaming). The rebuttal's robustness experiments should be fully incorporated in the main paper to address that. In addition, the authors should explicitly discuss the simulated nature of their experiments as a limitation and highlight data-driven directions as an important avenue for future work.